# Understanding the Effect of Stochasticity in Policy Optimization

**Jincheng Mei** [1 3 *]  **Bo Dai** [3]  **Chenjun Xiao** [1 3]  **Csaba Szepesvári** [2 1 † *]  **Dale Schuurmans** [3 1 †]

[1]University of Alberta  [2]DeepMind  [3]Google Research, Brain Team  [†]equal advising

## Abstract

We study the effect of stochasticity in on-policy policy optimization, and make the following four contributions. *First*, we show that the preferability of optimization methods depends critically on whether stochastic versus exact gradients are used. In particular, unlike the true gradient setting, geometric information *cannot* be easily exploited in the stochastic case for accelerating policy optimization without detrimental consequences or impractical assumptions. *Second*, to explain these findings we introduce the concept of *committal rate* for stochastic policy optimization, and show that this can serve as a criterion for determining almost sure convergence to global optimality. *Third*, we show that in the absence of external oracle information, which allows an algorithm to determine the difference between optimal and sub-optimal actions given only on-policy samples, there is an inherent trade-off between exploiting geometry to accelerate convergence versus achieving optimality almost surely. That is, an uninformed algorithm either converges to a globally optimal policy with probability 1 but at a rate no better than $O(1/t)$, or it achieves faster than $O(1/t)$ convergence but then must fail to converge to the globally optimal policy with some positive probability. *Finally*, we use the committal rate theory to explain why practical policy optimization methods are sensitive to random initialization, then develop an ensemble method that can be guaranteed to achieve near-optimal solutions with high probability.

## 1  Introduction

Policy optimization is a central problem in reinforcement learning (RL) that provides a foundation for both policy-based and actor-critic RL methods. Until recently it had generally been assumed that methods based on following the policy gradient (PG) [1] could not be guaranteed to converge to globally optimal solutions, given that the policy value function is not concave. However, this assumption has been contradicted by recent findings that policy gradient methods can indeed prove to converge to global optima, at least in the tabular setting. In particular, the standard softmax PG method with a constant learning rate has been shown to converge to a globally optimal policy at a $\Theta(1/t)$ rate for finite MDPs [2], albeit with challenging problem and initialization dependent constants [3, 4]. Several techniques have been developed to further improve standard PG and achieve better rates and constants. For example, adding *entropy regularization* has been shown to produce faster $O(e^{-c \cdot t})$ convergence ($c > 0$) to the optimal regularized policy [2, 5, 6]. By exploiting natural geometries based on Bregman divergences, *natural PG (NPG) or mirror descent (MD)* have been shown to achieve better constants than standard PG [7, 5] and faster $O(e^{-c \cdot t})$ rates, with [5, 6] and without regularization [8]. Alternative policy parameterizations, such as the escort parameterization, have been shown to improve the constants achieved by softmax and yield faster plateau escaping [3].

---

[*]Correspondence to: Jincheng Mei and Csaba Szepesvári {jmei2,szepesva}@ualberta.ca

35th Conference on Neural Information Processing Systems (NeurIPS 2021).

More recently, a *geometry-aware normalized PG (GNPG)* approach has been proposed to exploit the non-uniformity of the value function, achieving faster $O(e^{-c \cdot t})$ rates with improved constants [9].

A key observation is that each of these four techniques—*(i)* entropy regularization, *(ii)* NPG (or MD), *(iii)* alternative policy parameterization, and *(iv)* GNPG—accelerates the convergence of standard softmax PG by better exploiting the geometry of the optimization landscape. In particular, entropy regularization makes the regularized objective behave more like a quadratic [2, 5, 6], which significantly improves the near-linear character of the softmax policy value [2]. Natural PG (or MD) performs non-Euclidean updates in the parameter space, which is quite different from the Euclidean geometry characterizing standard softmax PG updates [7, 5, 8]. The escort policy parameterization induces an alternative policy-parameter relation [3]. GNPG exploits the non-uniform smoothness in the optimization landscape via a simple gradient normalization operation [9].

However, these advantages have only been established for the true gradient setting. A natural question therefore is whether geometry can also be exploited to accelerate convergence to global optimality in *stochastic* gradient settings. In this paper, we show that in a certain fundamental sense, the answer is *no*. That is, there exists a fundamental trade-off between leveraging geometry to accelerate convergence and overcoming the noise introduced by stochastic gradients (possibly infinite); in particular, no uninformed algorithm can improve the $O(1/t)$ convergence rate without incurring a positive probability of failure (i.e. diverging or converging to a sub-optimal stationary point).

The conditions used in vanilla stochastic gradient convergence analysis, *i.e.*, unbiased and variance-bounded gradient estimator [10], has been exploited to attempt to explain such a trade-off in policy gradients [11, 6]. However, the bounded variance requires the sample policy to be bounded away from zero everywhere, which is impractical. Meanwhile, a variant of NPG can converge even with unbounded variance [12]. These gaps raise the question that if not the bounded variance, then what is the key factor to ensure the convergence of stochastic policy optimization algorithms? Motivated by this question, we introduce the concept *committal rate* to characterize the update behaviors, which significantly affect whether convergence to a correct solution can be guaranteed in the stochastic on-policy setting. In particular, we make the following contributions.

- *First*, we illustrate the anomaly that the preferability of policy optimization algorithms (softmax PG vs. NPG and GNPG) changes dramatically depending on whether true versus on-policy stochastic gradients are considered, and reveal the impracticality and unnecessity of a bounded variance requirement in Section 2;
- *Second*, we introduce the concept of the *committal rate* in Section 3 to characterize the aggressiveness of an update, which provide us tools for analyzing the stochasticity effect in convergences;
- *Third*, we use the committal rate to study general stochastic policy optimization behaviors rigorously and reveal the inherent geometry-convergence trade-off in Section 4;
- *Finally*, we explain the sensitivity to random initialization in practical policy optimization algorithms. From these results, we then develop an ensemble method that can achieve fast convergence to global optima with high probability in Section 5.

## 2   Understanding Algorithm Preferability in On-line Policy Optimization

To illustrate the key aspects of policy optimization methods and their comparative preferability, it suffices to consider deterministic, single-state, finite-action Markov decision processes (MDPs). The main results extend to general finite MDPs, but for clarity of exposition we restrict attention to one-state MDPs.

A deterministic, single-state, finite-action MDP can be simply be specified by an action space is $[K] := \{1, 2, \ldots K\}$ and a $K$-dimensional reward vector $r \in \mathbb{R}^K$. The problem is to maximize the expected reward of a parametric policy $\pi_\theta$,

$$\max_{\theta:[K] \to \mathbb{R}} \mathbb{E}_{a \sim \pi_\theta(\cdot)} [r(a)]. \tag{1}$$

where $\pi_\theta$ is parameterized by $\theta$ using the standard softmax transform,

$$\pi_\theta(a) = \frac{\exp\{\theta(a)\}}{\sum_{a' \in [K]} \exp\{\theta(a')\}}, \quad \text{for all } a \in [K]. \tag{2}$$

Without loss of generality, we assume there exists a unique optimal action $a^* = \arg\max_{a \in [K]} r(a)$, hence there exists a unique optimal deterministic policy $\pi^*$ such that ${\pi^*}^\top r = \sup_{\theta \in \mathbb{R}^K} \pi_\theta^\top r = r(a^*)$. We make the following assumption on the reward.

**Assumption 1** (Positive reward). $r(a) \in (0, 1], \forall a \in [K]$.

### 2.1 Exact Gradient Setting

It is known that Eq. (1) is a non-concave maximization over the policy parameter $\theta$ [2]. Nevertheless, it has recently become better understood how policy gradient (PG) methods still converge to global optima for Eq. (1) when exact gradients are used. To illustrate the main considerations, we focus on the following three representative algorithms that have recently been proved to achieve convergence to global optima but at different rates: softmax policy gradient (PG), natural PG (NPG), and geometry-aware normalized PG (GNPG), while similar conclusions can be drawn for other variants [12, 13].

#### 2.1.1 Softmax PG

The standard softmax PG method is specified by the following update.

**Update 1** (Softmax PG, true gradient). $\theta_{t+1} \leftarrow \theta_t + \eta \cdot \frac{d\pi_{\theta_t}^\top r}{d\theta_t}$, where $\frac{d\pi_\theta^\top r}{d\theta} = \left( diag(\pi_\theta) - \pi_\theta \pi_\theta^\top \right) r$, and thus $\frac{d\pi_\theta^\top r}{d\theta(a)} = \pi_\theta(a) \cdot (r(a) - \pi_\theta^\top r)$ for all $a \in [K]$.

As shown in Mei et al. [2], the convergence of this update to a globally optimal policy, given exact gradients, can be established by considering the following non-uniform Łojasiewicz (NŁ) inequality,

**Lemma 1** (NŁ, [2]). $\left\| \frac{d\pi_\theta^\top r}{d\theta} \right\|_2 \geq \pi_\theta(a^*) \cdot (\pi^* - \pi_\theta)^\top r$.

By considering smoothness of $\pi_\theta^\top r$, Mei et al. [2] shows that the progress in each iteration of PG can be lower bounded by the squared norm of the gradient, $\left\| \frac{d\pi_{\theta_t}^\top r}{d\theta_t} \right\|_2^2$, which leads to a $O(1/t)$ rate.

**Proposition 1** (PG upper bound [2]). *Using Update 1 with $\eta = 2/5$, we have $(\pi^* - \pi_{\theta_t})^\top r \leq 5/(c^2 \cdot t)$ for all $t \geq 1$, such that $c = \inf_{t \geq 1} \pi_{\theta_t}(a^*) > 0$ is a constant that depends on $r$ and $\theta_1$, but it does not depend on the time $t$. In particular, if $\pi_{\theta_1}(a) = 1/K\ \forall a$ then $c \geq 1/K$.*

**Proposition 2** (PG lower bound [2]). *For sufficiently large $t \geq 1$, Update 1 with $\eta \in (0, 1]$ exhibits $(\pi^* - \pi_{\theta_t})^\top r \geq \Delta^2 / (6 \cdot t)$, where $\Delta = r(a^*) - \max_{a \neq a^*} r(a) > 0$ is the reward gap of $r$.*

**Remark 1.** *The constant dependence of PG follows a $\Omega(1/c)$ lower bound for one-state MDPs [3], while $c$ can be exponentially small in terms of the number of states for general finite MDPs [4].*

To summarize, using $\eta \in O(1)$, softmax PG achieves convergence to a global optima, but with a $\Theta(1/t)$ rate that exhibits poor constant dependence.

#### 2.1.2 Natural PG (NPG)

An alternative method, natural PG (NPG) [14], provides the prototype for many practical policy optimization methods, such as TRPO and PPO [15, 16]. NPG is based on the following update.

**Update 2** (Natural PG (NPG), true gradient). $\theta_{t+1} \leftarrow \theta_t + \eta \cdot r$.

For softmax policies, it turns out that Update 2 is identical to mirror descent (MD) with a Kullback-Leibler (KL) divergence. Therefore a standard MD analysis shows that Update 2 achieves convergence to a global optimum at a rate of $O(1/t)$ [7]. Very recently, work concurrent to this submission [8] has shown that Update 2 actually enjoys a much faster $O(e^{-c \cdot t})$ rate. In fact, here too we can establish the same $O(e^{-c \cdot t})$ rate, but using a simpler argument based on the following variant of the NŁ inequality for natural gradients. These results are new to this paper. **Due to space limitation, we postpone all the proofs to the appendix.**

**Lemma 2** (Natural NŁ inequality, continuous). $\left\langle \frac{d\pi_\theta^\top r}{d\theta}, r \right\rangle \geq \pi_\theta(a^*) \cdot \Delta \cdot (\pi^* - \pi_\theta)^\top r$.

**Lemma 3** (Natural NŁ, discrete). *Let $\pi'(a) := \frac{\pi(a) \cdot e^{\eta \cdot r(a)}}{\sum_{a'} \pi(a') \cdot e^{\eta \cdot r(a')}}, \forall a \in [K]$, where $\eta > 0$. Then,*

$$(\pi' - \pi)^\top r \geq \left[ 1 - \frac{1}{\pi(a^*) \cdot (e^{\eta \cdot \Delta} - 1) + 1} \right] \cdot (\pi^* - \pi)^\top r. \tag{3}$$

In particular, by using a non-Euclidean update and analysis, the progress of each iteration of NPG can be lower bounded by the larger bound $\left\langle \frac{d\pi_{\theta_t}^\top r}{d\theta_t}, r \right\rangle$ instead of the weaker bound $\left\| \frac{d\pi_{\theta_t}^\top r}{d\theta_t} \right\|_2^2$ established for standard PG. Based on this inequality, one can easily establish a much faster $O(e^{-c \cdot t})$ convergence to a globally optimal solution for NPG, making it far preferable to PG if true gradients are available.

**Theorem 1** (NPG upper bound). *Using Update 2 with any $\eta > 0$, we have, for all $t \geq 1$,*

$$\left(\pi^* - \pi_{\theta_t}\right)^\top r \leq \left(\pi^* - \pi_{\theta_1}\right)^\top r \cdot e^{-c \cdot (t-1)}, \tag{4}$$

*where $c := \log\left(\pi_{\theta_1}(a^*) \cdot \left(e^{\eta \cdot \Delta} - 1\right) + 1\right) > 0$ for any $\eta > 0$, and $\Delta = r(a^*) - \max_{a \neq a^*} r(a) > 0$.*

### 2.1.3 Geometry-aware Normalized PG (GNPG)

The Geometry-aware Normalized PG (GNPG) update is investigated in [9] to accelerate the convergence of PG by exploiting local smoothness properties of the optimization landscape.

**Update 3** (Geometry-aware Normalized PG (GNPG), true gradient). $\theta_{t+1} \leftarrow \theta_t + \eta \cdot \frac{d\pi_{\theta_t}^\top r}{d\theta_t} / \left\| \frac{d\pi_{\theta_t}^\top r}{d\theta_t} \right\|_2$.

The analysis in [9] focuses on exploiting non-uniform smoothness (NS) rather than improving the NŁ inequality as for NPG above.

**Lemma 4** (NS, [9]). *The spectral radius of Hessian matrix $\frac{d^2 \pi_\theta^\top r}{d\theta^2}$ is upper bounded by $3 \cdot \left\| \frac{d\pi_\theta^\top r}{d\theta} \right\|_2$.*

Given this NS property, [9] shows that the progress in GNPG can be lower bounded by the larger quantity $\left\| \frac{d\pi_{\theta_t}^\top r}{d\theta_t} \right\|_2$ instead of the weaker $\left\| \frac{d\pi_{\theta_t}^\top r}{d\theta_t} \right\|_2^2$ for standard PG. Then, using the same NŁ inequality as for PG, GNPG also converges to a globally optimal solution at rate $O(e^{-c \cdot t})$. Again, one naturally concludes that GNPG is preferable to PG if exact gradients are used.

**Proposition 3** (GNPG upper bound [9]). *Using Update 3 with $\eta = 1/6$, we have, for all $t \geq 1$,*

$$\left(\pi^* - \pi_{\theta_t}\right)^\top r \leq \left(\pi^* - \pi_{\theta_1}\right)^\top r \cdot e^{-\frac{c \cdot (t-1)}{12}}, \tag{5}$$

*where $c = \inf_{t \geq 1} \pi_{\theta_t}(a^*) > 0$ does not depend on $t$. If $\pi_{\theta_1}(a) = 1/K$, $\forall a$, then $c \geq 1/K$.*

## 2.2 The Anomalous Behaviour of Some On-policy Stochastic Gradient Updates

Although the above results show that exploiting geometric information can allow linear convergence to an optimal solution given true gradients—obviously $O(e^{-c \cdot t})$ represents an exponential speedup over the $\Omega(1/t)$ lower bound for standard PG—it is critical to understand whether such advantages can also be obtained in the more natural stochastic gradient setting. Given the previous results, it would seem natural to prefer accelerated algorithms over PG in practice, and there is some evidence that such thinking has become mainstream based on the popularity of TRPO and PPO over PG. Indeed, TRPO and PPO are often interpreted as instances of NPG and the faster convergence of NPG is used to explain their empirical success. However, by more closely examining the behavior of these algorithms when true gradients are replaced by on-policy stochastic estimates, serious shortcomings begin to emerge, as empirically observed in Chung et al. [12], and it is far from obvious that similar advantages from the true gradient case might be recoverable in the more practical stochastic scenario.

We begin by examining the behavior of the previous algorithms in the context of on-policy stochastic gradients. To enable this analysis, first note that each of the above PG methods, Updates 1 to 3, can be adapted to the stochastic setting by using on-policy importance sampling (IS) to provide an unbiased estimate of the true reward. We do not make assumptions like each action is sufficiently explored, since $\pi_{\theta_t}$ is the behaviour policy as well as the policy to be optimized. It is possible that $\pi_{\theta_t}$ approaches a near deterministic policy, ruling out positive results based on such assumptions [11].

**Definition 1** (On-policy IS). *At iteration $t$, sample one action $a_t \sim \pi_{\theta_t}(\cdot)$. The IS reward estimator $\hat{r}_t$ is constructed as $\hat{r}_t(a) = \frac{\mathbb{I}\{a_t = a\}}{\pi_{\theta_t}(a)} \cdot r(a)$ for all $a \in [K]$.*

**Remark 2.** *We consider sampling one action in each iteration, but the results continue to hold for sampling a constant $B > 0$ mini-batch of actions. A significant limitation of our results is that the reward is observed without noise, which is an idealized case. It remains to be seen which conclusions of this work can be extended to the more general case when the rewards are observed in noise.*

In the next subsections we consider the mentioned three on-policy update rules. As we shall see, only the first update rule, vanilla policy gradient with softmax parameterization is sound.

### 2.2.1 Softmax PG

**Update 4** (Softmax PG, on-policy stochastic gradient). $\theta_{t+1} \leftarrow \theta_t + \eta \cdot \frac{d\pi_{\theta_t}^\top \hat{r}_t}{d\theta_t}$, where $\frac{d\pi_{\theta_t}^\top \hat{r}_t}{d\theta_t(a)} = \pi_{\theta_t}(a) \cdot (\hat{r}_t(a) - \pi_{\theta_t}^\top \hat{r}_t)$ for all $a \in [K]$.

Using the IS reward estimate, the softmax PG is unbiased and bounded by constant:

**Lemma 5.** *Let $\hat{r}$ be the IS estimator using on-policy sampling $a \sim \pi_\theta(\cdot)$. The stochastic softmax PG estimator is unbiased and bounded, i.e., $\mathbb{E}_{a\sim\pi_\theta(\cdot)}\left[\frac{d\pi_\theta^\top \hat{r}}{d\theta}\right] = \frac{d\pi_\theta^\top r}{d\theta}$, and $\mathbb{E}_{a\sim\pi_\theta(\cdot)}\left\|\frac{d\pi_\theta^\top \hat{r}}{d\theta}\right\|_2^2 \le 2$.*

These observations imply that stochastic softmax PG converges to a global optimum in probability, which was also proved by Chung et al. [12]. Here we use the non-uniform smoothness in Lemma 4 to prove that $\mathbb{E}_{a_t\sim\pi_{\theta_t}(\cdot)}\left[(\pi^* - \pi_{\theta_t})^\top r\right] \in O(1/\sqrt{t}) \to 0$ as $t \to \infty$, which implies that $\lim_{t\to\infty} \Pr\left((\pi^* - \pi_{\theta_t})^\top r > 0\right) \to 0$, i.e., sub-optimality converges to 0 in probability.

**Theorem 2.** *Using Update 4, $(\pi^* - \pi_{\theta_t})^\top r \to 0$ as $t \to \infty$ in probability.*

### 2.2.2 NPG

Similarly, we can use on-policy IS estimation to adapt NPG to the stochastic setting.

**Update 5** (NPG, on-policy stochastic gradient). $\theta_{t+1} \leftarrow \theta_t + \eta \cdot \hat{r}_t$.

Although the NPG is unbiased, its variance can be possibly unbounded in the on-policy setting.

**Lemma 6.** *For NPG, we have, $\mathbb{E}_{a\sim\pi_\theta(\cdot)}[\hat{r}] = r$, and $\mathbb{E}_{a\sim\pi_\theta(\cdot)}\|\hat{r}\|_2^2 = \sum_{a\in[K]} \frac{r(a)^2}{\pi_\theta(a)}$.*

The variance becomes unbounded as $\pi_\theta(a) \to 0$, which predicts trouble when using the standard analysis for stochastic gradient methods[2] (e.g., [10]). In fact, we provide a more direct result showing that stochastic NPG has a positive probability of converging to a sub-optimal deterministic policy.

**Theorem 3.** *Using Update 5, we have: (i) with positive probability, $\sum_{a\neq a^*} \pi_{\theta_t}(a) \to 1$ as $t \to \infty$; (ii) $\forall a \in [K]$, with positive probability, $\pi_{\theta_t}(a) \to 1$, as $t \to \infty$.*

This result extends the result of [12] for the two-action ($K = 2$) case only. The intuition is that the stochastic NPG accumulates too much probability on sampled sub-optimal actions and cannot recover due to the "vicious circle" between sampling and updating.

### 2.2.3 GNPG

Finally, we consider the stochastic version of GNPG.

**Update 6** (GNPG, on-policy stochastic gradient). $\theta_{t+1} \leftarrow \theta_t + \eta \cdot \frac{d\pi_{\theta_t}^\top \hat{r}_t}{d\theta_t} \Big/ \left\|\frac{d\pi_{\theta_t}^\top \hat{r}_t}{d\theta_t}\right\|_2$.

Unfortunately, this estimator involves a ratio of random variables, and its bias can be large. As for NPG we can show that stochastic GNPG fails with positive probability in the stochastic case.

**Theorem 4.** *Using Update 6, we have, $\forall a \in [K]$, with positive probability, $\pi_{\theta_t}(a) \to 1$, as $t \to \infty$.*

## 2.3 Why Consider the On-policy Stochastic Setting?

The findings of the previous sections are summarized in Table 1. The two methods that converge faster when the exact gradient is available are exactly those that fail in the worse possible way in the on-policy setting. This raises the question of should one even consider the on-policy setting?

One possible reason to consider this setting is because on-policy sampling is the simplest and most straightforward approach to extend algorithms developed for the "exact gradient" setting and with

---

[2]Standard treatment of stochastic approximation algorithms does deal with unbounded noise in a controlled way to still get positive results [17], which means that bounded variance is far from being necessary.

|  | Softmax PG | NPG | GNPG |
|---|---|---|---|
| True gradient | converges $\Theta(1/t)$ ✓✓ | converges $O(e^{-c \cdot t})$ ✓✓✓ | converges $O(e^{-c \cdot t})$ ✓✓✓ |
| Stochastic on-policy | converges in prob. ✓ | fails w.p. $> 0$ ✗ | fails w.p. $> 0$ ✗ |

Table 1: Convergence properties of softmax PG, NPG and GNPG in the alternative settings.

a minor twist, Occam's razor dictates that one should consider simple solutions before considering more complex ones. Indeed, off-policy algorithms are more complex with many more choices to be made and while having the extra freedom may ultimately be useful (and even perhaps necessary), it is worthwhile to first thoroughly examine whether this complexity can be avoided. Indeed, there is some empirical evidence that the simple, on-policy approach may sometimes be a reasonable one: The method PPO [16] uses on-policy sampling and yet, remarkably, it achieved outstanding results on challenging tasks, a good example of which is to learn dexterous in-hand manipulation [18].

A second reason is that the on-policy setting presents unique challenges and as such is interesting on its own for learning about how to design and reason about stochastic methods. Indeed, the standard approach in analyzing stochastic update rules, such as SGD, is to start with the assumption that the gradient estimates are unbiased and have a uniformly bounded variance. This has been used both in the analysis of SGD [10], and later adopted to policy gradient methods [11, 6, 19, 20]. However, such conditions are only *sufficient* and not necessary as the numerous results in the literature of the analysis and design of stochastic approximation methods also show [17]. In fact, the bounded variance assumption can be difficult to satisfy. For example, in the problems studied here this assumption requires that the probabilities induced by a behaviour policy are bounded away from $0$ everywhere [12], which is impractical for large state and action spaces and impossible when they are infinite.

Another observation that suggests that it is worthwhile to consider methods which potentially unbounded variance is made by Chung et al. [12] who explored the role of baselines in policy optimization. They show that variance reduction techniques are not able to overcome unbounded variance, while NPG can still achieve global convergence almost surely with a judicious choice of baseline even though its variance remains *unbounded* (see Update 7 for details). This is another example that shows that bounded variance is not necessary for convergence, and some other factors rather than variance account for the convergence behaviour of stochastic policy optimization algorithms.

This leave us an important question to be answered to bridge the gap between theory and practice,

*What are the key factors determining the convergence of stochastic policy optimization?*

As an answer to this question we propose a new notion, the *committal rate* of policy optimization methods and will demonstrate that small committal rates are necessary to ensure the convergent behavior of policy optimization methods.

## 3 Committal Rate of Stochastic Policy Optimization Algorithms

Although the baseline study [12] only focuses on two- and three-action bandits primarily, it develops a useful intuition that stochastic policy optimization in practical settings consists of separate "sampling" and "updating" steps that become coupled in the on-policy setting. Building from this observation, and seeking to explain the outcomes in Section 2, we formalize the following "committal rate" function of a policy optimization algorithm. The main idea is to decouple the "sampling" and "updating" by fixing sampling one action and characterizing the aggressiveness of an update in a deterministic way. Thus, in what follows, by a policy optimization algorithm $\mathcal{A}$ we mean a mapping from all sequences of pairs of action-reward pairs to the set of parameter vectors.

**Definition 2** (Committal Rate). *Fix a reward function $r \in (0, 1]^K$ and an initial parameter vector $\theta_1 \in \mathbb{R}^K$. Consider a policy optimization algorithm $\mathcal{A}$. Let action $a$ be the sampled action **forever** after initialization and let $\theta_t$ be the resulting parameter vector obtained by using $\mathcal{A}$ on the first $t$ observations. The committal rate of algorithm $\mathcal{A}$ on action $a$ (given $r$ and $\theta_1$) is then defined as*

$$\kappa(\mathcal{A}, a) = \sup \left\{ \alpha \geq 0 : \limsup_{t \to \infty} t^\alpha \cdot [1 - \pi_{\theta_t}(a)] < \infty \right\}. \tag{6}$$

Note that in the definition we have suppressed the dependence of $\kappa$ on the rewards and the initial parameter vector. Definition 2 accounts for **how aggressive an update rule is**: An algorithm with

committal rate $\alpha$ will make $\pi_{\theta_t}(a)$ approach 1 at the polynomial rate of $1/t^\alpha$ provided that the sampling rule only chooses action $a$. Thus, a larger value of $\kappa(\mathcal{A}, a)$ indicates an algorithm that quickly commits to the action $a$. For example, if $\pi_{\theta_t}(a) = 1 - 1/(t \cdot \log(t))$, then $\kappa(\mathcal{A}, a) = 1$. Similarly, if $\pi_{\theta_t}(a) = 1 - 1/e^t$, then $\kappa(\mathcal{A}, a) = \infty$, which means $\pi_{\theta_t}(a)$ approaches 1 extremely quickly. On the other hand, if $1 - \pi_{\theta_t}(a) \in \Omega(1)$, then $\kappa(\mathcal{A}, a) = 0$, implying that $\pi_{\theta_t}$ never becomes committal, since $\pi_{\theta_t}(a)$ never approaches 1.

Our next results shows that a small committal rate with respect to sub-optimal actions is necessary for almost sure convergence to a globally optimal policy.

**Theorem 5** (Committal rate main theorem). *Consider a policy optimization method $\mathcal{A}$, together with $r \in (0, 1]^K$ and an initial parameter vector $\theta_1 \in \mathbb{R}^K$. Then,*

$$\max_{a : r(a) < r(a^*), \pi_{\theta_1}(a) > 0} \kappa(\mathcal{A}, a) \leq 1 \tag{7}$$

*is a necessary condition for ensuring the almost sure convergence of the policies obtained using $\mathcal{A}$ and online sampling to the global optimum starting from $\theta_1$.*

In words, Eq. (7) shows that slow reaction to constantly sampling sub-optimal actions is necessary for the success of policy optimization methods when they are used with online sampling.

Using this result, we can now interrogate the committal rates of the previously listed algorithms.

**Theorem 6.** *Let Assumption 1 holds. For the stochastic updates NPG and GNPG from Updates 5 and 6 we obtain $\kappa(NPG, a) = \infty$ and $\kappa(GNPG, a) = \infty$ for all $a \in [K]$ respectively.*

Theorem 6 explains why stochastic NPG and GNPG have a non-zero failure probability in the on-policy stochastic setting: they do not obey a necessary condition for almost sure global convergence. Intuitively, these algorithms can fail by prematurely allocating too much probability to a sub-optimal action: each sampling of an action $a \in [K]$ increments its parameter by $\Theta(1)$, so if $a$ is sampled $t$ times successively, then we have $1 - \pi_{\theta_t}(a) \in O(e^{-c \cdot t})$, which means $\kappa(\mathcal{A}, a) = \infty$. According to Theorem 5, there is a positive probability that a single sub-optimal action can receive a long enough sampling run to ensure the other actions will never again be sampled.

By contrast, we can compare these outcomes to the committal rate of the softmax PG algorithm.

**Theorem 7.** *Let $r(a) > 0$ and $\pi_{\theta_1}(a) > 0$. Softmax PG obtains $\kappa(PG, a) = 1$ for all $a \in [K]$.*

Theorems 5 and 7 provide (partial) explanations of the observations in Section 2: stochastic NPG and GNPG can fail while PG almost surely converges to a global optimum, but their committal rates lie on different sides of the necessary condition. Since $\kappa(PG, a) = 1$ for softmax PG, it follows that $\prod_{t=1}^{\infty} \pi_{\theta_t}(a) = 0$ (see Lemma 18), hence it is not possible to sample sub-optimal actions forever, and the optimal action $a^*$ always has a sufficient chance to be sampled, which ensures learning.

Next, following [12], we consider NPG using an "oracle baseline", which assumes the knowledge of the gap $\Delta$. Chung et al. [12] considered this baseline to point out that convergence in on-policy stochastic gradient methods can happen even if the variance of the gradient estimates "explodes":

**Update 7** (NPG with oracle baseline). $\theta_{t+1} \leftarrow \theta_t + \eta \cdot (\hat{r}_t - \hat{b}_t)$, *where* $\hat{b}_t(a) = \left( \frac{\mathbb{I}\{a_t = a\}}{\pi_{\theta_t}(a)} - 1 \right) \cdot b$ *for all $a \in [K]$, and $b \in (r(a^*) - \Delta, r(a^*))$.*

**Theorem 8.** *Using Update 7, $(\pi^* - \pi_{\theta_t})^\top r \to 0$ as $t \to \infty$ with probability 1.*

As noted, while the variance of the updates provably explodes [12], the necessary condition in Theorem 5 is satisfied. Indeed, if $a_t \neq a^*$, $\pi_{\theta_{t+1}}(a_t) < \pi_{\theta_t}(a_t)$, while the optimal action's probability always increases after any update. Therefore, we have $\kappa(\mathcal{A}, a^*) = \infty$ and $\kappa(\mathcal{A}, a) = 0$ for all $a \neq a^*$. This example shows that the committal rate gives useful information regardless of whether the variance of the update stays bounded.

## 4 The Geometry-Convergence Trade-off in Stochastic Policy Optimization

Theorem 7 raises the question of whether $\kappa(\mathcal{A}, a) \leq 1$ for all sub-optimal actions $a \in [K]$ is sufficient to ensure an algorithm $\mathcal{A}$ converges to an optimal policy almost surely. Unfortunately, this is not the case, and the complete picture of global optimality in stochastic policy optimization is more complex and requires detailed study of different iteration behaviors.

## 4.1 Iteration Behaviours

**Remark 3.** *The condition that $\kappa(\mathcal{A}, a) \leq 1$ for all sub-optimal actions $a \in [K]$ is **not** sufficient for ensuring almost sure convergence to global optimality. In addition to "convergence to a sub-optimal policy with positive probability" and "convergence to a globally optimal policy with probability $1$" there exist other possible optimization behaviours, such as "not converging to any policy".*

In particular, consider the following update behaviors.

**Staying.** For the stationary update $\mathcal{A} : \theta_{t+1} \leftarrow \theta_t$ we obtain $\kappa(\mathcal{A}, a) = 0 \leq 1$ for all $a \in [K]$, yet $\pi_{\theta_t} = \pi_{\theta_1}$ does not converge to the optimal policy nor any sub-optimal deterministic policy.

**Wandering** (NPG with a large baseline). Consider $\mathcal{A} : \theta_{t+1} \leftarrow \theta_t + \eta \cdot \left( \hat{r}_t - \hat{b}_t \right)$ with $\hat{b}_t(a) = \left( \frac{\mathbb{I}\{a_t = a\}}{\pi_{\theta_t}(a)} - 1 \right) \cdot b$ for all $a \in [K]$. If $b > r(a^*)$, then we have $\pi_{\theta_{t+1}}(a_t) < \pi_{\theta_t}(a_t)$, i.e., a selected action's probability will decrease after updating, hence $\kappa(\mathcal{A}, a) = 0$ for all $a \in [K]$. However, $\pi_{\theta_t}(a) \not\to 1$ as $t \to \infty$ for all $a \in [K]$, therefore $\pi_{\theta_t}$ will wander within the simplex forever.

The above examples show that not converging to a sub-optimal policy does not necessarily imply converging to an optimal policy almost surely, and a stronger condition is needed to eliminate unreasonable behaviors like $\theta_{t+1} \leftarrow \theta_t$. We leave it as an open question to identify necessary and sufficient conditions for almost sure convergence to a global optimum.

## 4.2 Geometry-Convergence Trade-off

In Section 2 we see that NPG and GNGP can use true gradients to significantly accelerate PG by better exploiting geometry. However, in the stochastic setting, any estimated geometry might be inaccurate, and intuitively, accelerated methods risk leveraging inaccurate information too aggressively. On the one hand, if progress is sufficiently fast (i.e., with a large committal rate), then an algorithm might never recover from aggressive yet inaccurate updates (Theorem 5). On the other hand, large progress is necessary for fast convergence. The tension between these observations suggest that there might be an inherent trade-off between exploiting geometry and avoiding premature convergence in stochastic policy optimization. We formalize this intuition with the following results. For the first result, we need to restrict to the class of policy optimization methods that do not decrease the probability of the optimal action whenever that action is chosen: In particular, a policy optimization method is said to be *optimality-smart* if for any $t \geq 1$, $\pi_{\tilde{\theta}_t}(a^*) \geq \pi_{\theta_t}(a^*)$ holds where $\tilde{\theta}_t$ is the parameter vector obtained when $a^*$ is chosen in every time step, starting at $\theta_1$, while $\theta_t$ is *any* parameter vector that can be obtained with $t$ updates (regardless of the action sequence chosen), but also starting from $\theta_1$.

**Theorem 9.** *Let $\mathcal{A}$ be optimality-smart and pick a bandit instance. If $\mathcal{A}$ together with on-policy sampling leads to $\{\theta_t\}_{t \geq 1}$ such that $\{\pi_{\theta_t}\}_{t \geq 1}$ converges to a globally optimal policy at a rate $O(1/t^\alpha)$ with positive probability, for $\alpha > 0$, then $\kappa(\mathcal{A}, a^*) \geq \alpha$.*

This theorem implies that a large committal rate for the optimal action is necessary for achieving fast convergence to the globally optimal policy, since the sub-optimality dominates how close the optimal action's probability is to 1, i.e., $(\pi^* - \pi_{\theta_t})^\top r \geq (1 - \pi_{\theta_t}(a^*)) \cdot \Delta$. Therefore $(\pi^* - \pi_{\theta_t})^\top r \in O(1/t^\alpha)$ implies $1 - \pi_{\theta_t}(a^*) \in O(1/t^\alpha)$. Combining this result with Theorem 5 formally establishes the following inherent trade-off between exploiting geometry to accelerate convergence versus achieving global optimality almost surely (aggressiveness vs. stability).

**Theorem 10** (Geometry-Convergence trade-off). *If an algorithm $\mathcal{A}$ is optimality-smart, and $\kappa(\mathcal{A}, a^*) = \kappa(\mathcal{A}, a)$ for at least one $a \neq a^*$, then $\mathcal{A}$ with on-policy sampling can only exhibit at most one of the following two behaviors: (i) $\mathcal{A}$ converges to a globally optimal policy almost surely; (ii) $\mathcal{A}$ converges to a deterministic policy at a rate faster than $O(1/t)$ with positive probability.*

In other words, if $\mathcal{A}$ has a chance to converge to a global optimum, then either $\mathcal{A}$ converges to the globally optimal policy with probability 1 ($\mathcal{A}$ is stable) but at a rate no better than $O(1/t)$, or it achieves a faster than $O(1/t)$ convergence rate ($\mathcal{A}$ is aggressive) but fails to converge to the globally optimal policy with some positive probability. This trade-off between the geometry and convergence is faced by any stochastic policy optimization algorithm that is not informed by external oracle information that allows it to distinguish optimal and sub-optimal actions based on on-policy samples.

**Remark 4.** *Theorem* [10] *implies that an algorithm can achieve at most one of the mentioned two results. It is possible that an algorithm achieves neither (e.g., staying or wandering).*

### 4.3 Exploiting External Information

In Theorem [10], the condition of $\kappa(\mathcal{A}, a^*) = \kappa(\mathcal{A}, a)$ for at least one sub-optimal action $a \in [K]$ is necessary for the trade-off to hold. If this condition can somehow be bypassed, then it is possible to simultaneously achieve faster rates and almost sure convergence to a global optimum. For example, consider Update [7]. As mentioned before, we have $\kappa(\mathcal{A}, a^*) = \infty$ and $\kappa(\mathcal{A}, a) = 0$ for all $a \neq a^*$, breaking the mentioned condition, which allows $\mathcal{A}$ to enjoy almost sure global convergence as well as a $O(e^{-c \cdot t})$ rate. Of course, such a fortuitous outcome required a very specific baseline that is aware of both the optimal reward and the reward gap. Without introducing external mechanisms that inform an on-policy algorithm it appears that such information cannot be recovered sufficiently quickly from sample data alone [21]. Nevertheless, it remains an open question to prove that this is not possible, or whether some other strategy might allow an on-policy stochastic policy optimization algorithm to avoid the condition of Theorem [10] and achieve both fast rates and almost sure global convergence.

| Property I | $\kappa(\mathcal{A}, a) > 1$ | $\kappa(\mathcal{A}, a) \leq 1,$ for all sub-optimal action $a \in [K]$ | | |
|---|---|---|---|---|
| Algorithm | NPG GNPG | Softmax PG SAMBA | Staying Wandering (NPG + large baseline) | NPG + oracle baseline |
| Property II | $\kappa(\mathcal{A}, a^*) = \kappa(\mathcal{A}, a),$ for at least one sub-optimal action $a \in [K]$ | | | $\kappa(\mathcal{A}, a^*) \neq \kappa(\mathcal{A}, a)$ |

Figure 1: Different algorithmic behaviours subdivided by two properties of committal rate. SAMBA [13] does not use parametric policies and is discussed in the appendix.

Figure [1] summarizes all the iteration behaviours we studied in this paper, organized by two properties of committal rate: **(i)** possible failure if $\kappa(\mathcal{A}, a) > 1$ for at least one sub-optimal action $a$; and **(ii)** an inherent geometry-convergence trade-off if $\kappa(\mathcal{A}, a^*) = \kappa(\mathcal{A}, a)$ for at least one sub-optimal action $a$. It remains open to study where other algorithms suit themselves in this diagram.

## 5 Initialization Sensitivity and Ensemble Methods

We use the committal rate to further reveal mystery observed in practice about the initialization sensitivity [22]. With the understanding of this unavoidable phenomenon, we introduce ensemble method and quantitatively characterize the successful rate in terms of number of trials.

### 5.1 Initialization Sensitivity

It has been observed empirically that RL algorithms are sensitive to initialization in practice: the same algorithm can produce remarkably different performance given different random seeds [22]. Some existing work has attempted to explain initialization sensitivity due to the softmax transform [3], but such results only hold for true gradients and apply to standard PG methods.

Using the committal rate theory developed above, we can provide a new explanation and additional understanding of the initialization sensitivity of practical policy optimization algorithms. Most well-performing policy optimization algorithms in practice, such as TRPO and PPO [15, 16], are based on NPG, which exploits geometry to accelerate PG in true gradient settings. However, according to Theorem [10], such fast convergence must incur a positive probability of failing to reach a global optimum, even in bandit settings. Therefore, the need to attempt multiple random seeds to achieve success is an unavoidable consequence of using these algorithms according to this theory.

### 5.2 Ensemble Methods

The committal rate theory also explains why ensemble methods [23–25], *i.e.*, running a policy optimization algorithm in multiple parallel threads and picking the best performing one, can provably work well. This is because a fast algorithm for the true gradient setting can have a positive probability of success or failure across different initializations while always converging quickly. In which case,

multiple independent runs can then be used to reduce the failure probability to any desired positive value, while retaining efficiency (if full parallelism can be maintained).

**Theorem 11.** *With probability $1 - \delta$, the best single run among $O(\log{(1/\delta)})$ independent runs of NPG (GNPG) converges to a globally optimal policy at an $O(e^{-c \cdot t})$ rate.*

It is known that softmax PG can get stuck on long plateaus for even true gradient settings [3, 4], which means almost sure global convergence does not necessarily imply good practical performance. Therefore, it is a reasonable choice to perform well with the compromise of small failure probability. Here we consider simply best selection, and it remains open to understand other practical training tricks, such as proximal update [16, 26] and regularization [27] under stochastic settings.

# 6 Discussions

## 6.1 Sufficient and Necessary Conditions for Almost Sure Global Convergence

We make the following conjecture with some intuitions for the sufficient and necessary condition for global convergence, a question left open in Section 4.

**Conjecture 1.** *Given a stochastic policy optimization algorithm $\mathcal{A}$, if $\kappa(\mathcal{A}, a^*) = \kappa(\mathcal{A}, a)$ for at least one sub-optimal action $a$, then $\kappa(\mathcal{A}, a^*) \in (0, 1]$ is a sufficient and necessary condition for global convergence to $\pi^*$ with **polynomial convergence rate** of $O(1/t^\alpha)$, where $\alpha > 0$.*

The necessary condition is from Theorem 5. For the sufficient condition, Theorem 9 can potentially be strengthened to $\kappa(\mathcal{A}, a^*) \geq \alpha$ is a sufficient and necessary condition for global convergence rate $O(1/t^\alpha)$ ($\alpha > 0$). The observation here is that Assumption 1 leads to $(\pi^* - \pi_{\theta_t})^\top r \leq 1 - \pi_{\theta_t}(a^*)$. This suggests that if $1 - \pi_{\theta_t}(a^*) \in O(1/t^\alpha)$, then $(\pi^* - \pi_{\theta_t})^\top r \in O(1/t^\alpha)$. However, a gap here is $\kappa(\mathcal{A}, a^*) \geq \alpha$ means "$1 - \pi_{\theta_t}(a^*) \in O(1/t^\alpha)$ if we fix sampling $a^*$ forever", and it is not clear if this implies "$1 - \pi_{\theta_t}(a^*) \in O(1/t^\alpha)$ if we run the algorithm $\mathcal{A}$ using on-policy sampling $a_t \sim \pi_{\theta_t}(\cdot)$".

## 6.2 Lower Bounds in Bandit Literature

In the bandit literature [28], the $\Omega(\log T)$ result implies that the convergence speed in terms of sub-optimality ("average regret") cannot be faster than $O(1/t)$. However, the lower bound construction there holds for stochastic reward settings. Theorem 10 holds for a simpler optimization setting: the reward is fixed and deterministic, but the policy gradient is estimated by on-policy sampling. Therefore, the difficulty and trade-off are from the restriction on the action-selection scheme (balancing the aggressiveness and the stability), not from estimating or tracking the reward signal.

## 6.3 General MDPs

The one-state MDP results already show the main findings, since a large portion is about constructing counterexamples showing that the stochastic policy optimization algorithms do not perform well as in the true gradient setting. A counterexample for one-state MDPs is also a counterexample for general MDPs. Therefore, there is no loss of generality by establishing negative results using one-state MDPs. We include extensions to general finite MDPs in Appendix E for completeness.

# 7 Conclusion and Future Work

This paper introduces the committal rate theory, which not only explains why faster policy optimization algorithms in the true gradient setting become dominated by slower counterparts in the on-policy stochastic setting, but also reveals an inherent geometry-convergence trade-off in stochastic policy optimization. The theory also explains empirical observations of sensitivity to random initialization for practical policy optimization algorithms as well as the effectiveness of ensemble methods.

One interesting future direction is to study necessary and sufficient conditions for almost sure global convergence, which could be weaker than the bounded variance assumption. Another important direction is to investigate whether other techniques might be used in on-policy settings to break the condition of Theorem 10 to achieve almost sure global convergence with a fast rate. One also expects that some generalized versions of committal rate would be meaningful in stochastic reward settings.

## Acknowledgments and Disclosure of Funding

The authors would like to thank anonymous reviewers for their valuable comments. Jincheng Mei, Bo Dai and Dale Schuurmans would like to thank Lihong Li for helpful early discussions. Jincheng Mei and Bo Dai would like to thank Nicolas Le Roux for providing feedback on a draft of this manuscript. Jincheng Mei would like to thank Michael Bowling for carefully checking the paper draft in a thesis chapter. Csaba Szepesvári and Dale Schuurmans gratefully acknowledge funding from the Canada CIFAR AI Chairs Program, Amii and NSERC.

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
