# Appendix

The appendix is organized as follows.

# A  Proofs for Algorithm Preferability (Section 2)

## A.1  True Gradient Setting

### A.1.1  Softmax PG

**Lemma 1** (Non-uniform Łojasiewicz (NŁ), [2]) . Let $a^*$ be the uniqe optimal action. Denote $\pi^* = \arg\max_{\pi \in \Delta} \pi^\top r$. Then,

$$\left\| \frac{d\pi_\theta^\top r}{d\theta} \right\|_2 \geq \pi_\theta(a^*) \cdot (\pi^* - \pi_\theta)^\top r. \tag{8}$$

*Proof.* See the proof in [2, Lemma 3]. We include a proof for completeness.

Using the expression of the policy gradient in Update 1, we have,

$$\left\| \frac{d\pi_\theta^\top r}{d\theta} \right\|_2 = \left[ \sum_{a \in \mathcal{A}} \pi_\theta(a)^2 \cdot (r(a) - \pi_\theta^\top r)^2 \right]^{\frac{1}{2}} \tag{9}$$

$$\geq \pi_\theta(a^*) \cdot (r(a^*) - \pi_\theta^\top r). \qquad \square$$

**Proposition 1** (PG upper bound [2]). Using Update 1 with $\eta = 2/5$, we have, for all $t \geq 1$,

$$(\pi^* - \pi_{\theta_t})^\top r \leq 5/(c^2 \cdot t), \tag{10}$$

such that $c = \inf_{t \geq 1} \pi_{\theta_t}(a^*) > 0$ is a constant that depends on $r$ and $\theta_1$, but it does not depend on the time $t$. In particular, if $\pi_{\theta_1}(a) = 1/K$, $\forall a$, then $c \geq 1/K$, i.e.,

$$(\pi^* - \pi_{\theta_t})^\top r \leq 5K^2/t. \tag{11}$$

*Proof.* See the proof in [2, Theorem 2]. We include a proof for completeness.

**First part.** Eq. (10).

According to [2, Lemma 2], for any $r \in [0,1]^K$, $\theta \mapsto \pi_\theta^\top r$ is $5/2$-smooth,

$$\left| (\pi_{\theta'} - \pi_\theta)^\top r - \left\langle \frac{d\pi_\theta^\top r}{d\theta}, \theta' - \theta \right\rangle \right| \leq \frac{5}{4} \cdot \|\theta' - \theta\|_2^2. \tag{12}$$

Denote $\delta(\theta_t) \coloneqq (\pi^* - \pi_{\theta_t})^\top r$. We have, for all $t \geq 1$,

$$\delta(\theta_{t+1}) - \delta(\theta_t) = -\pi_{\theta_{t+1}}^\top r + \pi_{\theta_t}^\top r + \left\langle \frac{d\pi_{\theta_t}^\top r}{d\theta_t}, \theta_{t+1} - \theta_t \right\rangle - \left\langle \frac{d\pi_{\theta_t}^\top r}{d\theta_t}, \theta_{t+1} - \theta_t \right\rangle \tag{13}$$

$$\leq \frac{5}{4} \cdot \|\theta_{t+1} - \theta_t\|_2^2 - \left\langle \frac{d\pi_{\theta_t}^\top r}{d\theta_t}, \theta_{t+1} - \theta_t \right\rangle \qquad \text{(by Eq. (12))} \tag{14}$$

$$= -\frac{1}{5} \cdot \left\| \frac{d\pi_{\theta_t}^\top r}{d\theta_t} \right\|_2^2 \qquad \text{(using Update 1 and } \eta = 2/5) \tag{15}$$

$$\leq -\frac{1}{5} \cdot \left[ \pi_{\theta_t}(a^*) \cdot (\pi^* - \pi_{\theta_t})^\top r \right]^2 \qquad \text{(by Lemma 1)} \tag{16}$$

$$\leq -\frac{c^2}{5} \cdot \delta(\theta_t)^2, \tag{17}$$

where $c = \inf_{t \geq 1} \pi_{\theta_t}(a^*) > 0$ is from [2, Lemma 5]. Then we have, for all $t \geq 1$,

$$\frac{1}{\delta(\theta_t)} = \frac{1}{\delta(\theta_1)} + \sum_{s=1}^{t-1} \left[ \frac{1}{\delta(\theta_{s+1})} - \frac{1}{\delta(\theta_s)} \right] \tag{18}$$

$$= \frac{1}{\delta(\theta_1)} + \sum_{s=1}^{t-1} \frac{1}{\delta(\theta_{s+1}) \cdot \delta(\theta_s)} \cdot (\delta(\theta_s) - \delta(\theta_{s+1})) \tag{19}$$

$$\geq \frac{1}{\delta(\theta_1)} + \sum_{s=1}^{t-1} \frac{1}{\delta(\theta_{s+1}) \cdot \delta(\theta_s)} \cdot \frac{c^2}{5} \cdot \delta(\theta_s)^2 \qquad \text{(by Eq. (13))} \tag{20}$$

$$\geq \frac{1}{\delta(\theta_1)} + \frac{c^2}{5} \cdot (t-1) \qquad (0 < \delta(\theta_{t+1}) \leq \delta(\theta_t), \text{ by Eq. (13)}) \tag{21}$$

$$\geq \frac{c^2}{5} \cdot t, \qquad \left( \delta(\theta_1) \leq 1 < 5/c^2 \right) \tag{22}$$

which implies Eq. (10).

**Second part.** Eq. (11).

Suppose $\pi_{\theta_1}(a) = 1/K$, $\forall a$. Using similar arguments in [2, Proposition 2], we prove that if $\pi_{\theta_t}(a^*) \geq \pi_{\theta_t}(a)$, for all $a \neq a^*$, then $\pi_{\theta_{t+1}}(a^*) \geq \pi_{\theta_t}(a^*)$. We have, $\forall a \neq a^*$,

$$\frac{d\pi_{\theta_t}^\top r}{d\theta_t(a^*)} = \pi_{\theta_t}(a^*) \cdot \left( r(a^*) - \pi_{\theta_t}^\top r \right) \tag{23}$$

$$\geq \pi_{\theta_t}(a^*) \cdot \left( r(a) - \pi_{\theta_t}^\top r \right) \qquad \left( r(a^*) - \pi_\theta^\top r > 0 \text{ and } r(a^*) > r(a) \right) \tag{24}$$

$$> \pi_{\theta_t}(a) \cdot \left( r(a) - \pi_{\theta_t}^\top r \right) \qquad (\pi_{\theta_t}(a^*) \geq \pi_{\theta_t}(a), \text{ by assumption}) \tag{25}$$

$$= \frac{d\pi_{\theta_t}^\top r}{d\theta_t(a)}. \tag{26}$$

After one step policy gradient update, we have,

$$\pi_{\theta_{t+1}}(a^*) = \frac{\exp\{\theta_{t+1}(a^*)\}}{\sum_a \exp\{\theta_{t+1}(a)\}} \tag{27}$$

$$= \frac{\exp\left\{\theta_t(a^*) + \eta \cdot \frac{d\pi_{\theta_t}^\top r}{d\theta_t(a^*)}\right\}}{\sum_a \exp\left\{\theta_t(a) + \eta \cdot \frac{d\pi_{\theta_t}^\top r}{d\theta_t(a)}\right\}} \tag{28}$$

$$\geq \frac{\exp\left\{\theta_t(a^*) + \eta \cdot \frac{d\pi_{\theta_t}^\top r}{d\theta_t(a^*)}\right\}}{\sum_a \exp\left\{\theta_t(a) + \eta \cdot \frac{d\pi_{\theta_t}^\top r}{d\theta_t(a^*)}\right\}} \qquad \text{(by Eq. (23))} \tag{29}$$

$$= \frac{\exp\{\theta_t(a^*)\}}{\sum_a \exp\{\theta_t(a)\}} = \pi_{\theta_t}(a^*). \tag{30}$$

Note that $\pi_{\theta_1}(a^*) \geq \pi_{\theta_1}(a)$, and thus we have,

$$c = \inf_{t \geq 1} \pi_{\theta_t}(a^*) \geq \pi_{\theta_1}(a^*) = 1/K. \qquad \square$$

**Proposition 2** (PG lower bound [2]). For sufficiently large $t \geq 1$, Update 1 with $\eta \in (0, 1]$ exhibits

$$(\pi^* - \pi_{\theta_t})^\top r \geq \Delta^2 / (6 \cdot t), \tag{31}$$

where $\Delta = r(a^*) - \max_{a \neq a^*} r(a) > 0$ is the reward gap of $r$.

*Proof.* See the proof in [2, Theorem 9]. We include a proof for completeness.

According to [2, Lemma 17],

$$\left\| \frac{d\pi_\theta^\top r}{d\theta} \right\|_2 \leq \frac{\sqrt{2}}{\Delta} \cdot (\pi^* - \pi_\theta)^\top r. \tag{32}$$

Denote $\delta(\theta_t) := (\pi^* - \pi_{\theta_t})^\top r$. We have, for all $t \geq 1$,

$$\delta(\theta_t) - \delta(\theta_{t+1}) = (\pi_{\theta_{t+1}} - \pi_{\theta_t})^\top r - \left\langle \frac{d\pi_{\theta_t}^\top r}{d\theta_t}, \theta_{t+1} - \theta_t \right\rangle + \left\langle \frac{d\pi_{\theta_t}^\top r}{d\theta_t}, \theta_{t+1} - \theta_t \right\rangle \tag{33}$$

$$\leq \frac{5}{4} \cdot \|\theta_{t+1} - \theta_t\|_2^2 + \left\langle \frac{d\pi_{\theta_t}^\top r}{d\theta_t}, \theta_{t+1} - \theta_t \right\rangle \qquad \text{(by Eq. (12))} \tag{34}$$

$$\leq \left( \frac{5}{4} + 1 \right) \cdot \left\| \frac{d\pi_{\theta_t}^\top r}{d\theta_t} \right\|_2^2 \qquad \text{(using Update 1 and } \eta \in (0,1]) \tag{35}$$

$$\leq \frac{9}{2} \cdot \frac{1}{\Delta^2} \cdot \delta(\theta_t)^2. \qquad \text{(by Eq. (32))} \tag{36}$$

According to Proposition 1, we have $\delta(\theta_t) \to 0$ as $t \to \infty$. We show that for all large enough $t \geq 1$, $\delta(\theta_t) \leq \frac{10}{9} \cdot \delta(\theta_{t+1})$ by contradiction. Suppose $\delta(\theta_t) > \frac{10}{9} \cdot \delta(\theta_{t+1})$. We have,

$$\delta(\theta_{t+1}) \geq \delta(\theta_t) - \frac{9}{2} \cdot \frac{1}{\Delta^2} \cdot \delta(\theta_t) \qquad \text{(by Eq. (33))} \tag{37}$$

$$> \frac{10}{9} \cdot \delta(\theta_{t+1}) - \frac{9}{2} \cdot \frac{1}{\Delta^2} \cdot \left( \frac{10}{9} \cdot \delta(\theta_{t+1}) \right)^2 \tag{38}$$

$$= \frac{10}{9} \cdot \delta(\theta_{t+1}) - \frac{50}{9} \cdot \frac{1}{\Delta^2} \cdot \delta(\theta_{t+1})^2, \tag{39}$$

where the second inequality is because of the function $f : x \mapsto x - a \cdot x^2$ with $a > 0$ is monotonically increasing for all $0 < x \leq \frac{1}{2a}$. Eq. (37) implies that,

$$\delta(\theta_{t+1}) > \frac{\Delta^2}{50}, \tag{40}$$

for large enough $t \geq 1$, which is a contradiction with $\delta(\theta_t) \to 0$ as $t \to \infty$. Thus we have $\frac{\delta(\theta_{t+1})}{\delta(\theta_t)} \geq \frac{9}{10}$ holds for all large enough $t \geq 1$. Next, we have,

$$\frac{1}{\delta(\theta_{t+1})} - \frac{1}{\delta(\theta_t)} = \frac{1}{\delta(\theta_{t+1}) \cdot \delta(\theta_t)} \cdot (\delta(\theta_t) - \delta(\theta_{t+1})) \tag{41}$$

$$\leq \frac{1}{\delta(\theta_{t+1}) \cdot \delta(\theta_t)} \cdot \frac{9}{2} \cdot \frac{1}{\Delta^2} \cdot \delta(\theta_t)^2 \qquad \text{(by Eq. (33))} \tag{42}$$

$$\leq \frac{5}{\Delta^2}. \qquad \left( \frac{\delta(\theta_t)}{\delta(\theta_{t+1})} \leq \frac{10}{9} \right) \tag{43}$$

Summing up from $T_1$ (some large enough time) to $T_1 + t$, we have

$$\frac{1}{\delta(\theta_{T_1+t})} - \frac{1}{\delta(\theta_{T_1})} \leq \frac{5}{\Delta^2} \cdot (t-1) \leq \frac{5}{\Delta^2} \cdot t. \tag{44}$$

Since $T_1$ is a finite time, $\delta(\theta_{T_1}) \geq 1/C$ for some constant $C > 0$. Rearranging, we have

$$(\pi^* - \pi_{\theta_{T_1+t}})^\top r = \delta(\theta_{T_1+t}) \geq \frac{1}{\frac{1}{\delta_{T_1}} + \frac{5}{\Delta^2} \cdot t} \geq \frac{1}{C + \frac{5}{\Delta^2} \cdot t} \geq \frac{1}{C + \frac{5}{\Delta^2} \cdot (T_1+t)}. \tag{45}$$

By abusing notation $t := T_1 + t$ and $C \leq \frac{t}{\Delta^2}$, we have

$$(\pi^* - \pi_{\theta_t})^\top r \geq \frac{1}{C + \frac{5}{\Delta^2} \cdot t} \geq \frac{1}{\frac{t}{\Delta^2} + \frac{5}{\Delta^2} \cdot t} = \frac{\Delta^2}{6 \cdot t}, \tag{46}$$

for all large enough $t \geq 1$. $\qquad \square$

### A.1.2 NPG

**Lemma 2** (Natural Non-uniform Łojasiewicz (NŁ) inequality, continuous)**.** Let $r \in (0,1)^K$. Denote $\Delta(a) := r(a^*) - r(a)$, and $\Delta := r(a^*) - \max_{a \neq a^*} r(a)$ as the reward gap of $r$. We have, for any policy $\pi_\theta := \text{softmax}(\theta)$,

$$\left\langle \frac{d\pi_\theta^\top r}{d\theta}, r \right\rangle \geq \pi_\theta(a^*) \cdot \Delta \cdot (\pi^* - \pi_\theta)^\top r. \tag{47}$$

*Proof.* Without loss of generality, let $r(1) > r(2) > \cdots > r(K)$. We have,

$$\left\langle \frac{d\pi_\theta^\top r}{d\theta}, r \right\rangle = r^\top \left( \mathrm{diag}(\pi_\theta) - \pi_\theta \pi_\theta^\top \right) r \tag{48}$$

$$= \sum_{i=1}^{K} \pi_\theta(i) \cdot r(i)^2 - \left[ \sum_{i=1}^{K} \pi_\theta(i) \cdot r(i) \right]^2 \tag{49}$$

$$= \sum_{i=1}^{K} \pi_\theta(i) \cdot r(i)^2 - \sum_{i=1}^{K} \pi_\theta(i)^2 \cdot r(i)^2 - 2 \cdot \sum_{i=1}^{K-1} \pi_\theta(i) \cdot r(i) \cdot \sum_{j=i+1}^{K} \pi_\theta(j) \cdot r(j) \tag{50}$$

$$= \sum_{i=1}^{K} \pi_\theta(i) \cdot r(i)^2 \cdot [1 - \pi_\theta(i)] - 2 \cdot \sum_{i=1}^{K-1} \pi_\theta(i) \cdot r(i) \cdot \sum_{j=i+1}^{K} \pi_\theta(j) \cdot r(j) \tag{51}$$

$$= \sum_{i=1}^{K} \pi_\theta(i) \cdot r(i)^2 \cdot \sum_{j \neq i} \pi_\theta(j) - 2 \cdot \sum_{i=1}^{K-1} \pi_\theta(i) \cdot r(i) \cdot \sum_{j=i+1}^{K} \pi_\theta(j) \cdot r(j) \tag{52}$$

$$= \sum_{i=1}^{K-1} \pi_\theta(i) \cdot \sum_{j=i+1}^{K} \pi_\theta(j) \cdot \left[ r(i)^2 + r(j)^2 \right] - 2 \cdot \sum_{i=1}^{K-1} \pi_\theta(i) \cdot r(i) \cdot \sum_{j=i+1}^{K} \pi_\theta(j) \cdot r(j) \tag{53}$$

$$= \sum_{i=1}^{K-1} \pi_\theta(i) \cdot \sum_{j=i+1}^{K} \pi_\theta(j) \cdot [r(i) - r(j)]^2, \tag{54}$$

which can be lower bounded as,

$$\left\langle \frac{d\pi_\theta^\top r}{d\theta}, r \right\rangle \geq \pi_\theta(1) \cdot \sum_{j=2}^{K} \pi_\theta(j) \cdot [r(1) - r(j)]^2 \qquad \text{(fewer terms)} \tag{55}$$

$$= \pi_\theta(a^*) \cdot \sum_{a \neq a^*} \pi_\theta(a) \cdot \Delta(a)^2 \qquad (a^* = 1) \tag{56}$$

$$\geq \pi_\theta(a^*) \cdot \Delta \cdot \sum_{a \neq a^*} \pi_\theta(a) \cdot \Delta(a) \qquad (\Delta(a) \geq \Delta) \tag{57}$$

$$= \pi_\theta(a^*) \cdot \Delta \cdot (\pi^* - \pi_\theta)^\top r. \qquad \square \tag{58}$$

**Remark 5.** *The natural NŁ inequality of Lemma 2 is tight. Consider $K = 2$, we have,*

$$r^\top \left( diag(\pi_\theta) - \pi_\theta \pi_\theta^\top \right) r = \pi_\theta(1) \cdot r(1)^2 + \pi_\theta(2) \cdot r(2)^2 - [\pi_\theta(1) \cdot r(1) + \pi_\theta(2) \cdot r(2)]^2 \tag{58}$$

$$= \pi_\theta(1) \cdot r(1)^2 \cdot [1 - \pi_\theta(1)] + \pi_\theta(2) \cdot r(2)^2 \cdot [1 - \pi_\theta(2)] - 2 \cdot \pi_\theta(1) \cdot r(1) \cdot \pi_\theta(2) \cdot r(2) \tag{59}$$

$$= \pi_\theta(1) \cdot r(1)^2 \cdot \pi_\theta(2) + \pi_\theta(2) \cdot r(2)^2 \cdot \pi_\theta(1) - 2 \cdot \pi_\theta(1) \cdot r(1) \cdot \pi_\theta(2) \cdot r(2) \qquad (\pi_\theta(1) + \pi_\theta(2) = 1) \tag{60}$$

$$= \pi_\theta(1) \cdot \pi_\theta(2) \cdot [r(1) - r(2)]^2 \tag{61}$$

$$= \pi_\theta(a^*) \cdot \Delta \cdot (\pi^* - \pi_\theta)^\top r, \qquad \left( a^* = 1, \ \Delta = r(1) - r(2), \ (\pi^* - \pi_\theta)^\top r = \pi_\theta(2) \cdot [r(1) - r(2)] \right) \tag{62}$$

*which means the equality holds for the above problem.*

**Remark 6.** *For the continuous natural PG flow: $\frac{d\theta_t}{dt} = \eta \cdot r$, and $\pi_{\theta_t} = \mathrm{softmax}(\theta_t)$, Lemma 2 can be used to characterize the progress at each time step. We have, for all $t \geq 1$,*

$$\frac{d (\pi^* - \pi_{\theta_t})^\top r}{dt} = -\frac{d\pi_{\theta_t}^\top r}{dt} \tag{63}$$

$$= -\left( \frac{d\theta_t}{dt} \right)^\top \left( \frac{d\pi_{\theta_t}^\top r}{d\theta_t} \right) \tag{64}$$

$$= -\eta \cdot r^\top \left( diag(\pi_{\theta_t}) - \pi_{\theta_t} \pi_{\theta_t}^\top \right) r \qquad \text{(NPG flow)} \tag{65}$$

$$\leq -\eta \cdot \pi_{\theta_t}(a^*) \cdot \Delta \cdot (\pi^* - \pi_{\theta_t})^\top r, \qquad \text{(by Lemma 2)} \tag{66}$$

*which means the progress at time $t$ is proportional to the sub-optimality gap $(\pi^* - \pi_{\theta_t})^\top r$, leading to a linear convergence rate.*

**Lemma 3** (Natural NŁ inequality, discrete). Given any policy $\pi$, define $\pi'$ as

$$\pi'(a) := \frac{\pi(a) \cdot e^{\eta \cdot r(a)}}{\sum_{a'} \pi(a') \cdot e^{\eta \cdot r(a')}}, \quad \text{for all } a \in [K], \tag{67}$$

where $\eta > 0$ is the learning rate. We have,

$$(\pi' - \pi)^\top r \geq \left[ 1 - \frac{1}{\pi(a^*) \cdot (e^{\eta \cdot \Delta} - 1) + 1} \right] \cdot (\pi^* - \pi)^\top r. \tag{68}$$

*Proof.* Without loss of generality, let $r(1) > r(2) > \cdots > r(K)$. We have,

$$(\pi' - \pi)^\top r = \sum_{i=1}^{K} [\pi'(i) \cdot r(i) - \pi(i) \cdot r(i)] \tag{69}$$

$$= \sum_{i=1}^{K} \left[ \frac{\pi(i) \cdot e^{\eta \cdot r(i)} \cdot r(i)}{\sum_{j=1}^{K} \pi(j) \cdot e^{\eta \cdot r(j)}} - \pi(i) \cdot r(i) \right] \quad \text{(by definition of } \pi') \tag{70}$$

$$= \frac{1}{\sum_{j=1}^{K} \pi(j) \cdot e^{\eta \cdot r(j)}} \cdot \left[ \sum_{i=1}^{K} \pi(i) \cdot e^{\eta \cdot r(i)} \cdot r(i) - \sum_{i=1}^{K} \pi(i) \cdot r(i) \cdot \sum_{j=1}^{K} \pi(j) \cdot e^{\eta \cdot r(j)} \right]. \tag{71}$$

Next, we have,

$$\sum_{i=1}^{K} \pi(i) \cdot e^{\eta \cdot r(i)} \cdot r(i) - \sum_{i=1}^{K} \pi(i) \cdot r(i) \cdot \sum_{j=1}^{K} \pi(j) \cdot e^{\eta \cdot r(j)} \tag{72}$$

$$= \sum_{i=1}^{K} \pi(i) \cdot e^{\eta \cdot r(i)} \cdot r(i) - \sum_{i=1}^{K} \pi(i)^2 \cdot e^{\eta \cdot r(i)} \cdot r(i) - \sum_{i=1}^{K} \pi(i) \cdot r(i) \cdot \sum_{j \neq i} \pi(j) \cdot e^{\eta \cdot r(j)} \tag{73}$$

$$= \sum_{i=1}^{K} \pi(i) \cdot e^{\eta \cdot r(i)} \cdot r(i) \cdot [1 - \pi(i)] - \sum_{i=1}^{K} \pi(i) \cdot r(i) \cdot \sum_{j \neq i} \pi(j) \cdot e^{\eta \cdot r(j)} \tag{74}$$

$$= \sum_{i=1}^{K} \pi(i) \cdot e^{\eta \cdot r(i)} \cdot r(i) \cdot \sum_{j \neq i} \pi(j) - \sum_{i=1}^{K} \pi(i) \cdot r(i) \cdot \sum_{j \neq i} \pi(j) \cdot e^{\eta \cdot r(j)} \tag{75}$$

$$= \sum_{i=1}^{K-1} \pi(i) \cdot \sum_{j=i+1}^{K} \pi(j) \cdot \left[ e^{\eta \cdot r(i)} \cdot r(i) + e^{\eta \cdot r(j)} \cdot r(j) \right] - \sum_{i=1}^{K-1} \pi(i) \cdot \sum_{j=i+1}^{K} \pi(j) \cdot \left[ e^{\eta \cdot r(j)} \cdot r(i) + e^{\eta \cdot r(i)} \cdot r(j) \right] \tag{76}$$

$$= \sum_{i=1}^{K-1} \pi(i) \cdot \sum_{j=i+1}^{K} \pi(j) \cdot \left[ e^{\eta \cdot r(i)} - e^{\eta \cdot r(j)} \right] \cdot [r(i) - r(j)], \tag{77}$$

which can be lower bounded as,

$$\sum_{i=1}^{K} \pi(i) \cdot e^{\eta \cdot r(i)} \cdot r(i) - \sum_{i=1}^{K} \pi(i) \cdot r(i) \cdot \sum_{j=1}^{K} \pi(j) \cdot e^{\eta \cdot r(j)} \tag{78}$$

$$\geq \pi(1) \cdot \sum_{j=2}^{K} \pi(j) \cdot \left[ e^{\eta \cdot r(1)} - e^{\eta \cdot r(j)} \right] \cdot [r(1) - r(j)] \qquad \text{(fewer terms)} \tag{79}$$

$$\geq \pi(1) \cdot \sum_{j=2}^{K} \pi(j) \cdot \left[ e^{\eta \cdot r(1)} - e^{\eta \cdot r(2)} \right] \cdot [r(1) - r(j)] \qquad (r(j) \leq r(2), \text{ for all } j \geq 2) \tag{80}$$

$$= \pi(1) \cdot e^{\eta \cdot r(2)} \cdot \left( e^{\eta \cdot \Delta} - 1 \right) \cdot \sum_{a \neq a^*} \pi(a) \cdot \Delta(a) \qquad (\Delta = r(1) - r(2)) \tag{81}$$

$$= \pi(1) \cdot e^{\eta \cdot r(2)} \cdot \left( e^{\eta \cdot \Delta} - 1 \right) \cdot (\pi^* - \pi)^\top r. \tag{82}$$

Combining Eqs. (69) and (72), we have,

$$(\pi' - \pi)^\top r \geq \frac{\pi(1) \cdot e^{\eta \cdot r(2)} \cdot \left( e^{\eta \cdot \Delta} - 1 \right)}{\pi(1) \cdot e^{\eta \cdot r(1)} + \sum_{j=2}^{K} \pi(j) \cdot e^{\eta \cdot r(j)}} \cdot (\pi^* - \pi)^\top r \tag{83}$$

$$= \frac{\pi(1) \cdot \left( e^{\eta \cdot \Delta} - 1 \right)}{\pi(1) \cdot e^{\eta \cdot \Delta} + \sum_{j=2}^{K} \pi(j) \cdot e^{\eta \cdot [r(j) - r(2)]}} \cdot (\pi^* - \pi)^\top r \tag{84}$$

$$\geq \frac{\pi(1) \cdot \left( e^{\eta \cdot \Delta} - 1 \right)}{\pi(1) \cdot e^{\eta \cdot \Delta} + \sum_{j=2}^{K} \pi(j)} \cdot (\pi^* - \pi)^\top r \qquad (r(j) - r(2) \leq 0, \text{ for all } j \geq 2) \tag{85}$$

$$= \frac{\pi(1) \cdot \left( e^{\eta \cdot \Delta} - 1 \right)}{\pi(1) \cdot e^{\eta \cdot \Delta} + 1 - \pi(1)} \cdot (\pi^* - \pi)^\top r \tag{86}$$

$$= \left[ 1 - \frac{1}{\pi(a^*) \cdot (e^{\eta \cdot \Delta} - 1) + 1} \right] \cdot (\pi^* - \pi)^\top r. \qquad (a^* = 1) \qquad \square$$

**Remark 7.** *The natural NŁ inequality of Lemma 3 is tight. Consider $K = 2$, we have,*

$$(\pi' - \pi)^\top r = \frac{\pi(1) \cdot e^{\eta \cdot r(1)} \cdot r(1) + \pi(2) \cdot e^{\eta \cdot r(2)} \cdot r(2)}{\pi(1) \cdot e^{\eta \cdot r(1)} + \pi(2) \cdot e^{\eta \cdot r(2)}} - [\pi(1) \cdot r(1) + \pi(2) \cdot r(2)] \tag{87}$$

$$= \frac{\pi(1) \cdot \pi(2) \cdot [r(1) - r(2)] \cdot \left[ e^{\eta \cdot r(1)} - e^{\eta \cdot r(2)} \right]}{\pi(1) \cdot e^{\eta \cdot r(1)} + \pi(2) \cdot e^{\eta \cdot r(2)}} \tag{88}$$

$$= \frac{\pi(1) \cdot \left( e^{\eta \cdot [r(1) - r(2)]} - 1 \right)}{\pi(1) \cdot e^{\eta \cdot [r(1) - r(2)]} + \pi(2)} \cdot \pi(2) \cdot [r(1) - r(2)] \tag{89}$$

$$= \frac{\pi(a^*) \cdot \left( e^{\eta \cdot \Delta} - 1 \right)}{\pi(a^*) \cdot e^{\eta \cdot \Delta} + 1 - \pi(a^*)} \cdot (\pi^* - \pi)^\top r, \qquad (a^* = 1, \ \Delta = r(1) - r(2)) \tag{90}$$

*which means the equality holds for the above problem.*

**Theorem 1** (NPG upper bound). *Using Update 2 with any $\eta > 0$, i.e., $\forall t \geq 1$,*

$$\theta_{t+1} \leftarrow \theta_t + \eta \cdot r, \ \text{ and } \pi_{\theta_{t+1}} = \text{softmax}(\theta_{t+1}), \tag{91}$$

*where $\eta > 0$ is the learning rate. We have, for all $t \geq 1$,*

$$(\pi^* - \pi_{\theta_t})^\top r \leq (\pi^* - \pi_{\theta_1})^\top r \cdot e^{-c \cdot (t-1)}, \tag{92}$$

*where $c := \log \left( \pi_{\theta_1}(a^*) \cdot \left( e^{\eta \cdot \Delta} - 1 \right) + 1 \right) > 0$ for any $\eta > 0$, and $\Delta = r(a^*) - \max_{a \neq a^*} r(a) > 0$.*

*Proof.* We have, for all $t \geq 1$,

$$\left(\pi^* - \pi_{\theta_{t+1}}\right)^\top r = \left(\pi^* - \pi_{\theta_t}\right)^\top r - \left(\pi_{\theta_{t+1}} - \pi_{\theta_t}\right)^\top r \tag{93}$$

$$\leq \frac{1}{\pi_{\theta_t}(a^*) \cdot (e^{\eta \cdot \Delta} - 1) + 1} \cdot \left(\pi^* - \pi_{\theta_t}\right)^\top r \qquad \text{(by Lemma 3)} \tag{94}$$

$$\leq \frac{1}{\pi_{\theta_1}(a^*) \cdot (e^{\eta \cdot \Delta} - 1) + 1} \cdot \left(\pi^* - \pi_{\theta_t}\right)^\top r \qquad \text{(see below)} \tag{95}$$

$$\leq \frac{1}{\left[\pi_{\theta_1}(a^*) \cdot (e^{\eta \cdot \Delta} - 1) + 1\right]^t} \cdot \left(\pi^* - \pi_{\theta_1}\right)^\top r \tag{96}$$

$$= \frac{\left(\pi^* - \pi_{\theta_1}\right)^\top r}{e^{c \cdot t}}, \tag{97}$$

where the second inequality is because of for all $t \geq 1$,

$$\pi_{\theta_{t+1}}(a^*) = \frac{\pi_{\theta_t}(a^*) \cdot e^{\eta \cdot r(a^*)}}{\sum_a \pi_{\theta_t}(a) \cdot e^{\eta \cdot r(a)}} \tag{98}$$

$$= \frac{\pi_{\theta_t}(a^*)}{\sum_a \pi_{\theta_t}(a) \cdot e^{-\eta \cdot \Delta(a)}} \tag{99}$$

$$\geq \pi_{\theta_t}(a^*). \qquad (\Delta(a) \geq 0) \qquad \square$$

### A.1.3  GNPG

**Lemma 4** (Non-uniform Smoothness (NS), [9])**.** The spectral radius (largest absolute eigenvalue) of Hessian matrix $\frac{d^2 \pi_\theta^\top r}{d\theta^2}$ is upper bounded by $3 \cdot \left\| \frac{d\pi_\theta^\top r}{d\theta} \right\|_2$.

*Proof.* See the proof in [9, Lemma 2]. We include a proof for completeness.

Let $S \coloneqq S(r, \theta) \in \mathbb{R}^{K \times K}$ be the second derivative of the value map $\theta \mapsto \pi_\theta^\top r$. Denote $H(\pi_\theta) \coloneqq \mathrm{diag}(\pi_\theta) - \pi_\theta \pi_\theta^\top$ as the Jacobian of $\theta \mapsto \mathrm{softmax}(\theta)$. Now, by its definition we have

$$S = \frac{d}{d\theta} \left\{ \frac{d\pi_\theta^\top r}{d\theta} \right\} \tag{100}$$

$$= \frac{d}{d\theta} \left\{ H(\pi_\theta) r \right\} \tag{101}$$

$$= \frac{d}{d\theta} \left\{ (\mathrm{diag}(\pi_\theta) - \pi_\theta \pi_\theta^\top) r \right\}. \tag{102}$$

Continuing with our calculation fix $i, j \in [K]$. Then,

$$S_{(i,j)} = \frac{d\{\pi_\theta(i) \cdot (r(i) - \pi_\theta^\top r)\}}{d\theta(j)} \tag{103}$$

$$= \frac{d\pi_\theta(i)}{d\theta(j)} \cdot (r(i) - \pi_\theta^\top r) + \pi_\theta(i) \cdot \frac{d\{r(i) - \pi_\theta^\top r\}}{d\theta(j)} \tag{104}$$

$$= (\delta_{ij} \pi_\theta(j) - \pi_\theta(i) \pi_\theta(j)) \cdot (r(i) - \pi_\theta^\top r) - \pi_\theta(i) \cdot (\pi_\theta(j) r(j) - \pi_\theta(j) \pi_\theta^\top r) \tag{105}$$

$$= \delta_{ij} \pi_\theta(j) \cdot (r(i) - \pi_\theta^\top r) - \pi_\theta(i) \pi_\theta(j) \cdot (r(i) - \pi_\theta^\top r) - \pi_\theta(i) \pi_\theta(j) \cdot (r(j) - \pi_\theta^\top r), \tag{106}$$

where

$$\delta_{ij} = \begin{cases} 1, & \text{if } i = j, \\ 0, & \text{otherwise} \end{cases} \tag{107}$$

is Kronecker's $\delta$-function. To show the bound on the spectral radius of $S$, pick $y \in \mathbb{R}^K$. Then,

$$\left| y^\top S y \right| = \left| \sum_{i=1}^{K} \sum_{j=1}^{K} S_{(i,j)} \cdot y(i) \cdot y(j) \right| \tag{108}$$

$$= \left| \sum_i \pi_\theta(i)(r(i) - \pi_\theta^\top r)y(i)^2 - 2 \sum_i \pi_\theta(i)(r(i) - \pi_\theta^\top r)y(i) \sum_j \pi_\theta(j)y(j) \right| \tag{109}$$

$$= \left| (H(\pi_\theta)r)^\top (y \odot y) - 2 \cdot (H(\pi_\theta)r)^\top y \cdot \left( \pi_\theta^\top y \right) \right| \tag{110}$$

$$\leq \|H(\pi_\theta)r\|_\infty \cdot \|y \odot y\|_1 + 2 \cdot \|H(\pi_\theta)r\|_2 \cdot \|y\|_2 \cdot \|\pi_\theta\|_1 \cdot \|y\|_\infty \tag{111}$$

$$\leq 3 \cdot \|H(\pi_\theta)r\|_2 \cdot \|y\|_2^2 \tag{112}$$

$$= 3 \cdot \left\| \frac{d\pi_\theta^\top r}{d\theta} \right\|_2 \cdot \|y\|_2^2, \tag{113}$$

where $\odot$ is Hadamard (component-wise) product, and the third last inequality uses Hölder's inequality together with the triangle inequality. $\square$

**Proposition 3** (GNPG upper bound [9]). Using Update 3 with $\eta = 1/6$, we have, for all $t \geq 1$,

$$(\pi^* - \pi_{\theta_t})^\top r \leq (\pi^* - \pi_{\theta_1})^\top r \cdot e^{-\frac{c \cdot (t-1)}{12}}, \tag{114}$$

where $c = \inf_{t \geq 1} \pi_{\theta_t}(a^*) > 0$ does not depend on $t$. If $\pi_{\theta_1}(a) = 1/K$, $\forall a$, then $c \geq 1/K$.

*Proof.* See the proof in [9, Theorem 2]. We include a proof for completeness.

Denote $\theta_{\zeta_t} := \theta_t + \zeta_t \cdot (\theta_{t+1} - \theta_t)$ with some $\zeta_t \in [0, 1]$. According to Taylor's theorem,

$$\left| (\pi_{\theta_{t+1}} - \pi_{\theta_t})^\top r - \left\langle \frac{d\pi_{\theta_t}^\top r}{d\theta_t}, \theta_{t+1} - \theta_t \right\rangle \right| = \frac{1}{2} \cdot \left| (\theta_{t+1} - \theta_t)^\top S(r, \theta_{\zeta_t}) (\theta_{t+1} - \theta_t) \right| \tag{115}$$

$$\leq \frac{3}{2} \cdot \left\| \frac{d\pi_{\theta_{\zeta_t}}^\top r}{d\theta_{\zeta_t}} \right\|_2 \cdot \|\theta_{t+1} - \theta_t\|_2^2 \qquad \text{(by Lemma 4)} \tag{116}$$

$$\leq 3 \cdot \left\| \frac{d\pi_{\theta_t}^\top r}{d\theta_t} \right\|_2 \cdot \|\theta_{t+1} - \theta_t\|_2^2, \tag{117}$$

and the last inequality is from [9, Lemma 3]. Denote $\delta(\theta_t) := (\pi^* - \pi_{\theta_t})^\top r$. We have, for all $t \geq 1$,

$$\delta(\theta_{t+1}) - \delta(\theta_t) = -\pi_{\theta_{t+1}}^\top r + \pi_{\theta_t}^\top r + \left\langle \frac{d\pi_{\theta_t}^\top r}{d\theta_t}, \theta_{t+1} - \theta_t \right\rangle - \left\langle \frac{d\pi_{\theta_t}^\top r}{d\theta_t}, \theta_{t+1} - \theta_t \right\rangle \tag{118}$$

$$\leq 3 \cdot \left\| \frac{d\pi_{\theta_t}^\top r}{d\theta_t} \right\|_2 \cdot \|\theta_{t+1} - \theta_t\|_2^2 - \left\langle \frac{d\pi_{\theta_t}^\top r}{d\theta_t}, \theta_{t+1} - \theta_t \right\rangle \qquad \text{(by Eq. (115))} \tag{119}$$

$$= -\frac{1}{12} \cdot \left\| \frac{d\pi_{\theta_t}^\top r}{d\theta_t} \right\|_2 \qquad \text{(using Update 3 and } \eta = 1/6) \tag{120}$$

$$\leq -\frac{1}{12} \cdot \pi_{\theta_t}(a^*) \cdot (\pi^* - \pi_{\theta_t})^\top r \qquad \text{(by Lemma 1)} \tag{121}$$

$$\leq -\frac{c}{12} \cdot \delta(\theta_t), \tag{122}$$

where $c = \inf_{t \geq 1} \pi_{\theta_t}(a^*) > 0$ is from [9, Lemma 4]. Then we have, for all $t \geq 1$,

$$\delta(\theta_t) \leq \delta(\theta_{t-1}) \cdot \left( 1 - \frac{c}{12} \right) \tag{123}$$

$$\leq \delta(\theta_{t-1}) \cdot e^{-\frac{c}{12}} \tag{124}$$

$$\leq \delta(\theta_1) \cdot e^{-\frac{c \cdot (t-1)}{12}}. \tag{125}$$

If $\pi_{\theta_1}(a) = 1/K$, $\forall a$, then similar arguments to Eq. (23) give $c \geq 1/K$. $\square$

## A.2 On-policy Stochastic Gradient Setting

### A.2.1 Softmax PG

**Lemma 5.** Let $\hat{r}$ be the IS estimator using on-policy sampling $a \sim \pi_\theta(\cdot)$. The stochastic softmax PG estimator is unbiased and bounded, i.e.,

$$\underset{a \sim \pi_\theta(\cdot)}{\mathbb{E}} \left[ \frac{d\pi_\theta^\top \hat{r}}{d\theta} \right] = \frac{d\pi_\theta^\top r}{d\theta}, \text{ and} \tag{126}$$

$$\underset{a \sim \pi_\theta(\cdot)}{\mathbb{E}} \left\| \frac{d\pi_\theta^\top \hat{r}}{d\theta} \right\|_2^2 \leq 2. \tag{127}$$

*Proof.* **First part.** $\mathbb{E}_{a \sim \pi_\theta(\cdot)} \left[ \frac{d\pi_\theta^\top \hat{r}}{d\theta} \right] = \frac{d\pi_\theta^\top r}{d\theta}$.

We have, for all $i \in [K]$, the true softmax PG is,

$$\frac{d\pi_\theta^\top r}{d\theta(i)} = \pi_\theta(i) \cdot \left( r(i) - \pi_\theta^\top r \right). \tag{128}$$

On the other hand, we have, for all $i \in [K]$,

$$\frac{d\pi_\theta^\top \hat{r}}{d\theta(i)} = \pi_\theta(i) \cdot \left( \hat{r}(i) - \pi_\theta^\top \hat{r} \right) \tag{129}$$

$$= \pi_\theta(i) \cdot \left( \frac{\mathbb{I}\{a = i\}}{\pi_\theta(i)} \cdot r(i) - \sum_j \mathbb{I}\{a = j\} \cdot r(j) \right) \quad \text{(by Definition 1)} \tag{130}$$

$$= \mathbb{I}\{a = i\} \cdot r(i) - \pi_\theta(i) \cdot r(a). \tag{131}$$

The expectation of stochastic softmax PG is,

$$\underset{a \sim \pi_\theta(\cdot)}{\mathbb{E}} \left[ \frac{d\pi_\theta^\top \hat{r}}{d\theta(i)} \right] = \sum_{a \in [K]} \pi_\theta(a) \cdot \left( \mathbb{I}\{a = i\} \cdot r(i) - \pi_\theta(i) \cdot r(a) \right) \tag{132}$$

$$= \pi_\theta(i) \cdot r(i) - \pi_\theta(i) \cdot \pi_\theta^\top r \tag{133}$$

$$= \frac{d\pi_\theta^\top r}{d\theta(i)}. \tag{134}$$

**Second part.** $\mathbb{E}_{a \sim \pi_\theta(\cdot)} \left\| \frac{d\pi_\theta^\top \hat{r}}{d\theta} \right\|_2^2 \leq 2$.

The squared stochastic PG norm is,

$$\left\| \frac{d\pi_\theta^\top \hat{r}}{d\theta} \right\|_2^2 = \sum_{i=1}^K \left( \frac{d\pi_\theta^\top \hat{r}}{d\theta(i)} \right)^2 = \sum_{i=1}^K \pi_\theta(i)^2 \cdot \left( \hat{r}(i) - \pi_\theta^\top \hat{r} \right)^2 \tag{135}$$

$$= \sum_{i=1}^K \pi_\theta(i)^2 \cdot \left[ \frac{(\mathbb{I}\{a = i\})^2}{\pi_\theta(i)^2} \cdot r(i)^2 - 2 \cdot \frac{\mathbb{I}\{a = i\}}{\pi_\theta(i)} \cdot r(i) \cdot \sum_{j=1}^K \mathbb{I}\{a = j\} \cdot r(j) + \left( \sum_{j=1}^K \mathbb{I}\{a = j\} \cdot r(j) \right)^2 \right] \tag{136}$$

$$= r(a)^2 - 2 \cdot \pi_\theta(a) \cdot r(a)^2 + \sum_{i=1}^K \pi_\theta(i)^2 \cdot r(a)^2 \tag{137}$$

$$= (1 - \pi_\theta(a)) \cdot r(a)^2 - \pi_\theta(a) \cdot r(a)^2 + \pi_\theta(a)^2 \cdot r(a)^2 + \sum_{a' \neq a} \pi_\theta(a')^2 \cdot r(a)^2 \tag{138}$$

$$= (1 - \pi_\theta(a))^2 \cdot r(a)^2 + \sum_{a' \neq a} \pi_\theta(a')^2 \cdot r(a)^2. \tag{139}$$

Taking expectation of $a \sim \pi_\theta(\cdot)$, the expected squared stochastic PG norm is,

$$\underset{a \sim \pi_\theta(\cdot)}{\mathbb{E}} \left\| \frac{d\pi_\theta^\top \hat{r}}{d\theta} \right\|_2^2 = \sum_{a \in [K]} \pi_\theta(a) \cdot (1 - \pi_\theta(a))^2 \cdot r(a)^2 + \sum_{a \in [K]} \pi_\theta(a) \cdot \sum_{a' \neq a} \pi_\theta(a')^2 \cdot r(a)^2 \tag{140}$$

$$\leq \sum_{a \in [K]} \pi_\theta(a) \cdot (1 - \pi_\theta(a))^2 \cdot r(a)^2 + \sum_{a \in [K]} \pi_\theta(a) \cdot \left[ \sum_{a' \neq a} \pi_\theta(a') \right]^2 \cdot r(a)^2 \qquad (\|x\|_2 \leq \|x\|_1) \tag{141}$$

$$= 2 \cdot \sum_{a \in [K]} \pi_\theta(a) \cdot (1 - \pi_\theta(a))^2 \cdot r(a)^2 \tag{142}$$

$$\leq 2 \cdot \sum_{a \in [K]} \pi_\theta(a) \qquad \left( r \in (0, 1]^K, \text{ and } \pi_\theta(a) \in (0, 1) \text{ for all } a \in [K] \right) \tag{143}$$

$$= 2. \qquad \qquad \square$$

**Lemma 7** (Non-uniform Smoothness (NS) between two iterations). *Let $\theta' = \theta + \eta \cdot \frac{d\pi_\theta^\top \hat{r}}{d\theta}$ (using stochastic PG update). We have, for $\eta = \frac{1}{12} \cdot \left\| \frac{d\pi_\theta^\top r}{d\theta} \right\|_2$ (using true PG norm in learning rate),*

$$\left| (\pi_{\theta'} - \pi_\theta)^\top r - \left\langle \frac{d\pi_\theta^\top r}{d\theta}, \theta' - \theta \right\rangle \right| \leq 3 \cdot \left\| \frac{d\pi_\theta^\top r}{d\theta} \right\|_2 \cdot \|\theta' - \theta\|_2^2. \tag{144}$$

*Proof.* Denote $\theta_\zeta := \theta + \zeta \cdot (\theta' - \theta)$ with some $\zeta \in [0, 1]$. According to Taylor's theorem, $\forall \theta, \theta'$,

$$\left| (\pi_{\theta'} - \pi_\theta)^\top r - \left\langle \frac{d\pi_\theta^\top r}{d\theta}, \theta' - \theta \right\rangle \right| = \frac{1}{2} \cdot \left| (\theta' - \theta)^\top \frac{d^2 \pi_{\theta_\zeta}^\top r}{d\theta_\zeta^2} (\theta' - \theta) \right| \tag{145}$$

$$\leq \frac{3}{2} \cdot \left\| \frac{d\pi_{\theta_\zeta}^\top r}{d\theta_\zeta} \right\|_2 \cdot \|\theta' - \theta\|_2^2. \qquad \text{(by Lemma 4)} \tag{146}$$

Denote $\zeta_1 := \zeta$. Also denote $\theta_{\zeta_2} := \theta + \zeta_2 \cdot (\theta_{\zeta_1} - \theta)$ with some $\zeta_2 \in [0, 1]$. We have,

$$\left\| \frac{d\pi_{\theta_{\zeta_1}}^\top r}{d\theta_{\zeta_1}} - \frac{d\pi_\theta^\top r}{d\theta} \right\|_2 = \left\| \int_0^1 \left\langle \frac{d^2 \{\pi_{\theta_{\zeta_2}}^\top r\}}{d\theta_{\zeta_2}^2}, \theta_{\zeta_1} - \theta \right\rangle d\zeta_2 \right\|_2 \tag{147}$$

$$\leq \int_0^1 \left\| \frac{d^2 \{\pi_{\theta_{\zeta_2}}^\top r\}}{d\theta_{\zeta_2}^2} \right\|_2 \cdot \|\theta_{\zeta_1} - \theta\|_2 \, d\zeta_2 \qquad \text{(by Cauchy–Schwarz)} \tag{148}$$

$$\leq \int_0^1 3 \cdot \left\| \frac{d\pi_{\theta_{\zeta_2}}^\top r}{d\theta_{\zeta_2}} \right\|_2 \cdot \zeta_1 \cdot \|\theta' - \theta\|_2 \, d\zeta_2 \qquad \text{(by Lemma 4)} \tag{149}$$

$$\leq \int_0^1 3 \cdot \left\| \frac{d\pi_{\theta_{\zeta_2}}^\top r}{d\theta_{\zeta_2}} \right\|_2 \cdot \eta \cdot \left\| \frac{d\pi_\theta^\top \hat{r}}{d\theta} \right\|_2 d\zeta_2, \qquad \left( \zeta_1 \in [0, 1], \text{ using } \theta' = \theta + \eta \cdot \frac{d\pi_\theta^\top \hat{r}}{d\theta} \right) \tag{150}$$

where the second inequality is because of the Hessian is symmetric, and its operator norm is equal to its spectral radius. Therefore we have,

$$\left\| \frac{d\pi_{\theta_{\zeta_1}}^\top r}{d\theta_{\zeta_1}} \right\|_2 \leq \left\| \frac{d\pi_\theta^\top r}{d\theta} \right\|_2 + \left\| \frac{d\pi_{\theta_{\zeta_1}}^\top r}{d\theta_{\zeta_1}} - \frac{d\pi_\theta^\top r}{d\theta} \right\|_2 \qquad \text{(by triangle inequality)} \tag{151}$$

$$\leq \left\| \frac{d\pi_\theta^\top r}{d\theta} \right\|_2 + 3\eta \cdot \left\| \frac{d\pi_\theta^\top \hat{r}}{d\theta} \right\|_2 \cdot \int_0^1 \left\| \frac{d\pi_{\theta_{\zeta_2}}^\top r}{d\theta_{\zeta_2}} \right\|_2 d\zeta_2. \qquad \text{(by Eq. (147))} \tag{152}$$

Denote $\theta_{\zeta_3} := \theta + \zeta_3 \cdot (\theta_{\zeta_2} - \theta)$ with $\zeta_3 \in [0, 1]$. Using similar calculation in Eq. (147), we have,

$$\left\| \frac{d\pi_{\theta_{\zeta_2}}^\top r}{d\theta_{\zeta_2}} \right\|_2 \leq \left\| \frac{d\pi_\theta^\top r}{d\theta} \right\|_2 + \left\| \frac{d\pi_{\theta_{\zeta_2}}^\top r}{d\theta_{\zeta_2}} - \frac{d\pi_\theta^\top r}{d\theta} \right\|_2 \tag{153}$$

$$\leq \left\| \frac{d\pi_\theta^\top r}{d\theta} \right\|_2 + 3\eta \cdot \left\| \frac{d\pi_\theta^\top \hat{r}}{d\theta} \right\|_2 \cdot \int_0^1 \left\| \frac{d\pi_{\theta_{\zeta_3}}^\top r}{d\theta_{\zeta_3}} \right\|_2 d\zeta_3. \tag{154}$$

Combining Eqs. (151) and (153), we have,

$$\left\| \frac{d\pi_{\theta_{\zeta_1}}^\top r}{d\theta_{\zeta_1}} \right\|_2 \leq \left( 1 + 3\eta \cdot \left\| \frac{d\pi_\theta^\top \hat{r}}{d\theta} \right\|_2 \right) \cdot \left\| \frac{d\pi_\theta^\top r}{d\theta} \right\|_2 + \left( 3\eta \cdot \left\| \frac{d\pi_\theta^\top \hat{r}}{d\theta} \right\|_2 \right)^2 \cdot \int_0^1 \int_0^1 \left\| \frac{d\pi_{\theta_{\zeta_3}}^\top r}{d\theta_{\zeta_3}} \right\|_2 d\zeta_3 d\zeta_2, \tag{155}$$

which implies,

$$\left\| \frac{d\pi_{\theta_{\zeta_1}}^\top r}{d\theta_{\zeta_1}} \right\|_2 \leq \sum_{i=0}^\infty \left( 3\eta \cdot \left\| \frac{d\pi_\theta^\top \hat{r}}{d\theta} \right\|_2 \right)^i \cdot \left\| \frac{d\pi_\theta^\top r}{d\theta} \right\|_2 \tag{156}$$

$$= \frac{1}{1 - 3\eta \cdot \left\| \frac{d\pi_\theta^\top \hat{r}}{d\theta} \right\|_2} \cdot \left\| \frac{d\pi_\theta^\top r}{d\theta} \right\|_2 \qquad \left( 3\eta \cdot \left\| \frac{d\pi_\theta^\top \hat{r}}{d\theta} \right\|_2 \in (0, 1), \text{ see below} \right) \tag{157}$$

$$= \frac{1}{1 - \frac{1}{4} \cdot \left\| \frac{d\pi_\theta^\top \hat{r}}{d\theta} \right\|_2 \cdot \left\| \frac{d\pi_\theta^\top r}{d\theta} \right\|_2} \cdot \left\| \frac{d\pi_\theta^\top r}{d\theta} \right\|_2 \qquad \left( \eta = \frac{1}{12} \cdot \left\| \frac{d\pi_\theta^\top r}{d\theta} \right\|_2 \right) \tag{158}$$

$$\leq \frac{1}{1 - \frac{1}{4} \cdot \left\| \frac{d\pi_\theta^\top \hat{r}}{d\theta} \right\|_2} \cdot \left\| \frac{d\pi_\theta^\top r}{d\theta} \right\|_2 \qquad \left( \left\| \frac{d\pi_\theta^\top r}{d\theta} \right\|_2 \leq 1, \text{ see below} \right) \tag{159}$$

$$\leq \frac{1}{1 - \frac{1}{2}} \cdot \left\| \frac{d\pi_\theta^\top r}{d\theta} \right\|_2 \qquad \left( \left\| \frac{d\pi_\theta^\top \hat{r}}{d\theta} \right\|_2 \leq 2, \text{ see below} \right) \tag{160}$$

$$= 2 \cdot \left\| \frac{d\pi_\theta^\top r}{d\theta} \right\|_2, \tag{161}$$

where the last inequality is from,

$$\left\| \frac{d\pi_\theta^\top \hat{r}}{d\theta} \right\|_2^2 = (1 - \pi_\theta(a))^2 \cdot r(a)^2 + \sum_{a' \neq a} \pi_\theta(a')^2 \cdot r(a)^2 \qquad \text{(by Eq. (135))} \tag{162}$$

$$\leq 2 \cdot (1 - \pi_\theta(a))^2 \cdot r(a)^2 \qquad (\|x\|_2 \leq \|x\|_1) \tag{163}$$

$$\leq 2, \qquad \left( r \in (0, 1]^K, \text{ and } \pi_\theta(a) \in (0, 1) \text{ for all } a \in [K] \right) \tag{164}$$

which implies that,

$$\left\| \frac{d\pi_\theta^\top \hat{r}}{d\theta} \right\|_2 \leq \sqrt{2} \leq 2, \tag{165}$$

and the second last inequality is from,

$$\left\| \frac{d\pi_\theta^\top r}{d\theta} \right\|_2^2 = \sum_{i=1}^K \pi_\theta(i)^2 \cdot \left( r(i) - \pi_\theta^\top r \right)^2 \tag{166}$$

$$\leq \sum_{i=1}^K \pi_\theta(i)^2 \qquad \left( r \in (0, 1]^K \right) \tag{167}$$

$$\leq \left[ \sum_{i=1}^K \pi_\theta(i) \right]^2 \qquad (\|x\|_2 \leq \|x\|_1) \tag{168}$$

$$= 1. \tag{169}$$

Combining Eqs. (145) and (156) finishes the proof. $\qquad \square$

**Theorem 2.** Using Update 4 with learning rate

$$\eta = \frac{1}{12} \cdot \left\| \frac{d\pi_{\theta_t}^\top r}{d\theta_t} \right\|_2, \tag{170}$$

for all $t \geq 1$, we have $(\pi^* - \pi_{\theta_t})^\top r \to 0$ as $t \to \infty$ in probability.

*Proof.* See [12, Proposition 4]. We provide a different proof using the non-uniform smoothness. Denote $\delta(\theta_t) := (\pi^* - \pi_{\theta_t})^\top r$. We have, for all $t \geq 1$,

$$\delta(\theta_{t+1}) - \delta(\theta_t) = -\pi_{\theta_{t+1}}^\top r + \pi_{\theta_t}^\top r + \left\langle \frac{d\pi_{\theta_t}^\top r}{d\theta_t}, \theta_{t+1} - \theta_t \right\rangle - \left\langle \frac{d\pi_{\theta_t}^\top r}{d\theta_t}, \theta_{t+1} - \theta_t \right\rangle \tag{171}$$

$$\leq 3 \cdot \left\| \frac{d\pi_{\theta_t}^\top r}{d\theta_t} \right\|_2 \cdot \|\theta_{t+1} - \theta_t\|_2^2 - \left\langle \frac{d\pi_{\theta_t}^\top r}{d\theta_t}, \theta_{t+1} - \theta_t \right\rangle \qquad \text{(by Lemma 7)} \tag{172}$$

$$= 3 \cdot \eta^2 \cdot \left\| \frac{d\pi_{\theta_t}^\top r}{d\theta_t} \right\|_2 \cdot \left\| \frac{d\pi_{\theta_t}^\top \hat{r}_t}{d\theta_t} \right\|_2^2 - \eta \cdot \left\langle \frac{d\pi_{\theta_t}^\top r}{d\theta_t}, \frac{d\pi_{\theta_t}^\top \hat{r}_t}{d\theta_t} \right\rangle. \qquad \text{(using Update 4)} \tag{173}$$

Next, taking expectation over the random sampling on Eq. (171), we have,

$$\mathbb{E}\left[\delta(\theta_{t+1})\right] - \mathbb{E}\left[\delta(\theta_t)\right] \leq 3 \cdot \eta^2 \cdot \left\| \frac{d\pi_{\theta_t}^\top r}{d\theta_t} \right\|_2 \cdot \mathbb{E}\left[ \left\| \frac{d\pi_{\theta_t}^\top \hat{r}_t}{d\theta_t} \right\|_2^2 \right] - \eta \cdot \left\langle \frac{d\pi_{\theta_t}^\top r}{d\theta_t}, \mathbb{E}\left[ \frac{d\pi_{\theta_t}^\top \hat{r}_t}{d\theta_t} \right] \right\rangle \tag{174}$$

$$= 3 \cdot \eta^2 \cdot \left\| \frac{d\pi_{\theta_t}^\top r}{d\theta_t} \right\|_2 \cdot \mathbb{E}\left[ \left\| \frac{d\pi_{\theta_t}^\top \hat{r}_t}{d\theta_t} \right\|_2^2 \right] - \eta \cdot \left\| \frac{d\pi_{\theta_t}^\top r}{d\theta_t} \right\|_2^2 \qquad \text{(unbiased PG, by Lemma 5)}$$
$$\tag{175}$$

$$\leq 3 \cdot 2 \cdot \eta^2 \cdot \left\| \frac{d\pi_{\theta_t}^\top r}{d\theta_t} \right\|_2 - \eta \cdot \left\| \frac{d\pi_{\theta_t}^\top r}{d\theta_t} \right\|_2^2 \qquad \left( \mathbb{E}\left[ \left\| \frac{d\pi_{\theta_t}^\top \hat{r}_t}{d\theta_t} \right\|_2^2 \right] \leq 2, \text{ by Lemma 5} \right) \tag{176}$$

$$= -\frac{1}{24} \cdot \left\| \frac{d\pi_{\theta_t}^\top r}{d\theta_t} \right\|_2^3 \qquad \left( \text{using } \eta = \frac{1}{12} \cdot \left\| \frac{d\pi_{\theta_t}^\top r}{d\theta_t} \right\|_2 \right) \tag{177}$$

$$\leq -\frac{1}{24} \cdot \mathbb{E}\left[ \pi_{\theta_t}(a^*)^3 \right] \cdot \mathbb{E}\left[ \delta(\theta_t)^3 \right] \qquad \text{(by Lemma 1)} \tag{178}$$

$$\leq -\frac{c}{24} \cdot \left( \mathbb{E}\left[ \delta(\theta_t) \right] \right)^3, \qquad \text{(by Jensen's inequality)} \tag{179}$$

where

$$c := \inf_{t \geq 1} \mathbb{E}\left[ \pi_{\theta_t}(a^*)^3 \right] \tag{180}$$

$$\geq \inf_{t \geq 1} \left( \mathbb{E}\left[ \pi_{\theta_t}(a^*) \right] \right)^3 \qquad \text{(by Jensen's inequality)} \tag{181}$$

$$> 0, \tag{182}$$

and the last inequality is from [2, Lemma 5], since the expected iteration equals the true gradient update, which converges to global optimal policy. Denote $\tilde{\delta}(\theta_t) \coloneqq \mathbb{E}\left[\delta(\theta_t)\right]$. We have, for all $t \geq 1$,

$$\frac{1}{\tilde{\delta}(\theta_t)^2} = \frac{1}{\tilde{\delta}(\theta_1)^2} + \sum_{s=1}^{t-1}\left[\frac{1}{\tilde{\delta}(\theta_{s+1})^2} - \frac{1}{\tilde{\delta}(\theta_s)^2}\right] \tag{183}$$

$$= \frac{1}{\tilde{\delta}(\theta_1)^2} + \sum_{s=1}^{t-1}\frac{1}{\tilde{\delta}(\theta_{s+1})^2}\cdot\left[1 - \frac{\tilde{\delta}(\theta_{s+1})^2}{\tilde{\delta}(\theta_s)^2}\right] \tag{184}$$

$$\geq \frac{1}{\tilde{\delta}(\theta_1)^2} + \sum_{s=1}^{t-1}\frac{2}{\cancel{\tilde{\delta}(\theta_{s+1})^2}}\cdot\frac{\cancel{\tilde{\delta}(\theta_{s+1})^2}}{\tilde{\delta}(\theta_s)^2}\cdot\left[1 - \frac{\tilde{\delta}(\theta_{s+1})}{\tilde{\delta}(\theta_s)}\right] \qquad \left(1 - x^2 \geq 2\cdot x^2\cdot(1-x) \text{ for all } x \in (0,1]\right) \tag{185}$$

$$= \frac{1}{\tilde{\delta}(\theta_1)^2} + 2\cdot\sum_{s=1}^{t-1}\frac{1}{\tilde{\delta}(\theta_s)^3}\cdot\left(\tilde{\delta}(\theta_s) - \tilde{\delta}(\theta_{s+1})\right) \tag{186}$$

$$\geq \frac{1}{\tilde{\delta}(\theta_1)^2} + 2\cdot\sum_{s=1}^{t-1}\frac{1}{\cancel{\tilde{\delta}(\theta_s)^3}}\cdot\frac{c}{24}\cdot\cancel{\tilde{\delta}(\theta_s)^3} \qquad \text{(by Eq. (174))} \tag{187}$$

$$= \frac{1}{\tilde{\delta}(\theta_1)^2} + \frac{c}{12}\cdot(t-1) \tag{188}$$

$$\geq \frac{c\cdot t}{12}, \qquad \left(\tilde{\delta}(\theta_1)^2 \leq 1 < \frac{12}{c}\right) \tag{189}$$

which implies that,

$$\mathbb{E}_{a_t \sim \pi_{\theta_t}(\cdot)}\left[(\pi^* - \pi_{\theta_t})^\top r\right] \leq \frac{\sqrt{12}}{\sqrt{c}}\cdot\frac{1}{\sqrt{t}}, \tag{190}$$

where $c$ is from Eq. (180). This implies $(\pi^* - \pi_{\theta_t})^\top r \to 0$ as $t \to \infty$ in probability, i.e.,

$$\lim_{t \to \infty}\Pr\left((\pi^* - \pi_{\theta_t})^\top r > \epsilon\right) = 0, \tag{191}$$

for all $\epsilon > 0$. $\qquad\qquad\square$

### A.2.2 NPG

**Lemma 6.** For NPG, we have, $\mathbb{E}_{a \sim \pi_\theta(\cdot)}\left[\hat{r}\right] = r$, and $\mathbb{E}_{a \sim \pi_\theta(\cdot)}\|\hat{r}\|_2^2 = \sum_{a \in [K]}\frac{r(a)^2}{\pi_\theta(a)}$.

*Proof.* **First part.** $\mathbb{E}_{a \sim \pi_\theta(\cdot)}\left[\hat{r}\right] = r$.

We have, for all $i \in [K]$,

$$\mathbb{E}_{a \sim \pi_\theta(\cdot)}\left[\hat{r}(i)\right] = \sum_{a \in [K]}\pi_\theta(a)\cdot\frac{\mathbb{I}\{a = i\}}{\pi_\theta(i)}\cdot r(i) = r(i). \tag{192}$$

**Second part.** $\mathbb{E}_{a \sim \pi_\theta(\cdot)}\|\hat{r}\|_2^2 = \sum_{a \in [K]}\frac{r(a)^2}{\pi_\theta(a)}$.

The squared $\ell_2$ norm of natural policy gradient is,

$$\|\hat{r}\|_2^2 = \sum_i \hat{r}(i)^2 = \sum_i \frac{(\mathbb{I}\{a = i\})^2}{\pi_\theta(i)^2}\cdot r(i)^2 = \sum_i \frac{\mathbb{I}\{a = i\}}{\pi_\theta(i)^2}\cdot r(i)^2. \tag{193}$$

The expected squared norm is,

$$\mathbb{E}_{a \sim \pi_\theta(\cdot)} \|\hat{r}\|_2^2 = \sum_{a \in [K]} \pi_\theta(a) \cdot \sum_i \frac{\mathbb{I}\{a = i\}}{\pi_\theta(i)^2} \cdot r(i)^2 \tag{194}$$

$$= \sum_{a \in [K]} \pi_\theta(a) \cdot \frac{1}{\pi_\theta(a)^2} \cdot r(a)^2 \tag{195}$$

$$= \sum_{a \in [K]} \frac{r(a)^2}{\pi_\theta(a)}. \qquad \square$$

**Theorem 3.** Using Update 5, we have: **(i)** with positive probability, $\sum_{a \neq a^*} \pi_{\theta_t}(a) \to 1$ as $t \to \infty$; **(ii)** $\forall a \in [K]$, with positive probability, $\pi_{\theta_t}(a) \to 1$, as $t \to \infty$.

*Proof.* **First part.** With positive probability, $\sum_{a \neq a^*} \pi_{\theta_t}(a) \to 1$ as $t \to \infty$.

Let Pr denote the probability measure that over the probability space $(\Omega, \mathcal{F})$ that holds all of our random variables. Let $\mathcal{B} = \{a \in [K] : a \neq a^*\}$. By abusing notation, for any $\pi : [K] \to [0, 1]$ map we let $\pi_{\theta_t}(\mathcal{B})$ to stand for $\sum_{a \in \mathcal{B}} \pi_{\theta_t}(a)$. Define for $t \geq 1$ the event $\mathcal{B}_t = \{a_t \neq a^*\}(= \{a_t \in \mathcal{B}\})$ and let $\mathcal{E}_t = \mathcal{B}_1 \cap \cdots \cap \mathcal{B}_t$. Thus, $\mathcal{E}_t$ is the event that $a^*$ was not chosen in the first $t$ time steps. Note that $\{\mathcal{E}_t\}_{t \geq 1}$ is a nested sequence and thus, by the monotone convergence theorem,

$$\lim_{t \to \infty} \Pr(\mathcal{E}_t) = \Pr(\mathcal{E}), \tag{196}$$

where $\mathcal{E} = \cap_{t \geq 1} \mathcal{B}_t$. We start with a lower bound on the probability of $\mathcal{E}_t$. The lower bound is stated in a generic form: In particular, let $(b_t)_{t \geq 1}$ be a deterministic sequence which satisfies that for any $t \geq 1$,

$$\mathbb{I}_{\mathcal{E}_{t-1}} \cdot \pi_{\theta_t}(\mathcal{B}) \geq \mathbb{I}_{\mathcal{E}_{t-1}} \cdot b_t \qquad \text{holds Pr-almost surely,} \tag{197}$$

where we let $\mathcal{E}_0 = \Omega$ and for an event $\mathcal{E}$, $\mathbb{I}_\mathcal{E}$ stands for the characteristic function of $\mathcal{E}$ (i.e., $\mathbb{I}_\mathcal{E}(\omega) = 1$ if $\omega \in \mathcal{E}$ and $\mathbb{I}_\mathcal{E}(\omega) = 0$, otherwise). We make the following claim:

Claim 1: Under the above assumption, for any $t \geq 1$ it holds that

$$\Pr(\mathcal{E}_t) \geq \prod_{s=1}^t b_s. \tag{198}$$

For the proof of this claim let $\mathcal{H}_t$ denote the sequence formed of the first $t$ actions:

$$\mathcal{H}_t := (a_1, a_2, \cdots, a_t). \tag{199}$$

By definition,

$$\theta_t = \mathcal{A}(\theta_1, a_1, r(a_1), \theta_2, a_2, r(a_2), \cdots, \theta_{t-1}, a_{t-1}, r(a_{t-1})). \tag{200}$$

By our assumption that the $t$th action is chosen from $\pi_{\theta_t}$, it follows that Pr satisfies that for all $a$ and $t \geq 1$,

$$\Pr(a_t = a \mid \mathcal{H}_{t-1}) = \pi_{\theta_t}(a) \qquad \text{Pr-almost surely.} \tag{201}$$

We prove the claim by induction on $t$. For $t = 1$, from Eqs. (197) and (201), using that $\mathcal{E}_0 = \Omega$ and $H_0 = ()$, we have that Pr-almost surely,

$$\Pr(\mathcal{E}_1) = \pi_{\theta_1}(\mathcal{B}). \tag{202}$$

Suppose the claim holds up to $t - 1$. We have,

$$\begin{aligned}
\Pr(\mathcal{E}_t) &= \mathbb{E}[\Pr(\mathcal{E}_t \mid \mathcal{H}_{t-1})] && \text{(by the tower rule)} \\
&= \mathbb{E}[\mathbb{I}_{\mathcal{E}_{t-1}} \cdot \Pr(\mathcal{B}_t \mid \mathcal{H}_{t-1})] && (\mathcal{E}_{t-1} \text{ is } \mathcal{H}_{t-1}\text{-measurable}) \\
&= \mathbb{E}[\mathbb{I}_{\mathcal{E}_{t-1}} \cdot \pi_{\theta_t}(\mathcal{B})] && \text{(by Eq. (201))} \\
&\geq \mathbb{E}[\mathbb{I}_{\mathcal{E}_{t-1}} \cdot b_t] && \text{(by Eq. (197))} \\
&= b_t \cdot \Pr(\mathcal{E}_{t-1}) && (b_t \text{ is deterministic}) \\
&= \prod_{s=1}^t b_s. && \text{(induction hypothesis)}
\end{aligned}$$

Now, we claim the following:

Claim 2: A suitable choice for $b_t$ is

$$b_t = \exp\left\{\frac{-\exp\{\theta_1(a^*)\}}{(K-1)\cdot\exp\left\{\frac{\sum_{a\neq a^*}\theta_1(a)+\eta\cdot r_{\min}\cdot(t-1)}{K-1}\right\}}\right\}. \tag{203}$$

Proof of Claim 2: Clearly, it suffices to show that for any sequence $(a_1,\ldots,a_{t-1})$ such that $a_s\neq a^*$, $\theta_t := \mathcal{A}(\theta_1, a_1, r(a_1),\ldots, a_{t-1}, r(a_{t-1}))$ is such that $\pi_{\theta_t}(\mathcal{B}) \geq b_t$ with $b_t$ as defined in Eq. (203).

We have, for each sub-optimal action $a \neq a^*$,

$$\theta_t(a) = \theta_1(a) + \eta\cdot\sum_{s=1}^{t-1}\hat{r}_s(a) \qquad \text{(by Update 5)} \tag{204}$$

$$= \theta_1(a) + \eta\cdot\sum_{s=1}^{t-1}\frac{\mathbb{I}\{a_s=a\}}{\pi_{\theta_s}(a)}\cdot r(a) \qquad \text{(by Definition 1)} \tag{205}$$

$$\geq \theta_1(a) + \eta\cdot\sum_{s=1}^{t-1}\mathbb{I}\{a_s=a\}\cdot r(a) \qquad (\pi_{\theta_s}(a)\in(0,1),\ r(a)\in(0,1]) \tag{206}$$

$$\geq \theta_1(a) + \eta\cdot r_{\min}\cdot\sum_{s=1}^{t-1}\mathbb{I}\{a_s=a\}, \qquad \left(r_{\min} := \min_{a\neq a^*}r(a)\right) \tag{207}$$

where $r_{\min}\in(0,1]$ according to Assumption 1, i.e., $r(a)\in(0,1]$ for all $a\in[K]$. Then we have,

$$\sum_{a\neq a^*}\exp\{\theta_t(a)\} \geq (K-1)\cdot\exp\left\{\frac{\sum_{a\neq a^*}\theta_t(a)}{K-1}\right\} \qquad \text{(by Jensen's inequality)} \tag{208}$$

$$\geq (K-1)\cdot\exp\left\{\frac{\sum_{a\neq a^*}\theta_1(a)+\eta\cdot r_{\min}\cdot\sum_{a\neq a^*}\sum_{s=1}^{t-1}\mathbb{I}\{a_s=a\}}{K-1}\right\} \qquad \text{(by Eq. (204))} \tag{209}$$

$$= (K-1)\cdot\exp\left\{\frac{\sum_{a\neq a^*}\theta_1(a)+\eta\cdot r_{\min}\cdot(t-1)}{K-1}\right\}. \qquad (a_1\neq a^*, a_2\neq a^*,\cdots, a_{t-1}\neq a^*) \tag{210}$$

On the other hand, we have,

$$\theta_t(a^*) = \theta_1(a^*) + \eta\cdot\sum_{s=1}^{t-1}\frac{\mathbb{I}\{a_s=a^*\}}{\pi_{\theta_s}(a^*)}\cdot r(a^*) \qquad \text{(by Update 5 and Definition 1)} \tag{211}$$

$$= \theta_1(a^*). \qquad (a_s\neq a^* \text{ for all } s\in\{1,2,\ldots,t-1\}) \tag{212}$$

Next, we have,

$$\sum_{a\neq a^*}\pi_{\theta_t}(a) = 1 - \pi_{\theta_t}(a^*) \tag{213}$$

$$= 1 - \frac{\exp\{\theta_t(a^*)\}}{\sum_{a\neq a^*}\exp\{\theta_t(a)\}+\exp\{\theta_t(a^*)\}} \tag{214}$$

$$\geq 1 - \frac{\exp\{\theta_1(a^*)\}}{(K-1)\cdot\exp\left\{\frac{\sum_{a\neq a^*}\theta_1(a)+\eta\cdot r_{\min}\cdot(t-1)}{K-1}\right\}+\exp\{\theta_1(a^*)\}}. \qquad \text{(by Eqs. (208) and (211))} \tag{215}$$

According to Lemma 14, for all $x\in(0,1)$,

$$1 - x \geq \exp\left\{\frac{-1}{1/x-1}\right\}. \tag{216}$$

Let

$$x = \frac{\exp\{\theta_1(a^*)\}}{(K-1) \cdot \exp\left\{\frac{\sum_{a \neq a^*} \theta_1(a) + \eta \cdot r_{\min} \cdot (t-1)}{K-1}\right\} + \exp\{\theta_1(a^*)\}} \in (0,1). \tag{217}$$

We have,

$$\sum_{a \neq a^*} \pi_{\theta_t}(a) \geq \exp\left\{\frac{-1}{\frac{(K-1)\cdot\exp\left\{\frac{\sum_{a \neq a^*} \theta_1(a) + \eta \cdot r_{\min} \cdot (t-1)}{K-1}\right\} + \exp\{\theta_1(a^*)\}}{\exp\{\theta_1(a^*)\}} - 1}\right\} \qquad \text{(by Eqs. (213) and (216))}$$

$$\tag{218}$$

$$= \exp\left\{\frac{-\exp\{\theta_1(a^*)\}}{(K-1) \cdot \exp\left\{\frac{\sum_{a \neq a^*} \theta_1(a) + \eta \cdot r_{\min} \cdot (t-1)}{K-1}\right\}}\right\} = b_t, \tag{219}$$

finishing the proof of the claim.

Combining Eq. (196) with the conclusions of Claim 1 and 2 together, we get

$$\Pr(\mathcal{E}) \geq \prod_{t=1}^{\infty} \exp\left\{\frac{-\exp\{\theta_1(a^*)\}}{(K-1) \cdot \exp\left\{\frac{\sum_{a \neq a^*} \theta_1(a) + \eta \cdot r_{\min} \cdot (t-1)}{K-1}\right\}}\right\} \qquad \text{(by Eq. (218))} \tag{220}$$

$$= \exp\left\{-\frac{\exp\{\theta_1(a^*)\}}{\exp\left\{\frac{\sum_{a \neq a^*} \theta_1(a)}{K-1}\right\}} \cdot \frac{\exp\left\{\frac{\eta \cdot r_{\min}}{K-1}\right\}}{K-1} \cdot \sum_{t=1}^{\infty} \frac{1}{\exp\left\{\frac{\eta \cdot r_{\min} \cdot t}{K-1}\right\}}\right\} \tag{221}$$

$$\geq \exp\left\{-\frac{\exp\{\theta_1(a^*)\}}{\exp\left\{\frac{\sum_{a \neq a^*} \theta_1(a)}{K-1}\right\}} \cdot \frac{\exp\left\{\frac{\eta \cdot r_{\min}}{K-1}\right\}}{K-1} \cdot \int_{t=0}^{\infty} \frac{1}{\exp\left\{\frac{\eta \cdot r_{\min} \cdot t}{K-1}\right\}} dt\right\} \tag{222}$$

$$= \exp\left\{-\frac{\exp\{\theta_1(a^*)\}}{\exp\left\{\frac{\sum_{a \neq a^*} \theta_1(a)}{K-1}\right\}} \cdot \frac{\exp\left\{\frac{\eta \cdot r_{\min}}{K-1}\right\}}{K-1} \cdot \frac{K-1}{\eta \cdot r_{\min}}\right\} \tag{223}$$

$$= \exp\left\{-\frac{\exp\{\theta_1(a^*)\}}{\exp\left\{\frac{\sum_{a \neq a^*} \theta_1(a)}{K-1}\right\}} \cdot \frac{\exp\left\{\frac{\eta \cdot r_{\min}}{K-1}\right\}}{\eta \cdot r_{\min}}\right\}. \tag{224}$$

Note that $r_{\min} \in \Theta(1)$, $\exp\{\theta_1(a^*)\} \in \Theta(1)$, $\eta \in \Theta(1)$, $\exp\left\{\frac{\eta \cdot r_{\min}}{K-1}\right\} \in \Theta(1)$ and,

$$\exp\left\{\frac{\sum_{a \neq a^*} \theta_1(a)}{K-1}\right\} \in \Theta(1). \tag{225}$$

Therefore, we have "the probability of sampling sub-optimal actions forever using on-policy sampling $a_t \sim \pi_{\theta_t}(\cdot)$" is lower bounded by a constant of $\frac{1}{\exp\{\Theta(1)\}} \in \Theta(1)$, which implies that with positive probability $\Theta(1)$, we have $\sum_{a \neq a^*} \pi_{\theta_t}(a) \to 1$ as $t \to \infty$.

**Second part.** $\forall a \in [K]$, with positive probability, $\pi_{\theta_t}(a) \to 1$, as $t \to \infty$.

The proof is similar to the first part. Let $\mathcal{B} = \{a\}$. For any $\pi : [K] \to [0,1]$ map we let $\pi_{\theta_t}(\mathcal{B})$ to stand for $\pi_{\theta_t}(a)$. Define for $t \geq 1$ the event $\mathcal{B}_t = \{a_t = a\}(= \{a_t \in \mathcal{B}\})$ and let $\mathcal{E}_t = \mathcal{B}_1 \cap \cdots \cap \mathcal{B}_t$. Thus, $\mathcal{E}_t$ is the event that $a$ was chosen in the first $t$ time steps. Note that $\{\mathcal{E}_t\}_{t \geq 1}$ is a nested sequence and thus, by the monotone convergence theorem, $\lim_{t \to \infty} \Pr(\mathcal{E}_t) = \Pr(\mathcal{E})$, where $\mathcal{E} = \cap_{t \geq 1} \mathcal{B}_t$. We show that by letting

$$b_t = \exp\left\{\frac{-\sum_{a' \neq a} \exp\{\theta_1(a')\}}{\exp\{\theta_1(a) + \eta \cdot r(a) \cdot (t-1)\}}\right\}, \tag{226}$$

we have Eqs. (197) and (198) hold using the arguments in the first part.

It suffices to show that for any sequence $(a_1, \ldots, a_{t-1})$ such that $a_s = a$, for all $s \in \{1, 2, \ldots, t-1\}$, $\theta_t := \mathcal{A}(\theta_1, a_1, r(a_1), \ldots, a_{t-1}, r(a_{t-1}))$ is such that $\pi_{\theta_t}(\mathcal{B}) \geq b_t$ with $b_t$ as defined in Eq. (226). Now suppose $a_1 = a, a_2 = a, \cdots, a_{t-1} = a$. We have,

$$\theta_t(a) = \theta_1(a) + \eta \cdot \sum_{s=1}^{t-1} \hat{r}_s(a) \qquad \text{(by Update 5)} \tag{227}$$

$$= \theta_1(a) + \eta \cdot \sum_{s=1}^{t-1} \frac{\mathbb{I}\{a_s = a\}}{\pi_{\theta_s}(a)} \cdot r(a) \qquad \text{(by Definition 1)} \tag{228}$$

$$= \theta_1(a) + \eta \cdot \sum_{s=1}^{t-1} \frac{r(a)}{\pi_{\theta_s}(a)} \qquad (a_s = a \text{ for all } s \in \{1, 2, \ldots, t-1\}) \tag{229}$$

$$\geq \theta_1(a) + \eta \cdot \sum_{s=1}^{t-1} r(a) \qquad (\pi_{\theta_s}(a) \in (0, 1)) \tag{230}$$

$$= \theta_1(a) + \eta \cdot r(a) \cdot (t-1). \tag{231}$$

On the other hand, we have, for any other action $a' \neq a$,

$$\theta_t(a') = \theta_1(a') + \eta \cdot \sum_{s=1}^{t-1} \frac{\mathbb{I}\{a_s = a'\}}{\pi_{\theta_s}(a')} \cdot r(a') \qquad \text{(by Update 5 and Definition 1)} \tag{232}$$

$$= \theta_1(a'). \qquad (a_s \neq a' \text{ for all } s \in \{1, 2, \ldots, t-1\}) \tag{233}$$

Therefore, we have,

$$\pi_{\theta_t}(a) = 1 - \sum_{a' \neq a} \pi_{\theta_t}(a') \tag{234}$$

$$= 1 - \frac{\sum_{a' \neq a} \exp\{\theta_t(a')\}}{\exp\{\theta_t(a)\} + \sum_{a' \neq a} \exp\{\theta_t(a')\}} \tag{235}$$

$$\geq 1 - \frac{\sum_{a' \neq a} \exp\{\theta_1(a')\}}{\exp\{\theta_1(a) + \eta \cdot r(a) \cdot (t-1)\} + \sum_{a' \neq a} \exp\{\theta_1(a')\}}. \qquad \text{(by Eqs. (227) and (232))} \tag{236}$$

Let

$$x = \frac{\sum_{a' \neq a} \exp\{\theta_1(a')\}}{\exp\{\theta_1(a) + \eta \cdot r(a) \cdot (t-1)\} + \sum_{a' \neq a} \exp\{\theta_1(a')\}} \in (0, 1). \tag{237}$$

We have,

$$\pi_{\theta_t}(a) \geq 1 - x \qquad \text{(by Eq. (234))} \tag{238}$$

$$\geq \exp\left\{ \frac{-1}{\frac{\exp\{\theta_1(a) + \eta \cdot r(a) \cdot (t-1)\} + \sum_{a' \neq a} \exp\{\theta_1(a')\}}{\sum_{a' \neq a} \exp\{\theta_1(a')\}} - 1} \right\} \qquad \text{(by Lemma 14)} \tag{239}$$

$$= \exp\left\{ \frac{-\sum_{a' \neq a} \exp\{\theta_1(a')\}}{\exp\{\theta_1(a) + \eta \cdot r(a) \cdot (t-1)\}} \right\} \tag{240}$$

$$= b_t. \tag{241}$$

Therefore we have,

$$\prod_{t=1}^{\infty} \pi_{\theta_t}(a) \geq \prod_{t=1}^{\infty} \exp\left\{ \frac{-\sum_{a' \neq a} \exp\{\theta_1(a')\}}{\exp\{\theta_1(a) + \eta \cdot r(a) \cdot (t-1)\}} \right\} \qquad \text{(by Eq. (238))} \tag{242}$$

$$= \exp\left\{ -\sum_{a' \neq a} \exp\{\theta_1(a')\} \cdot \frac{\exp\{\eta \cdot r(a)\}}{\exp\{\theta_1(a)\}} \cdot \sum_{t=1}^{\infty} \frac{1}{\exp\{\eta \cdot r(a) \cdot t\}} \right\} \tag{243}$$

$$\geq \exp\left\{ -\sum_{a' \neq a} \exp\{\theta_1(a')\} \cdot \frac{\exp\{\eta \cdot r(a)\}}{\exp\{\theta_1(a)\}} \cdot \int_{t=0}^{\infty} \frac{1}{\exp\{\eta \cdot r(a) \cdot t\}} dt \right\} \tag{244}$$

$$= \exp\left\{ -\frac{\exp\{\eta \cdot r(a)\}}{\eta \cdot r(a)} \cdot \frac{\sum_{a' \neq a} \exp\{\theta_1(a')\}}{\exp\{\theta_1(a)\}} \right\} \tag{245}$$

$$\in \Omega(1), \tag{246}$$

where the last line is due to $r(a) \in \Theta(1)$, $\exp\{\theta_1(a)\} \in \Theta(1)$ for all $a \in [K]$, and $\eta \in \Theta(1)$. With Eq. (242), we have "the probability of sampling action $a$ forever using on-policy sampling $a_t \sim \pi_{\theta_t}(\cdot)$" is lower bounded by a constant of $\Omega(1)$. Therefore, for all $a \in [K]$, with positive probability $\Omega(1)$, $\pi_{\theta_t}(a) \to 1$, as $t \to \infty$. $\qquad \square$

### A.2.3 GNPG

**Lemma 8.** *Using on-policy IS estimator of Definition 1, the stochastic GNPG is biased, i.e.,*

$$\mathbb{E}_{a \sim \pi_\theta(\cdot)} \left[ \frac{d\pi_\theta^\top \hat{r}}{d\theta} \bigg/ \left\| \frac{d\pi_\theta^\top \hat{r}}{d\theta} \right\|_2 \right] \neq \frac{d\pi_\theta^\top r}{d\theta} \bigg/ \left\| \frac{d\pi_\theta^\top r}{d\theta} \right\|_2. \tag{247}$$

*Proof.* Consider a two-action example with $r(1) > r(2)$. The true normalized PG of $a^* = 1$ is,

$$g(1) := \frac{d\pi_\theta^\top r}{d\theta(1)} \bigg/ \left\| \frac{d\pi_\theta^\top r}{d\theta} \right\|_2 \tag{248}$$

$$= \frac{\pi_\theta(1) \cdot \left( r(1) - \pi_\theta^\top r \right)}{\sqrt{\pi_\theta(1)^2 \cdot \left( r(1) - \pi_\theta^\top r \right)^2 + \pi_\theta(2)^2 \cdot \left( r(2) - \pi_\theta^\top r \right)^2}} \tag{249}$$

$$= \frac{\pi_\theta(1) \cdot \pi_\theta(2) \cdot (r(1) - r(2))}{\sqrt{\pi_\theta(1)^2 \cdot \pi_\theta(2)^2 \cdot (r(1) - r(2))^2 + \pi_\theta(1)^2 \cdot \pi_\theta(2)^2 \cdot (r(1) - r(2))^2}} \tag{250}$$

$$= \frac{1}{\sqrt{2}}. \tag{251}$$

On the other hand, the stochastic normalized PG of $a^* = 1$ is,

$$\hat{g}(1) := \mathbb{E}_{a \sim \pi_\theta(\cdot)} \left[ \frac{d\pi_\theta^\top \hat{r}}{d\theta(1)} \bigg/ \left\| \frac{d\pi_\theta^\top \hat{r}}{d\theta} \right\|_2 \right] \tag{252}$$

$$= \pi_\theta(1) \cdot \frac{\pi_\theta(1) \cdot \left( \frac{r(1)}{\pi_\theta(1)} - \pi_\theta(1) \cdot \frac{r(1)}{\pi_\theta(1)} \right)}{\sqrt{\pi_\theta(1)^2 \cdot \left( \frac{r(1)}{\pi_\theta(1)} - \pi_\theta(1) \cdot \frac{r(1)}{\pi_\theta(1)} \right)^2 + \pi_\theta(2)^2 \cdot \left( 0 - \pi_\theta(1) \cdot \frac{r(1)}{\pi_\theta(1)} \right)^2}} \tag{253}$$

$$+ \pi_\theta(2) \cdot \frac{\pi_\theta(1) \cdot \left( 0 - \pi_\theta(2) \cdot \frac{r(2)}{\pi_\theta(2)} \right)}{\sqrt{\pi_\theta(1)^2 \cdot \left( 0 - \pi_\theta(2) \cdot \frac{r(2)}{\pi_\theta(2)} \right)^2 + \pi_\theta(2)^2 \cdot \left( \frac{r(2)}{\pi_\theta(2)} - \pi_\theta(2) \cdot \frac{r(2)}{\pi_\theta(2)} \right)^2}} \tag{254}$$

$$= \pi_\theta(1) \cdot \frac{\pi_\theta(2) \cdot r(1)}{\sqrt{\pi_\theta(2)^2 \cdot r(1)^2 + \pi_\theta(2)^2 \cdot r(1)^2}} - \pi_\theta(2) \cdot \frac{\pi_\theta(1) \cdot r(2)}{\sqrt{\pi_\theta(1)^2 \cdot r(2)^2 + \pi_\theta(1)^2 \cdot r(2)^2}} \tag{255}$$

$$= \frac{1}{\sqrt{2}} \cdot (\pi_\theta(1) - \pi_\theta(2)). \tag{256}$$

It is clear that the true normalized PG of $a^* = 1$ is always positive $g(1) > 0$, while the expectation of the stochastic normalized PG estimator of $a^* = 1$ is negative when $\pi_\theta(1) < \pi_\theta(2)$. $\qquad\square$

**Theorem 4.** Using Update 6, we have, $\forall a \in [K]$, with positive probability, $\pi_{\theta_t}(a) \to 1$, as $t \to \infty$.

*Proof.* The proof is similar to the second part of Theorem 3. We first calculate the stochastic normalized PG in each iteration. Denote $a_t$ as the action sampled at $t$-th iteration. We have,

$$\frac{d\pi_{\theta_t}^\top \hat{r}_t}{d\theta_t(a_t)} = \pi_{\theta_t}(a_t) \cdot (\hat{r}_t(a_t) - \pi_{\theta_t}^\top \hat{r}_t) \tag{257}$$

$$= \pi_{\theta_t}(a_t) \cdot \left( \frac{r(a_t)}{\pi_{\theta_t}(a_t)} - \pi_{\theta_t}(a_t) \cdot \frac{r(a_t)}{\pi_{\theta_t}(a_t)} \right) \qquad \text{(by Definition 1)} \tag{258}$$

$$= (1 - \pi_{\theta_t}(a_t)) \cdot r(a_t). \tag{259}$$

On the other hand, for all $a' \neq a_t$,

$$\frac{d\pi_{\theta_t}^\top \hat{r}_t}{d\theta_t(a')} = \pi_{\theta_t}(a') \cdot (\hat{r}_s(a') - \pi_{\theta_t}^\top \hat{r}_t) \tag{260}$$

$$= \pi_{\theta_t}(a') \cdot \left( 0 - \pi_{\theta_t}(a_t) \cdot \frac{r(a_t)}{\pi_{\theta_t}(a_t)} \right) \qquad \text{(by Definition 1)} \tag{261}$$

$$= -\pi_{\theta_t}(a') \cdot r(a_t). \tag{262}$$

Therefore, the stochastic PG norm is,

$$\left\| \frac{d\pi_{\theta_t}^\top \hat{r}_t}{d\theta_t} \right\|_2 = \left[ \left( \frac{d\pi_{\theta_t}^\top \hat{r}_t}{d\theta_t(a_t)} \right)^2 + \sum_{a' \neq a_t} \left( \frac{d\pi_{\theta_t}^\top \hat{r}_t}{d\theta_t(a')} \right)^2 \right]^{\frac{1}{2}} \tag{263}$$

$$= \left[ (1 - \pi_{\theta_t}(a_t))^2 \cdot r(a_t)^2 + \sum_{a' \neq a_t} \pi_{\theta_t}(a')^2 \cdot r(a_t)^2 \right]^{\frac{1}{2}} \qquad \text{(by Eqs. (257) and (260))} \tag{264}$$

$$\leq \left[ (1 - \pi_{\theta_t}(a_t))^2 \cdot r(a_t)^2 + \left( \sum_{a' \neq a_t} \pi_{\theta_t}(a') \right)^2 \cdot r(a_t)^2 \right]^{\frac{1}{2}} \qquad (\|x\|_2 \leq \|x\|_1) \tag{265}$$

$$= \sqrt{2} \cdot (1 - \pi_{\theta_t}(a_t)) \cdot r(a_t). \tag{266}$$

The proof is then similar to the second part of Theorem 3. Let $\mathcal{B} = \{a\}$. For any $\pi : [K] \to [0, 1]$ map we let $\pi_{\theta_t}(\mathcal{B})$ to stand for $\pi_{\theta_t}(a)$. Define for $t \geq 1$ the event $\mathcal{B}_t = \{a_t = a\} (= \{a_t \in \mathcal{B}\})$ and let $\mathcal{E}_t = \mathcal{B}_1 \cap \cdots \cap \mathcal{B}_t$. Thus, $\mathcal{E}_t$ is the event that $a$ was chosen in the first $t$ time steps. Note that $\{\mathcal{E}_t\}_{t \geq 1}$ is a nested sequence and thus, by the monotone convergence theorem, $\lim_{t \to \infty} \Pr(\mathcal{E}_t) = \Pr(\mathcal{E})$, where $\mathcal{E} = \cap_{t \geq 1} \mathcal{B}_t$. We show that by letting

$$b_t = \exp \left\{ \frac{-\sum_{a' \neq a} \exp\{\theta_1(a')\}}{\exp\left\{ \theta_1(a) + \frac{\eta}{\sqrt{2}} \cdot (t - 1) \right\}} \right\}, \tag{267}$$

we have Eqs. (197) and (198) hold using the arguments in the first part of Theorem 3.

It suffices to show that for any sequence $(a_1, \ldots, a_{t-1})$ such that $a_s = a$, for all $s \in \{1, 2, \ldots, t-1\}$, $\theta_t := \mathcal{A}(\theta_1, a_1, r(a_1), \ldots, a_{t-1}, r(a_{t-1}))$ is such that $\pi_{\theta_t}(\mathcal{B}) \geq b_t$ with $b_t$ as defined in Eq. (267). Now suppose $a_1 = a, a_2 = a, \cdots, a_{t-1} = a$. We have,

$$\theta_t(a) = \theta_1(a) + \eta \cdot \sum_{s=1}^{t-1} \frac{d\pi_{\theta_s}^\top \hat{r}_s}{d\theta_s(a)} \Big/ \left\| \frac{d\pi_{\theta_s}^\top \hat{r}_s}{d\theta_s} \right\|_2 \qquad \text{(by Update 6)} \tag{268}$$

$$\geq \theta_1(a) + \eta \cdot \sum_{s=1}^{t-1} \frac{(1 - \pi_{\theta_s}(a)) \cdot r(a)}{\sqrt{2} \cdot (1 - \pi_{\theta_s}(a)) \cdot r(a)} \qquad \text{(by Eqs. (257) and (263))} \tag{269}$$

$$= \theta_1(a) + \frac{\eta}{\sqrt{2}} \cdot (t - 1). \tag{270}$$

On the other hand, for all $a' \neq a$, we have,

$$\theta_t(a') = \theta_1(a') - \eta \cdot \sum_{s=1}^{t-1} (\pi_{\theta_s}(a') \cdot r(a)) \Big/ \left\| \frac{d\pi_{\theta_s}^\top \hat{r}_s}{d\theta_s} \right\|_2 \qquad \text{(by Update 6 and Eq. (260))} \quad (271)$$

$$\leq \theta_1(a'). \tag{272}$$

Then we have,

$$\pi_{\theta_t}(a) = 1 - \frac{\sum_{a' \neq a} \exp\{\theta_t(a')\}}{\exp\{\theta_t(a)\} + \sum_{a' \neq a} \exp\{\theta_t(a')\}} \tag{273}$$

$$\geq 1 - \frac{\sum_{a' \neq a} \exp\{\theta_1(a')\}}{\exp\left\{\theta_1(a) + \frac{\eta}{\sqrt{2}} \cdot (t-1)\right\} + \sum_{a' \neq a} \exp\{\theta_1(a')\}} \qquad \text{(by Eqs. (268) and (271))}$$

$$\tag{274}$$

$$\geq \exp\left\{ \frac{-\sum_{a' \neq a} \exp\{\theta_1(a')\}}{\exp\left\{\theta_1(a) + \frac{\eta}{\sqrt{2}} \cdot (t-1)\right\}} \right\} \qquad \text{(by Lemma 14)} \tag{275}$$

$$= b_t. \tag{276}$$

Using similar calculation to Eq. (238), we have,

$$\prod_{t=1}^{\infty} \pi_{\theta_t}(a) \geq \prod_{t=1}^{\infty} \exp\left\{ \frac{-\sum_{a' \neq a} \exp\{\theta_1(a')\}}{\exp\left\{\theta_1(a) + \frac{\eta}{\sqrt{2}} \cdot (t-1)\right\}} \right\} \qquad \text{(by Eq. (273))} \tag{277}$$

$$= \exp\left\{ -\frac{\sum_{a' \neq a} \exp\{\theta_1(a')\}}{\exp\{\theta_1(a)\}} \cdot \exp\left\{\frac{\eta}{\sqrt{2}}\right\} \cdot \sum_{t=1}^{\infty} \frac{1}{\exp\left\{\frac{\eta}{\sqrt{2}} \cdot t\right\}} \right\} \tag{278}$$

$$\geq \exp\left\{ -\frac{\sum_{a' \neq a} \exp\{\theta_1(a')\}}{\exp\{\theta_1(a)\}} \cdot \exp\left\{\frac{\eta}{\sqrt{2}}\right\} \cdot \int_{t=0}^{\infty} \frac{1}{\exp\left\{\frac{\eta}{\sqrt{2}} \cdot t\right\}} dt \right\} \tag{279}$$

$$= \exp\left\{ -\frac{\sum_{a' \neq a} \exp\{\theta_1(a')\}}{\exp\{\theta_1(a)\}} \cdot \frac{\sqrt{2} \cdot \exp\left\{\frac{\eta}{\sqrt{2}}\right\}}{\eta} \right\} \tag{280}$$

$$\in \Omega(1), \tag{281}$$

where the last line is due to, $\exp\{\theta_1(a)\} \in \Theta(1)$ for all $a \in [K]$, and $\eta \in \Theta(1)$. With Eq. (277), we have "the probability of sampling action $a$ forever using on-policy sampling $a_t \sim \pi_{\theta_t}(\cdot)$" is lower bounded by a constant of $\Omega(1)$. Therefore, for all $a \in [K]$, with positive probability $\Omega(1)$, $\pi_{\theta_t}(a) \to 1$, as $t \to \infty$. $\qquad \square$

## B   Proofs for Committal Rate (Section 3)

**Theorem 5** (Committal rate main theorem). Consider a policy optimization method $\mathcal{A}$, together with $r \in (0,1]^K$ and an initial parameter vector $\theta_1 \in \mathbb{R}^K$. Then,

$$\max_{a: r(a) < r(a^*), \pi_{\theta_1}(a) > 0} \kappa(\mathcal{A}, a) \leq 1 \tag{282}$$

is a necessary condition for ensuring the almost sure convergence of the policies obtained using $\mathcal{A}$ and online sampling to the global optimum starting from $\theta_1$.

*Proof.* It suffices to prove that if $\kappa(\mathcal{A}, a) > 1$ happens for a suboptimal action $a \in [K]$ while $\pi_{\theta_1}(a) > 0$, then if we let $\{\theta_t\}_{t \geq 1}$ be the parameter sequence obtained by using $\mathcal{A}$ with online sampling, i.e., when $a_t \sim \pi_{\theta_t}(\cdot)$, then the event $\mathcal{E} = \{a_t = a \text{ holds for all } t \geq 1\}$ happens with positive probability, and it also holds that $\pi_{\theta_t}$ converges to a sub-optimal deterministic policy with positive probability.

For convenience, denote $\alpha := \kappa(\mathcal{A}, a)$. Define the history of actions for the first $t$ iterations,

$$\mathcal{H}_t := (a_1, a_2, \cdots, a_t). \tag{283}$$

Given the historical iterations, sampled actions and rewards, the next iteration is a deterministic result of the algorithm,

$$\theta_t = \mathcal{A}\left(\theta_1, a_1, r(a_1), \theta_2, a_2, r(a_2), \cdots, \theta_{t-1}, a_{t-1}, r(a_{t-1})\right). \tag{284}$$

Let Pr denote the probability measure that over the probability space $(\Omega, \mathcal{F})$ that holds all of our random variables. By construction, Pr satisfies that for all $a$ and $t \geq 1$,

$$\Pr\left(a_t = a \mid \mathcal{H}_{t-1}\right) = \pi_{\theta_t}(a) \qquad \text{Pr almost surely.} \tag{285}$$

Define the following event, for all $t \geq 1$,

$$\mathcal{E}_t := \{a_s = a, \text{ for all } 1 \leq s \leq t\}. \tag{286}$$

We have $\mathcal{E}_t \supseteq \mathcal{E}_{t+1}$, and $\mathcal{E}_t$ approaches the limit event,

$$\mathcal{E} := \{a_t = a, \text{ for all } t \geq 1\}. \tag{287}$$

We have $\Pr\left(\mathcal{E}_t\right)$ is monotonically decreasing and lower bounded by zero. According to monotone convergence theorem,

$$\Pr\left(\mathcal{E}\right) = \lim_{t \to \infty} \Pr\left(\mathcal{E}_t\right). \tag{288}$$

Next, we prove by induction on $t$ the following holds

$$\Pr\left(\mathcal{E}_t\right) = \Pr\left(a_t = a \mid \mathcal{E}_{t-1}\right) \cdot \Pr\left(\mathcal{E}_{t-1}\right) \tag{289}$$

$$= \prod_{s=1}^{t} \pi_{\tilde{\theta}_s}(a), \tag{290}$$

where $\tilde{\theta}_1 = \theta_1$, and,

$$\tilde{\theta}_t = \mathcal{A}\big(\theta_1, \underbrace{a, r(a)}_{s=1}, \cdots, \underbrace{a, r(a)}_{s=t-1}\big), \tag{291}$$

which means $a$ is used for the first $t-1$ iterations.

First, by definition of $\tilde{\theta}_1$, we have,

$$\Pr\left(\mathcal{E}_1\right) = \pi_{\theta_1}(a) = \pi_{\tilde{\theta}_1}(a), \tag{292}$$

where the first equation is from Eq. (285). Suppose the equation holds up to $t-1$. We have,

$$\Pr\left(\mathcal{E}_t\right) = \mathbb{E}\left[\Pr\left(a_t = a, \cdots, a_1 = a \mid \mathcal{H}_{t-1}\right)\right] \qquad \text{(by the tower rule)} \tag{293}$$

$$= \mathbb{E}\left[\mathbb{I}\left\{a_{t-1} = a, \cdots, a_1 = a\right\} \cdot \Pr\left(a_t = a \mid \mathcal{H}_{t-1}\right)\right] \qquad (\{a_1, \cdots, a_{t-1}\} \text{ is deterministic given } \mathcal{H}_{t-1}) \tag{294}$$

$$= \mathbb{E}\left[\mathbb{I}\left\{a_{t-1} = a, \cdots, a_1 = a\right\} \cdot \pi_{\theta_t}(a)\right] \qquad \text{(by Eq. (285))} \tag{295}$$

$$= \mathbb{E}\left[\mathbb{I}\left\{a_{t-1} = a, \cdots, a_1 = a\right\} \cdot \pi_{\tilde{\theta}_t}(a)\right] \qquad \text{(by Eq. (291))} \tag{296}$$

$$= \pi_{\tilde{\theta}_t}(a) \cdot \Pr\left(\mathcal{E}_{t-1}\right) \qquad \text{(from calculating the expectation)} \tag{297}$$

$$= \prod_{s=1}^{t} \pi_{\tilde{\theta}_s}(a). \qquad \text{(by induction hypothesis)} \tag{298}$$

Next, we show that $\prod_{t=1}^{\infty} \pi_{\tilde{\theta}_t}(a) > 0$, where $\pi_{\tilde{\theta}_t}(a)$ is the probability at $t$th iteration given $\mathcal{A}$ is used when in the first $t-1$ iterations action $a$ is used. This is the sequence used in the definition of committal rate $\kappa$. Further, for simplicity, assume that in the definition of $\kappa$, the supremum is achieved. It follows that there exists a universal constant $C > 0$ such that on $\mathcal{E}$, for all $t \geq 1$,

$$1 - \pi_{\tilde{\theta}_t}(a) = t^{\alpha} \cdot \left[1 - \pi_{\tilde{\theta}_t}(a)\right] \cdot \frac{1}{t^{\alpha}} \tag{299}$$

$$\leq \frac{C}{t^{\alpha}}. \qquad \text{(by Definition 2)} \tag{300}$$

Let $u_t := 1 - \pi_{\tilde{\theta}_t}(a) \in (0, 1)$ for all $t \geq 1$. We have,

$$\sum_{t=1}^{\infty} u_t \leq \sum_{t=1}^{\infty} \frac{C}{t^{\alpha}} \qquad \text{(by Eq. (299))} \tag{301}$$

$$< \infty. \qquad \text{(by Lemma 15, } \alpha := \kappa(\mathcal{A}, a) > 1) \tag{302}$$

Therefore we have,

$$\prod_{t=1}^{\infty} \pi_{\tilde{\theta}_t}(a) = \prod_{t=1}^{\infty} (1 - u_t) \tag{303}$$

$$> 0. \qquad \text{(by Lemma 16 and Eq. (301))} \tag{304}$$

Hence, we have,

$$\Pr(\mathcal{E}) = \lim_{T \to \infty} \Pr(\mathcal{E}_T) \qquad \text{(by Eq. (288))} \tag{305}$$

$$= \lim_{T \to \infty} \prod_{t=1}^{T} \pi_{\tilde{\theta}_t}(a) \qquad \text{(by Eq. (293))} \tag{306}$$

$$= \prod_{t=1}^{\infty} \pi_{\tilde{\theta}_t}(a) > 0, \qquad \text{(by Eq. (303))} \tag{307}$$

and thus $\pi_{\theta_t}(a) \to 1$ as $t \to \infty$. $\qquad \square$

**Theorem 6.** Let Assumption 1 holds. For the stochastic updates NPG and GNPG from Updates 5 and 6 we obtain $\kappa(\text{NPG}, a) = \infty$ and $\kappa(\text{GNPG}, a) = \infty$ for all $a \in [K]$ respectively.

*Proof.* **First part (NPG).** We first show that $\kappa(\text{NPG}, a) = \infty$ for all $a \in [K]$. According to Definition 2, let action $a$ be sampled forever after initialization. We have, for stochastic NPG update,

$$1 - \pi_{\theta_t}(a) = \sum_{a' \neq a} \pi_{\theta_t}(a') \tag{308}$$

$$\leq \frac{\sum_{a' \neq a} \exp\{\theta_1(a')\}}{\exp\{\theta_1(a) + \eta \cdot r(a) \cdot (t-1)\} + \sum_{a' \neq a} \exp\{\theta_1(a')\}}. \qquad \text{(by Eq. (234))} \tag{309}$$

Since $\exp\{\theta_1(i)\} \in \Theta(1)$ for all $i \in [K]$, we have, for any finite $\alpha \in (0, \infty)$,

$$\lim_{t \to \infty} t^{\alpha} \cdot [1 - \pi_{\theta_t}(a)] \leq \lim_{t \to \infty} \frac{t^{\alpha} \cdot \sum_{a' \neq a} \exp\{\theta_1(a')\}}{\exp\{\theta_1(a) + \eta \cdot r(a) \cdot (t-1)\} + \sum_{a' \neq a} \exp\{\theta_1(a')\}} \qquad \text{(by Eq. (308))} \tag{310}$$

$$= \lim_{t \to \infty} \frac{\Theta(t^{\alpha})}{\Theta(\exp\{\eta \cdot r(a) \cdot (t-1)\})} = 0, \tag{311}$$

which means $\kappa(\text{NPG}, a) = \infty$ for all $a \in [K]$.

**Second part (GNPG).** We next show that $\kappa(\text{GNPG}, a) = \infty$ for all $a \in [K]$. Let action $a$ be sampled forever after initialization. We have, for stochastic GNPG update,

$$1 - \pi_{\theta_t}(a) = \sum_{a' \neq a} \pi_{\theta_t}(a') \tag{312}$$

$$\leq \frac{\sum_{a' \neq a} \exp\{\theta_1(a')\}}{\exp\left\{\theta_1(a) + \frac{\eta}{\sqrt{2}} \cdot (t-1)\right\} + \sum_{a' \neq a} \exp\{\theta_1(a')\}}. \qquad \text{(by Eq. (273))} \tag{313}$$

Using similar arguments to Eq. (310), we have $\kappa(\text{GNPG}, a) = \infty$ for all $a \in [K]$. $\qquad \square$

**Theorem 7.** Softmax PG obtains $\kappa(\text{PG}, a) = 1$ for all $a \in [K]$.

*Proof.* **First part.** $\kappa(\text{PG}, a) \geq 1$.

According to Definition 2, let action $a$ be sampled forever after initialization. We have, for stochastic PG update,

$$\left(1 - \pi_{\theta_{t+1}}(a)\right) - \left(1 - \pi_{\theta_t}(a)\right) = \pi_{\theta_t}(a) - \pi_{\theta_{t+1}}(a) + \left\langle \frac{d\pi_{\theta_t}(a)}{d\theta_t}, \theta_{t+1} - \theta_t \right\rangle - \left\langle \frac{d\pi_{\theta_t}(a)}{d\theta_t}, \theta_{t+1} - \theta_t \right\rangle \tag{314}$$

$$\leq \frac{5}{4} \cdot \|\theta_{t+1} - \theta_t\|_2^2 - \left\langle \frac{d\pi_{\theta_t}(a)}{d\theta_t}, \theta_{t+1} - \theta_t \right\rangle \qquad \text{(by Eq. (12), smoothness)} \tag{315}$$

$$= \frac{5 \cdot \eta^2}{4} \cdot \left\| \frac{d\pi_{\theta_t}^\top \hat{r}_t}{d\theta_t} \right\|_2^2 - \eta \cdot \left\langle \frac{d\pi_{\theta_t}(a)}{d\theta_t}, \frac{d\pi_{\theta_t}^\top \hat{r}_t}{d\theta_t} \right\rangle \qquad \text{(using Update 4)} \tag{316}$$

$$= \frac{5 \cdot \eta^2}{4} \cdot \left( \sum_{a' \neq a} \pi_{\theta_t}(a')^2 \cdot r(a)^2 + (1 - \pi_{\theta_t}(a))^2 \cdot r(a)^2 \right) - \eta \cdot \left\langle \frac{d\pi_{\theta_t}(a)}{d\theta_t}, \frac{d\pi_{\theta_t}^\top \hat{r}_t}{d\theta_t} \right\rangle \qquad \text{(by Eqs. (257) and (260))} \tag{317}$$

$$\leq \frac{5 \cdot \eta^2}{2} \cdot (1 - \pi_{\theta_t}(a))^2 \cdot r(a)^2 - \eta \cdot \left\langle \frac{d\pi_{\theta_t}(a)}{d\theta_t}, \frac{d\pi_{\theta_t}^\top \hat{r}_t}{d\theta_t} \right\rangle \qquad (\|x\|_2 \leq \|x\|_1) \tag{318}$$

$$= \frac{5 \cdot \eta^2}{2} \cdot (1 - \pi_{\theta_t}(a))^2 \cdot r(a)^2 - \eta \cdot \pi_{\theta_t}(a) \cdot r(a) \cdot \left( \sum_{a' \neq a} \pi_{\theta_t}(a')^2 + (1 - \pi_{\theta_t}(a))^2 \right) \qquad \text{(see below)} \tag{319}$$

$$\leq \frac{5 \cdot \eta^2}{2} \cdot (1 - \pi_{\theta_t}(a))^2 \cdot r(a)^2 - \eta \cdot \pi_{\theta_t}(a) \cdot r(a) \cdot (1 - \pi_{\theta_t}(a))^2, \tag{320}$$

where the first inequality is because $\pi_\theta(a) = \pi_\theta^\top e_a$, where $e_a \in \{0,1\}^K$ with $e_a(a) = 1$ and $e_a(a') = 0$ for all $a' \neq a$, and the second last equality is because of

$$\frac{d\pi_{\theta_t}(a)}{d\theta_t(i)} = \begin{cases} \pi_{\theta_t}(i) \cdot (1 - \pi_{\theta_t}(i)), & \text{if } i = a, \\ -\pi_{\theta_t}(i) \cdot \pi_{\theta_t}(a). & \text{otherwise} \end{cases} \tag{321}$$

Using $\eta = \frac{\pi_{\theta_t}(a)}{5 \cdot r(a)}$, for all $t \geq 1$, we have,

$$\left(1 - \pi_{\theta_{t+1}}(a)\right) - \left(1 - \pi_{\theta_t}(a)\right) \leq -\frac{1}{10} \cdot \pi_{\theta_t}(a)^2 \cdot (1 - \pi_{\theta_t}(a))^2, \tag{322}$$

which means $\pi_{\theta_{t+1}}(a) \geq \pi_{\theta_t}(a)$ for all $t \geq 1$. Therefore, we have $\eta \geq \frac{\pi_{\theta_1}(a)}{5 \cdot r(a)} \in \Theta(1)$ and,

$$\left(1 - \pi_{\theta_{t+1}}(a)\right) - \left(1 - \pi_{\theta_t}(a)\right) \leq -\frac{1}{10} \cdot \pi_{\theta_1}(a)^2 \cdot (1 - \pi_{\theta_t}(a))^2. \tag{323}$$

Then we have,

$$\frac{1}{1 - \pi_{\theta_t}(a)} = \frac{1}{1 - \pi_{\theta_1}(a)} + \sum_{s=1}^{t-1} \left[ \frac{1}{1 - \pi_{\theta_{s+1}}(a)} - \frac{1}{1 - \pi_{\theta_s}(a)} \right] \tag{324}$$

$$= \frac{1}{1 - \pi_{\theta_1}(a)} + \sum_{s=1}^{t-1} \frac{1}{\left(1 - \pi_{\theta_{s+1}}(a)\right) \cdot \left(1 - \pi_{\theta_s}(a)\right)} \cdot \left[ (1 - \pi_{\theta_s}(a)) - \left(1 - \pi_{\theta_{s+1}}(a)\right) \right] \tag{325}$$

$$\geq \frac{1}{1 - \pi_{\theta_1}(a)} + \sum_{s=1}^{t-1} \frac{1}{\left(1 - \pi_{\theta_{s+1}}(a)\right) \cdot \left(1 - \pi_{\theta_s}(a)\right)} \cdot \frac{\pi_{\theta_1}(a)^2}{10} \cdot (1 - \pi_{\theta_s}(a))^2 \qquad \text{(by Eq. (323))} \tag{326}$$

$$\geq \frac{1}{1 - \pi_{\theta_1}(a)} + \frac{\pi_{\theta_1}(a)^2}{10} \cdot (t - 1) \qquad \left( \pi_{\theta_{t+1}}(a) \geq \pi_{\theta_t}(a) \right) \tag{327}$$

$$\geq \frac{\pi_{\theta_1}(a)^2}{10} \cdot t, \qquad \left( \frac{1}{1 - \pi_{\theta_1}(a)} \geq 1 \geq \frac{\pi_{\theta_1}(a)^2}{10} \right) \tag{328}$$

which implies for all $t \geq 1$,

$$t \cdot [1 - \pi_{\theta_t}(a)] \leq t \cdot \left[ \frac{10}{\pi_{\theta_1}(a)^2} \cdot \frac{1}{t} \right] \qquad \text{(by Eq. (324))} \tag{329}$$

$$= \frac{10}{\pi_{\theta_1}(a)^2}, \tag{330}$$

which means $\kappa(\mathrm{PG}, a) \geq 1$ for all $a \in [K]$ according to Definition 2.

**Second part.** $\kappa(\mathrm{PG}, a) \leq 1$.

Let action $a$ be sampled forever after initialization. We show that $1 - \pi_{\theta_t}(a)$ cannot decrease faster than $O(1/t)$. Similar to Eq. (314), we have,

$$\left(1 - \pi_{\theta_t}(a)\right) - \left(1 - \pi_{\theta_{t+1}}(a)\right) = \pi_{\theta_{t+1}}(a) - \pi_{\theta_t}(a) - \left\langle \frac{d\pi_{\theta_t}(a)}{d\theta_t}, \theta_{t+1} - \theta_t \right\rangle + \left\langle \frac{d\pi_{\theta_t}(a)}{d\theta_t}, \theta_{t+1} - \theta_t \right\rangle \tag{331}$$

$$\leq \frac{5}{4} \cdot \|\theta_{t+1} - \theta_t\|_2^2 + \left\langle \frac{d\pi_{\theta_t}(a)}{d\theta_t}, \theta_{t+1} - \theta_t \right\rangle \qquad \text{(by Eq. (12), smoothness)} \tag{332}$$

$$= \frac{5 \cdot \eta^2}{4} \cdot \left\| \frac{d\pi_{\theta_t}^\top \hat{r}_t}{d\theta_t} \right\|_2^2 + \eta \cdot \left\langle \frac{d\pi_{\theta_t}(a)}{d\theta_t}, \frac{d\pi_{\theta_t}^\top \hat{r}_t}{d\theta_t} \right\rangle \qquad \text{(using Update 4)} \tag{333}$$

$$\leq \frac{5 \cdot \eta^2}{2} \cdot (1 - \pi_{\theta_t}(a))^2 \cdot r(a)^2 + \eta \cdot \left\langle \frac{d\pi_{\theta_t}(a)}{d\theta_t}, \frac{d\pi_{\theta_t}^\top \hat{r}_t}{d\theta_t} \right\rangle \qquad \text{(by Eqs. (257) and (260))} \tag{334}$$

$$= \frac{5 \cdot \eta^2}{2} \cdot (1 - \pi_{\theta_t}(a))^2 \cdot r(a)^2 + \eta \cdot \pi_{\theta_t}(a) \cdot r(a) \cdot \left( \sum_{a' \neq a} \pi_{\theta_t}(a')^2 + (1 - \pi_{\theta_t}(a))^2 \right) \tag{335}$$

$$\leq \frac{5 \cdot \eta^2}{2} \cdot (1 - \pi_{\theta_t}(a))^2 \cdot r(a)^2 + 2 \cdot \eta \cdot \pi_{\theta_t}(a) \cdot r(a) \cdot (1 - \pi_{\theta_t}(a))^2 \qquad (\|x\|_2 \leq \|x\|_1) \tag{336}$$

$$\leq \frac{9}{2} \cdot (1 - \pi_{\theta_t}(a))^2 \cdot r(a), \tag{337}$$

where the last inequality is due to $\pi_{\theta_t}(a) \in (0, 1)$, $r(a) \in (0, 1]$, and $\eta \in (0, 1]$. Denote $\delta(\theta_t) := 1 - \pi_{\theta_t}(a)$. We have, for all $t \geq 1$,

$$\delta(\theta_t) - \delta(\theta_{t+1}) \leq \frac{9}{2} \cdot r(a) \cdot \delta(\theta_t)^2, \tag{338}$$

which is similar to Eq. (33). Therefore, using similar calculations in the proofs for Proposition 2, we have, for all large enough $t \geq 1$,

$$t \cdot [1 - \pi_{\theta_t}(a)] \geq t \cdot \left[ \frac{1}{6 \cdot r(a)} \cdot \frac{1}{t} \right] \tag{339}$$

$$= \frac{1}{6 \cdot r(a)}, \tag{340}$$

which means $\kappa(\mathrm{PG}, a) \leq 1$ for all $a \in [K]$ according to Definition 2. $\qquad \square$

**Theorem 8.** Using Update 7, $(\pi^* - \pi_{\theta_t})^\top r \to 0$ as $t \to \infty$ with probability 1.

*Proof.* Consider the sequence $\{\pi_{\theta_t}(a^*)\}_{t \geq 1}$ produced by Update 7 using on-policy sampling $a_t \sim \pi_{\theta_t}(\cdot)$. We show that $\pi_{\theta_t}(a^*) \to 1$ as $t \to \infty$ with probability 1.

First, for convenience, we duplicate Update 7 here.

**Update 7** (NPG with oracle baseline). $\theta_{t+1} \leftarrow \theta_t + \eta \cdot \left( \hat{r}_t - \hat{b}_t \right)$, where $\hat{b}_t(a) = \left( \frac{\mathbb{I}\{a_t = a\}}{\pi_{\theta_t}(a)} - 1 \right) \cdot b$ for all $a \in [K]$, and $b \in (r(a^*) - \Delta, r(a^*))$.

Note that Update 7 is equivalent to the following update,

$$\theta_{t+1}(a) = \begin{cases} \theta_t(a) + \frac{\eta}{\pi_{\theta_t}(a)} \cdot (r(a) - b), & \text{if } a = a_t, \\ \theta_t(a), & \text{otherwise} \end{cases} \tag{341}$$

Next, we show that $\pi_{\theta_{t+1}}(a^*) \geq \pi_{\theta_t}(a^*)$ using on-policy sampling $a_t \sim \pi_{\theta_t}(\cdot)$. There are two cases.

Case (a): If $a_t = a^*$, then we have,

$$\theta_{t+1}(a^*) = \theta_t(a^*) + \frac{\eta}{\pi_{\theta_t}(a^*)} \cdot (r(a^*) - b) \qquad \text{(by Eq. (341))} \tag{342}$$

$$> \theta_t(a^*), \qquad (r(a^*) > b) \tag{343}$$

while $\theta_{t+1}(a) = \theta_t(a)$ for all sub-optimal actions $a \neq a^*$. Then we have,

$$\pi_{\theta_{t+1}}(a^*) = \frac{\exp\{\theta_{t+1}(a^*)\}}{\exp\{\theta_{t+1}(a^*)\} + \sum_{a \neq a^*} \exp\{\theta_{t+1}(a)\}} \tag{344}$$

$$> \frac{\exp\{\theta_t(a^*)\}}{\exp\{\theta_t(a^*)\} + \sum_{a \neq a^*} \exp\{\theta_{t+1}(a)\}} \qquad \text{(by Eq. (342))} \tag{345}$$

$$= \frac{\exp\{\theta_t(a^*)\}}{\exp\{\theta_t(a^*)\} + \sum_{a \neq a^*} \exp\{\theta_t(a)\}} \qquad (\theta_{t+1}(a) = \theta_t(a), \text{ for all } a \neq a^*) \tag{346}$$

$$= \pi_{\theta_t}(a^*). \tag{347}$$

Case (b): If $a_t = a \neq a^*$, then we have,

$$\theta_{t+1}(a) = \theta_t(a) + \frac{\eta}{\pi_{\theta_t}(a)} \cdot (r(a) - b) \qquad \text{(by Eq. (341))} \tag{348}$$

$$< \theta_t(a), \qquad (r(a) \leq r(a^*) - \Delta < b) \tag{349}$$

where $\Delta = r(a^*) - \max_{a \neq a^*} r(a) > 0$ is the reward gap. Also $\theta_{t+1}(a') = \theta_t(a')$ for all the other actions $a' \neq a$. Then we have,

$$\pi_{\theta_{t+1}}(a^*) = \frac{\exp\{\theta_{t+1}(a^*)\}}{\exp\{\theta_{t+1}(a)\} + \sum_{a' \neq a} \exp\{\theta_{t+1}(a')\}} \tag{350}$$

$$> \frac{\exp\{\theta_{t+1}(a^*)\}}{\exp\{\theta_t(a)\} + \sum_{a' \neq a} \exp\{\theta_{t+1}(a')\}} \qquad \text{(by Eq. (348))} \tag{351}$$

$$= \frac{\exp\{\theta_t(a^*)\}}{\exp\{\theta_t(a)\} + \sum_{a' \neq a} \exp\{\theta_t(a')\}} \qquad (\theta_{t+1}(a') = \theta_t(a'), \text{ for all } a' \neq a) \tag{352}$$

$$= \pi_{\theta_t}(a^*). \tag{353}$$

Therefore, we have $\pi_{\theta_{t+1}}(a^*) \geq \pi_{\theta_t}(a^*)$, for all $t \geq 1$. Note that $\pi_{\theta_t}(a^*) \leq 1$. According to monotone convergence theorem, we have $\pi_{\theta_{t+1}}(a^*)$ approaches to some finite value as $t \to \infty$.

Suppose $\pi_{\theta_t}(a^*) \to \pi_{\theta_\infty}(a^*)$ as $t \to \infty$. We show that $\pi_{\theta_\infty}(a^*) = 1$ by contradiction. Suppose $\pi_{\theta_\infty}(a^*) < 1$. Then at the convergent point, according to Eqs. (344) and (350), we can further improve the probability of $a^*$ by online sampling and updating once, which is a contradiction with convergence.

Thus we have $\pi_{\theta_t}(a^*) \to 1$ as $t \to \infty$ with probability 1, which implies that $(\pi^* - \pi_{\theta_t})^\top r \to 0$ as $t \to \infty$ with probability 1. $\qquad \square$

The Stochastic Approximation Markov Bandit Algorithm (SAMBA) [13] algorithm is mentioned in Section 4 and Figure 1.

**Update 8** (SAMBA). *At iteration $t \geq 1$, denote the greedy action $\bar{a}_t := \arg\max_{a \in [K]} \pi_t(a)$. Sample action $a_t \sim \pi_t(\cdot)$. (i) If $a_t = \bar{a}_t$, then perform update $\pi_{t+1}(a') \leftarrow \pi_t(a') - \eta \cdot \pi_t(a')^2 \cdot \frac{r(a_t)}{\pi_t(a_t)}$ for all non-greedy action $a' \neq a_t$; (ii) If $a_t \neq \bar{a}_t$, then perform update $\pi_{t+1}(a_t) \leftarrow \pi_t(a_t) + \eta \cdot \pi_t(a_t)^2 \cdot \frac{r(a_t)}{\pi_t(a_t)}$. After doing (i) or (ii), calculate $\pi_{t+1}(\bar{a}_t) = 1 - \sum_{a' \neq \bar{a}_t} \pi_{t+1}(a')$.*

The SAMBA algorithm does not maintain parameters $\theta$, and the last step $\pi_{t+1}(\bar{a}_t) = 1 - \sum_{a' \neq \bar{a}_t} \pi_{t+1}(a')$ in Update 8 is a necessary projection to the probability simplex, such that $\pi_t$ is a valid probability distribution over $[K]$. As shown in [13], if the learning rate has the knowledge of the optimal action's reward and reward gap, i.e.,

$$\eta < \frac{\Delta}{r(a^*) - \Delta}, \tag{354}$$

then Update 8 converges to $\pi^*$ almost surely with a $O(1/t)$ rate, i.e.,

$$(\pi^* - \pi_t)^\top r \leq C/t. \tag{355}$$

We calculate the committal rate of SAMBA.

**Proposition 4.** *For SAMBA from Update 8, we have $\kappa(SAMBA, a) = 1$ for all $a \in [K]$.*

*Proof.* **First part.** $\kappa(SAMBA, a) \geq 1$.

According to Definition 2, let action $a$ be the greedy action and be sampled forever. According to *(i)* in Update 8, we have, for all $a' \neq a$,

$$\pi_{t+1}(a') = \pi_t(a') - \eta \cdot \pi_t(a')^2 \cdot \frac{r(a_t)}{\pi_t(a_t)} \tag{356}$$

$$= \pi_t(a') - \eta \cdot \pi_t(a')^2 \cdot \frac{r(a)}{\pi_t(a)} \qquad (a_t = a \text{ by fixed sampling}) \tag{357}$$

$$\leq \pi_t(a') - \eta \cdot \pi_t(a')^2 \cdot r(a). \qquad (\pi_t(a) \in (0, 1)) \tag{358}$$

Using similar calculations in Eq. (13), we have, for all $a' \neq a$,

$$\frac{1}{\pi_t(a')} = \frac{1}{\pi_1(a')} + \sum_{s=1}^{t-1} \left[ \frac{1}{\pi_{s+1}(a')} - \frac{1}{\pi_s(a')} \right] \tag{359}$$

$$= \frac{1}{\pi_1(a')} + \sum_{s=1}^{t-1} \frac{1}{\pi_{s+1}(a') \cdot \pi_s(a')} \cdot (\pi_s(a')) - \pi_{s+1}(a')) \tag{360}$$

$$\geq \frac{1}{\pi_1(a')} + \sum_{s=1}^{t-1} \frac{1}{\pi_{s+1}(a') \cdot \pi_s(a')} \cdot \eta \cdot \pi_s(a')^2 \cdot r(a) \qquad (\text{by Eq. (356)}) \tag{361}$$

$$\geq \frac{1}{\pi_1(a')} + \eta \cdot r(a) \cdot (t-1) \qquad (\pi_{t+1}(a') \leq \pi_t(a'), \text{ by Eq. (356)}) \tag{362}$$

$$\geq \eta \cdot r(a) \cdot t, \qquad \left( \frac{1}{\pi_1(a')} \geq 1 \geq \eta \cdot r(a) \right) \tag{363}$$

which implies, for all large enough $t \geq 1$,

$$t \cdot [1 - \pi_t(a)] = t \cdot \sum_{a' \neq a} \pi_t(a') \tag{364}$$

$$\leq t \cdot \sum_{a' \neq a} \frac{1}{\eta \cdot r(a) \cdot t} \qquad (\text{by Eq. (359)}) \tag{365}$$

$$= \sum_{a' \neq a} \frac{1}{\eta \cdot r(a)}, \tag{366}$$

which means $\kappa(SAMBA, a) \geq 1$ for all $a \in [K]$ according to Definition 2.

**Second part.** $\kappa(SAMBA, a) \leq 1$.

Let action $a$ be the greedy action and be sampled forever. According to *(i)* in Update 8, we have, for all $a' \neq a$,

$$\pi_{t+1}(a') = \pi_t(a') - \eta \cdot \pi_t(a')^2 \cdot \frac{r(a_t)}{\pi_t(a_t)} \tag{367}$$

$$= \pi_t(a') - \eta \cdot \pi_t(a')^2 \cdot \frac{r(a)}{\pi_t(a)} \qquad (a_t = a \text{ by fixed sampling}) \tag{368}$$

$$\geq \pi_t(a') - \eta \cdot K \cdot \pi_t(a')^2 \cdot r(a), \qquad (\pi_t(a) \geq 1/K, \ a \text{ is greedy action}) \tag{369}$$

which is similar to Eq. (33). Therefore, using similar calculations in the proofs for Proposition 2, we have, for all large enough $t \geq 1$, we have,

$$\frac{\pi_{t+1}(a')}{\pi_t(a')} \geq \frac{1}{2}. \tag{370}$$

Denote

$$t_0 := \min\left\{t \geq 1 : \frac{\pi_{t+1}(a')}{\pi_t(a')} \geq \frac{1}{2}, \text{ for all } s \geq t\right\}. \tag{371}$$

On the other hand, since $t_0 \in O(1)$, we have, for all $t < t_0$,

$$\pi_{t+1}(a') \geq c_0 > 0. \tag{372}$$

Next, we have, for all $t \geq t_0$,

$$\frac{1}{\pi_t(a')} = \frac{1}{\pi_1(a')} + \sum_{s=1}^{t_0-1} \frac{1}{\pi_{s+1}(a')} \cdot \left(1 - \frac{\pi_{s+1}(a')}{\pi_s(a')}\right) + \sum_{s=t_0}^{t-1} \frac{1}{\pi_{s+1}(a') \cdot \pi_s(a')} \cdot (\pi_s(a')) - \pi_{s+1}(a')) \tag{373}$$

$$\leq \frac{1}{c_0} + \sum_{s=1}^{t_0-1} \frac{1}{c_0} \cdot 1 + \sum_{s=t_0}^{t-1} \frac{1}{\pi_{s+1}(a') \cdot \pi_s(a')} \cdot \eta \cdot K \cdot \pi_s(a')^2 \cdot r(a) \qquad \text{(by Eqs. (367) and (372))} \tag{374}$$

$$\leq \frac{t_0}{c_0} + 2 \cdot \eta \cdot K \cdot r(a) \cdot (t - t_0), \qquad \text{(by Eq. (370))} \tag{375}$$

which implies, for all large enough $t \geq 1$,

$$t \cdot [1 - \pi_t(a)] = t \cdot \sum_{a' \neq a} \pi_t(a') \tag{376}$$

$$\geq t \cdot \sum_{a' \neq a} \frac{1}{t_0/c_0 + 2 \cdot \eta \cdot K \cdot r(a) \cdot (t - t_0)} \qquad \text{(by Eq. (373))} \tag{377}$$

$$\geq \sum_{a' \neq a} \frac{1}{3 \cdot \eta \cdot K \cdot r(a)}, \qquad (t_0/c_0 \leq \eta \cdot K \cdot r(a) \cdot t) \tag{378}$$

which means $\kappa(\text{SAMBA}, a) \leq 1$ for all $a \in [K]$ according to Definition 2. $\qquad\square$

## C  Proofs for Geometry-Convergence Trade-off (Section 4)

First, we show that the algorithms we study in this paper, i.e., softmax PG, NPG, and GNPG, are optimality-smart. Recall from the main paper that, a policy optimization method is said to be *optimality-smart* if for any $t \geq 1$, $\pi_{\tilde{\theta}_t}(a^*) \geq \pi_{\theta_t}(a^*)$ holds where $\tilde{\theta}_t$ is the parameter vector obtained when $a^*$ is chosen in every time step, starting at $\theta_1$, while $\theta_t$ is *any* parameter vector that can be obtained with $t$ updates (regardless of the action sequence chosen), but also starting from $\theta_1$.

**Proposition 5.** *Softmax PG, NPG, and GNPG are optimality-smart.*

*Proof.* We show that for softmax PG, NPG, and GNPG, if $a_t = a^*$, then $\pi_{\theta_{t+1}}(a^*) \geq \pi_{\theta_t}(a^*)$; if $a_t = a \neq a^*$, then $\pi_{\theta_{t+1}}(a^*) \leq \pi_{\theta_t}(a^*)$. For softmax PG and GNPG the later claim holds when $a^*$ is the dominating action at $t$th iteration, i.e., $\pi_{\theta_t}(a^*) \geq \pi_{\theta_t}(a')$ for all $a' \neq a^*$. From existing results (Propositions 1 and 3) we know that softmax PG and GNPG converge to $\pi^*$ as $t \to \infty$ (using true policy gradients; also holds for using fixed sampling $a_t = a^*$ for all $t \geq 1$), thus we have for all large enough $t \geq 1$, $\pi_{\theta_t}(a^*) \geq \pi_{\theta_t}(a')$ for all $a' \neq a^*$.

**First part.** Softmax PG and GNPG are optimality-smart.

If $a_t = a^*$, then we have,

$$\theta_{t+1}(a^*) = \theta_t(a^*) + \eta \cdot \frac{d\pi_{\theta_t}^\top \hat{r}_t}{d\theta_t(a^*)} \tag{379}$$

$$= \theta_t(a^*) + \eta \cdot (1 - \pi_{\theta_t}(a^*)) \cdot r(a^*) \qquad \text{(by Eq. (257))} \tag{380}$$

$$\geq \theta_t(a^*). \qquad \left(r \in (0, 1]^K\right) \tag{381}$$

And for any $a \neq a^*$, we have,

$$\theta_{t+1}(a) = \theta_t(a) + \eta \cdot \frac{d\pi_{\theta_t}^\top \hat{r}_t}{d\theta_t(a)} \tag{382}$$

$$= \theta_t(a) - \eta \cdot \pi_{\theta_t}(a) \cdot r(a^*) \qquad \text{(by Eq. (260))} \tag{383}$$

$$\leq \theta_t(a). \qquad \left(r \in (0,1]^K\right) \tag{384}$$

Therefore, we have,

$$\pi_{\theta_{t+1}}(a^*) = \frac{\exp\{\theta_{t+1}(a^*)\}}{\exp\{\theta_{t+1}(a^*)\} + \sum_{a \neq a^*} \exp\{\theta_{t+1}(a)\}} \tag{385}$$

$$\geq \frac{\exp\{\theta_t(a^*)\}}{\exp\{\theta_t(a^*)\} + \sum_{a \neq a^*} \exp\{\theta_t(a)\}} \qquad \text{(by Eqs. (379) and (382))} \tag{386}$$

$$= \pi_{\theta_t}(a^*). \tag{387}$$

On the other hand, given $a_t = a \neq a^*$, we show that if $\pi_{\theta_t}(a^*) \geq \pi_{\theta_t}(a')$ for all $a' \neq a^*$, then $\pi_{\theta_{t+1}}(a^*) \leq \pi_{\theta_t}(a^*)$. We have,

$$\theta_{t+1}(a) = \theta_t(a) + \eta \cdot (1 - \pi_{\theta_t}(a)) \cdot r(a) \qquad \text{(by Eq. (257))} \tag{388}$$

$$\geq \theta_t(a) - \eta \cdot \pi_{\theta_t}(a^*) \cdot r(a). \tag{389}$$

And for any $a' \neq a$, we have,

$$\theta_{t+1}(a') = \theta_t(a') - \eta \cdot \pi_{\theta_t}(a') \cdot r(a) \qquad \text{(by Eq. (260))} \tag{390}$$

$$\geq \theta_t(a') - \eta \cdot \pi_{\theta_t}(a^*) \cdot r(a). \qquad (\pi_{\theta_t}(a^*) \geq \pi_{\theta_t}(a')) \tag{391}$$

Therefore, we have,

$$\pi_{\theta_{t+1}}(a^*) = \frac{\exp\{\theta_{t+1}(a^*)\}}{\exp\{\theta_{t+1}(a)\} + \sum_{a' \neq a} \exp\{\theta_{t+1}(a')\}} \tag{392}$$

$$\leq \frac{\exp\{\theta_t(a^*) - \eta \cdot \pi_{\theta_t}(a^*) \cdot r(a)\}}{\exp\{\theta_t(a) - \eta \cdot \pi_{\theta_t}(a^*) \cdot r(a)\} + \sum_{a' \neq a} \exp\{\theta_t(a') - \eta \cdot \pi_{\theta_t}(a^*) \cdot r(a)\}} \qquad \text{(by Eqs. (388) and (390))} \tag{393}$$

$$= \frac{\exp\{\theta_t(a^*)\}}{\exp\{\theta_t(a)\} + \sum_{a' \neq a} \exp\{\theta_t(a')\}} \tag{394}$$

$$= \pi_{\theta_t}(a^*). \tag{395}$$

**Second part.** NPG is optimality-smart.

If $a_t = a^*$, then we have,

$$\theta_{t+1}(a^*) = \theta_t(a^*) + \eta \cdot \frac{r(a^*)}{\pi_{\theta_t}(a^*)} \tag{396}$$

$$> \theta_t(a^*). \tag{397}$$

while $\theta_{t+1}(a) = \theta_t(a)$ for all sub-optimal actions $a \neq a^*$. Then we have,

$$\pi_{\theta_{t+1}}(a^*) = \frac{\exp\{\theta_{t+1}(a^*)\}}{\exp\{\theta_{t+1}(a^*)\} + \sum_{a \neq a^*} \exp\{\theta_{t+1}(a)\}} \tag{398}$$

$$\geq \frac{\exp\{\theta_t(a^*)\}}{\exp\{\theta_t(a^*)\} + \sum_{a \neq a^*} \exp\{\theta_t(a)\}} \tag{399}$$

$$= \pi_{\theta_t}(a^*). \tag{400}$$

If $a_t = a \neq a^*$, then we have,

$$\theta_{t+1}(a) = \theta_t(a) + \eta \cdot \frac{r(a)}{\pi_{\theta_t}(a)} \tag{401}$$

$$\geq \theta_t(a), \tag{402}$$

while $\theta_{t+1}(a') = \theta_t(a')$ for all the other actions $a' \neq a$. Then we have,

$$\pi_{\theta_{t+1}}(a^*) = \frac{\exp\{\theta_{t+1}(a^*)\}}{\exp\{\theta_{t+1}(a)\} + \sum_{a' \neq a} \exp\{\theta_{t+1}(a')\}} \tag{403}$$

$$\leq \frac{\exp\{\theta_t(a^*)\}}{\exp\{\theta_t(a)\} + \sum_{a' \neq a} \exp\{\theta_t(a')\}} \tag{404}$$

$$= \pi_{\theta_t}(a^*). \qquad \square$$

**Theorem 9.** Let $\mathcal{A}$ be optimality-smart and pick a bandit instance. If $\mathcal{A}$ together with on-policy sampling leads to $\{\theta_t\}_{t \geq 1}$ such that $\{\pi_{\theta_t}\}_{t \geq 1}$ converges to a globally optimal policy at a rate $O(1/t^\alpha)$ with positive probability, for $\alpha > 0$, then $\kappa(\mathcal{A}, a^*) \geq \alpha$.

*Proof.* Fix an instance $r \in (0, 1]^K$ with a unique optimal action $a^*$. For any $\theta \in \mathbb{R}^K$, we have,

$$(\pi^* - \pi_\theta)^\top r = \sum_{a \neq a^*} \pi_\theta(a) \cdot (r(a^*) - r(a)) \tag{405}$$

$$\geq (1 - \pi_\theta(a^*)) \cdot \Delta, \tag{406}$$

where $\Delta = r(a^*) - \max_{a \neq a^*} r(a) > 0$ is the reward gap. Let $\{\theta_t\}_{t \geq 1}$ be the sequence obtained by using $\mathcal{A}$ together with online sampling on $r$. For $\alpha > 0$ let $\mathcal{E}_\alpha$ be the event when for all $t \geq 1$,

$$(\pi^* - \pi_{\theta_t})^\top r \leq \frac{C}{t^\alpha}, \tag{407}$$

By our assumption, there exists $\alpha > 0$ such that $\Pr(\mathcal{E}_\alpha) > 0$. On this event, for any $t \geq 1$,

$$t^\alpha \cdot (1 - \pi_{\theta_t}(a^*)) \leq \frac{1}{\Delta} \cdot t^\alpha \cdot (\pi^* - \pi_{\theta_t})^\top r \qquad \text{(by Eq. (405))} \tag{408}$$

$$\leq \frac{C}{\Delta}. \qquad \text{(by Eq. (407))} \tag{409}$$

Let $\{\tilde{\theta}_t\}_{t \geq 1}$ with $\tilde{\theta}_1 = \theta_1$ be the sequence obtained by using $\mathcal{A}$ with fixed sampling on $r$, such that $a_t = a^*$ for all $t \geq 1$. Since, by the assumption, $\mathcal{A}$ is optimality-smart, we have $\pi_{\tilde{\theta}_t}(a^*) \geq \pi_{\theta_t}(a^*)$. Then, on $\mathcal{E}_\alpha$, for any $t \geq 1$

$$t^\alpha \cdot (1 - \pi_{\tilde{\theta}_t}(a^*)) \leq t^\alpha \cdot (1 - \pi_{\theta_t}(a^*)) \tag{410}$$

$$\leq \frac{C}{\Delta}, \qquad \text{(by Eq. (408))} . \tag{411}$$

Since $\mathbb{P}(\mathcal{E}_\alpha) > 0$ and $t^\alpha \cdot (1 - \pi_{\tilde{\theta}_t}(a^*))$ is non-random, it follows that for any $t \geq 1$, $t^\alpha \cdot (1 - \pi_{\tilde{\theta}_t}(a^*)) \leq C/\Delta$, which, by Definition 2, means that $\kappa(\mathcal{A}, a^*) \geq \alpha$. $\square$

**Theorem 10** (Geometry-Convergence trade-off). If an algorithm $\mathcal{A}$ is optimality-smart, and $\kappa(\mathcal{A}, a^*) = \kappa(\mathcal{A}, a)$ for at least one $a \neq a^*$, then $\mathcal{A}$ with on-policy sampling can only exhibit at most one of the following two behaviors: **(i)** $\mathcal{A}$ converges to a globally optimal policy almost surely; **(ii)** $\mathcal{A}$ converges to a deterministic policy at a rate faster than $O(1/t)$ with positive probability.

*Proof.* We prove that $\mathcal{A}$ cannot achieve both of the two behaviors at the same time by contradiction. Suppose an algorithm $\mathcal{A}$ can **(i)** converge to a globally optimal policy almost surely; and **(ii)** converges at a rate $O(1/t^\alpha)$ with positive probability, where $\alpha > 1$.

Since **(ii)** holds, according to Theorem 9, we have $\kappa(\mathcal{A}, a^*) \geq \alpha > 1$. By condition, there exists at least one sub-optimal action $a \neq a^*$, such that $\kappa(\mathcal{A}, a) = \kappa(\mathcal{A}, a^*) > 1$. According to Theorem 5, we have $\pi_{\theta_t}(a) \to 1$ as $t \to \infty$ with positive probability, which contradicts **(i)**. Therefore, **(i)** and **(ii)** cannot hold simultaneously. $\square$

# D Proofs for Ensemble Methods (Section 5)

**Theorem 11.** With probability $1 - \delta$, the best single run among $O(\log{(1/\delta)})$ independent runs of NPG (GNPG) converges to a globally optimal policy at an $O(e^{-c \cdot t})$ rate.

*Proof.* According to Theorem 3, stochastic NPG of Update 5 will sample the optimal action $a^*$ forever (thus converge to the optimal policy) with probability at least

$$p(\text{NPG}, a^*) := \exp\left\{ -\frac{\exp\{\eta \cdot r(a^*)\}}{\eta \cdot r(a^*)} \cdot \frac{\sum_{a \neq a^*} \exp\{\theta_1(a)\}}{\exp\{\theta_1(a^*)\}} \right\} \qquad \text{(by Eq. (242))} \qquad (412)$$

$$\in \Omega(1). \tag{413}$$

Moreover, with probability at least $p(\text{NPG}, a^*)$, the convergence rate is,

$$(\pi^* - \pi_{\theta_t})^\top r = \sum_{a \neq a^*} \pi_{\theta_t}(a) \cdot (r(a^*) - r(a)) \tag{414}$$

$$\leq 1 - \pi_{\theta_t}(a^*) \qquad \left( r \in (0, 1]^K \right) \tag{415}$$

$$\leq \frac{\sum_{a \neq a^*} \exp\{\theta_1(a)\}}{\exp\{\theta_1(a^*) + \eta \cdot r(a^*) \cdot (t - 1)\} + \sum_{a \neq a^*} \exp\{\theta_1(a)\}} \qquad \text{(by Eq. (308))} \qquad (416)$$

$$\in O(e^{-c \cdot t}). \tag{417}$$

Consider $n(\text{NPG}) \in O(\log{(1/\delta)})$ independent runs of NPG, where

$$n(\text{NPG}) := \frac{1}{\log\left( \frac{1}{1 - p(\text{NPG}, a^*)} \right)} \cdot \log{(1/\delta)}. \tag{418}$$

The probability that all the $n(\text{NPG})$ runs do not converge to global optimal policy is at most

$$[1 - p(\text{NPG}, a^*)]^{n(\text{NPG})} = \left[ \exp\left\{ \log{\left( 1 - p(\text{NPG}, a^*) \right)} \right\} \right]^{n(\text{NPG})} \tag{419}$$

$$= \exp\left\{ -\log\left( \frac{1}{1 - p(\text{NPG}, a^*)} \right) \cdot \frac{1}{\log\left( \frac{1}{1 - p(\text{NPG}, a^*)} \right)} \cdot \log{(1/\delta)} \right\} \qquad \text{(by Eq. (418))} \tag{420}$$

$$= e^{-\log{(1/\delta)}} = \delta, \tag{421}$$

which means with probability at least $1 - \delta$, the best single run converges to a globally optimal policy at an $O(e^{-c \cdot t})$ rate.

For stochastic GNPG of Update 6, similar calculations show that with probability at least $1 - \delta$, the best single run among $n(\text{GNPG}) \in O(\log{(1/\delta)})$ independent runs of GNPG converges to a globally optimal policy at an $O(e^{-c \cdot t})$ rate, where

$$n(\text{GNPG}) := \frac{1}{\log\left( \frac{1}{1 - p(\text{GNPG}, a^*)} \right)} \cdot \log{(1/\delta)}, \tag{422}$$

and

$$p(\text{GNPG}, a^*) := \exp\left\{ -\frac{\sum_{a \neq a^*} \exp\{\theta_1(a)\}}{\exp\{\theta_1(a^*)\}} \cdot \frac{\sqrt{2} \cdot \exp\left\{ \frac{\eta}{\sqrt{2}} \right\}}{\eta} \right\} \qquad \text{(by Eq. (277))} \qquad (423)$$

$$\in \Omega(1), \tag{424}$$

thus finishing the proof. $\qquad \square$

# E General MDPs

This section is devoted to results of general MDPs. **(i)** For convergence rate results in true gradient settings, we review relevant results in literature [2, 8, 9] without detailed proofs (except for the NPG method, for which we have a new analysis using the natural NŁ inequality), since the conclusions are similar to one-state MDPs in Section 2. **(ii)** For results in on-policy stochastic settings, we discuss the main ideas for the similar conclusions as in Section 2. **(iii)** For results of the committal rate and the trade-off, we provide some calculations.

## E.1 RL Settings and Notations

Given a finite set $\mathcal{X}$, we denote $\Delta(\mathcal{X})$ as the set of all probability distributions on $\mathcal{X}$. A finite MDP $\mathcal{M} \coloneqq (\mathcal{S}, \mathcal{A}, \mathcal{P}, r, \gamma)$ is determined by the finite state space $\mathcal{S}$, action space $\mathcal{A}$, transition function $\mathcal{P} : \mathcal{S} \times \mathcal{A} \rightarrow \Delta(\mathcal{S})$, reward function $r : \mathcal{S} \times \mathcal{A} \rightarrow \mathbb{R}$, and discount factor $\gamma \in [0, 1)$.

An agent maintains a policy $\pi : \mathcal{S} \rightarrow \Delta(\mathcal{A})$. At time $t$, the agent is given a state $s_t$, and it takes an action $a_t \sim \pi(\cdot|s_t)$, receives a scalar reward $r(s_t, a_t)$ and a next-state $s_{t+1} \sim \mathcal{P}(\cdot|s_t, a_t)$. The value function of $\pi$ under $s$ is defined as

$$V^\pi(s) \coloneqq \mathop{\mathbb{E}}_{\substack{s_0 = s, a_t \sim \pi(\cdot|s_t), \\ s_{t+1} \sim \mathcal{P}(\cdot|s_t, a_t)}} \left[ \sum_{t=0}^\infty \gamma^t r(s_t, a_t) \right]. \tag{425}$$

The state-action value of $\pi$ at $(s, a) \in \mathcal{S} \times \mathcal{A}$ is defined as

$$Q^\pi(s, a) \coloneqq r(s, a) + \gamma \cdot \sum_{s'} \mathcal{P}(s'|s, a) \cdot V^\pi(s'). \tag{426}$$

The advantage function of $\pi$ is defined as

$$A^\pi(s, a) \coloneqq Q^\pi(s, a) - V^\pi(s). \tag{427}$$

The state distribution of $\pi$ is defined as,

$$d_{s_0}^\pi(s) \coloneqq (1 - \gamma) \cdot \sum_{t=0}^\infty \gamma^t \cdot \Pr(s_t = s|s_0, \pi, \mathcal{P}). \tag{428}$$

We also denote $V^\pi(\rho) \coloneqq \mathbb{E}_{s \sim \rho(\cdot)}[V^\pi(s)]$ and $d_\rho^\pi(s) \coloneqq \mathbb{E}_{s_0 \sim \rho(\cdot)}[d_{s_0}^\pi(s)]$, where $\rho \in \Delta(\mathcal{S})$ is an initial state distribution. The optimal policy $\pi^*$ satisfies $V^{\pi^*}(\rho) = \sup_{\pi : \mathcal{S} \rightarrow \Delta(\mathcal{A})} V^\pi(\rho)$. For conciseness, we denote $V^* \coloneqq V^{\pi^*}$. Given tabular parameters $\theta(s, a) \in \mathbb{R}$ for all $(s, a)$, the policy $\pi_\theta$ can be parameterized by $\theta$ as $\pi_\theta(\cdot|s) = \text{softmax}(\theta(s, \cdot))$:

$$\pi_\theta(a|s) = \frac{\exp\{\theta(s, a)\}}{\sum_{a' \in \mathcal{A}} \exp\{\theta(s, a')\}}, \text{ for all } (s, a) \in \mathcal{S} \times \mathcal{A}. \tag{429}$$

Without loss of generality, we assume $r(s, a) \in (0, 1)$ for all $(s, a) \in \mathcal{S} \times \mathcal{A}$. Generalizing expected reward maximization of Eq. (1), the problem here is then to maximize the value function, i.e.,

$$\sup_{\theta : \mathcal{S} \times \mathcal{A} \rightarrow \mathbb{R}} V^{\pi_\theta}(\rho). \tag{430}$$

Now we consider the above three algorithms and their results in general MDPs.

## E.2 True Gradient Settings

### E.2.1 Softmax PG

First, softmax PG has $\Theta(1/t)$ global convergence rates with the following assumption.

**Assumption 2** (Sufficient exploration). *The initial state distribution satisfies $\min_s \mu(s) > 0$.*

---
**Algorithm 1** Softmax PG, true gradient

    **Input:** Learning rate $\eta > 0$.
    **Output:** Policies $\pi_{\theta_t} = \text{softmax}(\theta_t)$.
    Initialize parameter $\theta_1(s, a)$ for all $(s, a) \in \mathcal{S} \times \mathcal{A}$.
    **while** $t \geq 1$ **do**
        $\theta_{t+1} \leftarrow \theta_t + \eta \cdot \frac{\partial V^{\pi_{\theta_t}}(\mu)}{\partial \theta_t}$.
    **end while**

---

As shown in Mei et al. [2], the following non-uniform Łojasiewicz (NŁ) inequality holds for value function, generalizing Lemma 1.

**Lemma 9** (NŁ, [2]). *Let $a^*(s)$ be the action that $\pi^*$ selects in state $s$. We have, for all $\theta \in \mathbb{R}^{S \times A}$,*

$$\left\| \frac{\partial V^{\pi_\theta}(\mu)}{\partial \theta} \right\|_2 \geq \left\| \frac{d_\rho^{\pi^*}}{d_\mu^{\pi_\theta}} \right\|_\infty^{-1} \cdot \min_s \pi_\theta(a^*(s)|s) \cdot \frac{1}{\sqrt{S}} \cdot (V^*(\rho) - V^{\pi_\theta}(\rho)). \tag{431}$$

The proof for Lemma 9 can be found in [2, Lemma 8].

**Proposition 6** (PG upper bound [2]). *Let Assumption 2 hold and let $\{\theta_t\}_{t \geq 1}$ be generated using Algorithm 1 with $\eta = (1-\gamma)^3/8$. Then, for all $t \geq 1$,*

$$V^*(\rho) - V^{\pi_{\theta_t}}(\rho) \leq \frac{16 \cdot S}{c^2 \cdot (1-\gamma)^6 \cdot t} \cdot \left\| \frac{d_\mu^{\pi^*}}{\mu} \right\|_\infty^2 \cdot \left\| \frac{1}{\mu} \right\|_\infty, \tag{432}$$

*where $c := \inf_{s \in \mathcal{S}, t \geq 1} \pi_{\theta_t}(a^*(s)|s) > 0$.*

The proof for Proposition 6 can be found in [2, Theorem 4].

**Proposition 7** (PG lower bound [2]). *For large enough $t \geq 1$, using Algorithm 1 with $\eta_t \in (0, 1]$,*

$$V^*(\mu) - V^{\pi_{\theta_t}}(\mu) \geq \frac{(1-\gamma)^5 \cdot (\Delta^*)^2}{12 \cdot t}, \tag{433}$$

*where $\Delta^* := \min_{s \in \mathcal{S}, a \neq a^*(s)}\{Q^*(s, a^*(s)) - Q^*(s, a)\} > 0$ is the optimal value gap of the MDP.*

The proof for Proposition 7 can be found in [2, Theorem 10].

### E.2.2 NPG

The following NPG algorithm enjoys $O(e^{-c \cdot t})$ global convergence rate in general MDPs.

---
**Algorithm 2** Natural PG (NPG), true gradient

---
**Input:** Learning rate $\eta > 0$.
**Output:** Policies $\pi_{\theta_t} = \text{softmax}(\theta_t)$.
Initialize parameter $\theta_1(s, a)$ for all $(s, a) \in \mathcal{S} \times \mathcal{A}$.
**while** $t \geq 1$ **do**
$\quad \theta_{t+1} \leftarrow \theta_t + \eta \cdot Q^{\pi_{\theta_t}}$.
**end while**

---

**Proposition 8** (NPG upper bound [8]). *Let Assumption 2 hold and let $\{\theta_t\}_{t \geq 1}$ be generated using Algorithm 2 with $\eta > 0$, and $\pi_{\theta_1}(a|s) = 1/A$ for all $(s, a)$. We have, for all $t \geq 1$,*

$$V^*(\rho) - V^{\pi_{\theta_t}}(\rho) \leq \frac{1}{(1-\gamma)^2} \cdot e^{-(t-\kappa) \cdot (1-1/\lambda) \cdot \eta \cdot \Delta^*}, \tag{434}$$

*where $\kappa = \frac{\lambda}{\Delta^*} \cdot \left[ \frac{\log A}{\eta} + \frac{1}{(1-\gamma)^2} \right]$ and $\lambda > 1$.*

The proof for Proposition 8 can be found in [8, Theorem 3.1].

We provide a different analysis using the natural non-uniform Łojasiewciz (NŁ) inequality. The following Lemma 10 generalizes the natural NŁ inequality of Lemma 3. Lemma 10 and Theorem 12 are consistent with existing work of using adaptive / large learning rates and line search in NPG [8, 29]. As shown later in Eq. (436), since the $c(\theta_t)$ quantity could be very small, it is necessary to use a large learning rate $\eta > 0$ to get constant progresses.

**Lemma 10** (Natural NŁ inequality, discrete). *Using Algorithm 2, we have, for all $t \geq 1$,*

$$V^{\pi_{\theta_{t+1}}}(\rho) - V^{\pi_{\theta_t}}(\rho) \geq c(\theta_t) \cdot (1-\gamma) \cdot \left\| \frac{d_\rho^{\pi^*}}{\rho} \right\|_\infty^{-1} \cdot \left[ V^{\pi^*}(\rho) - V^{\pi_{\theta_t}}(\rho) \right], \tag{435}$$

*where $c(\theta_t) > 0$ depends on $\theta_t$ is given by*

$$c(\theta_t) := \min_{s \in \mathcal{S}} \left[ 1 - \frac{1}{\pi_{\theta_t}(\bar{a}_t(s)|s) \cdot (e^{\eta \cdot \Delta_t(s)} - 1) + 1} \right] \in (0, 1), \tag{436}$$

and $\bar{a}_t(s)$ is the greedy action under state $s$, i.e.,

$$\bar{a}_t(s) := \arg\max_{a \in \mathcal{A}} Q^{\pi_{\theta_t}}(s, a), \tag{437}$$

and $\Delta_t(s)$ is the gap w.r.t. the greedy action $\bar{a}_t(s)$ under state $s$, i.e.,

$$\Delta_t(s) := Q^{\pi_{\theta_t}}(s, \bar{a}_t(s)) - \max_{a \neq \bar{a}_t(s)} \{Q^{\pi_{\theta_t}}(s, a)\} > 0. \tag{438}$$

*Proof.* According to the performance difference Lemma 19, we have

$$V^{\pi_{\theta_{t+1}}}(\rho) - V^{\pi_{\theta_t}}(\rho) = \frac{1}{1-\gamma} \cdot \sum_s d_\rho^{\pi_{\theta_{t+1}}}(s) \cdot \sum_a \left(\pi_{\theta_{t+1}}(a|s) - \pi_{\theta_t}(a|s)\right) \cdot Q^{\pi_{\theta_t}}(s, a) \quad (439)$$

$$\geq \sum_s \frac{d_\rho^{\pi_{\theta_{t+1}}}(s)}{1-\gamma} \cdot \left[1 - \frac{1}{\pi_{\theta_t}(\bar{a}_t(s)|s) \cdot \left(e^{\eta \cdot \Delta_t(s)} - 1\right) + 1}\right] \cdot \sum_a \left(\bar{\pi}_t(a|s) - \pi_{\theta_t}(a|s)\right) \cdot Q^{\pi_{\theta_t}}(s, a) \quad \text{(Lemma 3)}$$

$$\tag{440}$$

$$\geq \sum_s \frac{d_\rho^{\pi_{\theta_{t+1}}}(s)}{1-\gamma} \cdot \left[1 - \frac{1}{\pi_{\theta_t}(\bar{a}_t(s)|s) \cdot \left(e^{\eta \cdot \Delta_t(s)} - 1\right) + 1}\right] \cdot \sum_a \left(\pi^*(a|s) - \pi_{\theta_t}(a|s)\right) \cdot Q^{\pi_{\theta_t}}(s, a),$$

$$\tag{441}$$

where in the second last inequality $\bar{\pi}_t(\cdot|s)$ is the greedy policy under state $s$, i.e.,

$$\sum_{a \in \mathcal{A}} \bar{\pi}_t(a|s) \cdot Q^{\pi_{\theta_t}}(s, a) = \max_{\pi: \mathcal{S} \to \Delta(\mathcal{A})} \sum_{a \in \mathcal{A}} \pi(a|s) \cdot Q^{\pi_{\theta_t}}(s, a). \tag{442}$$

Next we have,

$$V^{\pi_{\theta_{t+1}}}(\rho) - V^{\pi_{\theta_t}}(\rho) \geq \frac{c(\theta_t)}{1-\gamma} \cdot \sum_s d_\rho^{\pi_{\theta_{t+1}}}(s) \cdot \sum_a \left(\pi^*(a|s) - \pi_{\theta_t}(a|s)\right) \cdot Q^{\pi_{\theta_t}}(s, a) \tag{443}$$

$$= \frac{c(\theta_t)}{1-\gamma} \cdot \sum_s d_\rho^{\pi^*}(s) \cdot \frac{d_\rho^{\pi_{\theta_{t+1}}}(s)}{d_\rho^{\pi^*}(s)} \cdot \sum_a \left(\pi^*(a|s) - \pi_{\theta_t}(a|s)\right) \cdot Q^{\pi_{\theta_t}}(s, a) \tag{444}$$

$$\geq c(\theta_t) \cdot \left\|\frac{d_\rho^{\pi^*}}{d_\rho^{\pi_{\theta_{t+1}}}}\right\|_\infty^{-1} \cdot \frac{1}{1-\gamma} \cdot \sum_s d_\rho^{\pi^*}(s) \cdot \sum_a \left(\pi^*(a|s) - \pi_{\theta_t}(a|s)\right) \cdot Q^{\pi_{\theta_t}}(s, a) \tag{445}$$

$$= c(\theta_t) \cdot \left\|\frac{d_\rho^{\pi^*}}{d_\rho^{\pi_{\theta_{t+1}}}}\right\|_\infty^{-1} \cdot \left[V^{\pi^*}(\rho) - V^{\pi_{\theta_t}}(\rho)\right] \qquad \text{(by Lemma 19)} \tag{446}$$

$$\geq c(\theta_t) \cdot (1-\gamma) \cdot \left\|\frac{d_\rho^{\pi^*}}{\rho}\right\|_\infty^{-1} \cdot \left[V^{\pi^*}(\rho) - V^{\pi_{\theta_t}}(\rho)\right]. \qquad \text{(by Eq. (523))} \qquad \square$$

**Theorem 12** (NPG upper bound). *Using Algorithm 2 with the following learning rate, for all $t \geq 1$,*

$$\eta_t = \frac{1}{\min_{s \in \mathcal{S}}\{\pi_{\theta_t}(\bar{a}_t(s)|s) \cdot \Delta_t(s)\}}, \tag{447}$$

*where $\bar{a}_t(s)$ and $\Delta_t(s)$ are defined in Eqs. (437) and (438), we have, for all $t \geq 1$,*

$$V^{\pi^*}(\rho) - V^{\pi_{\theta_t}}(\rho) \leq \exp\left\{-\frac{1-\gamma}{2} \cdot \left\|\frac{d_\rho^{\pi^*}}{\rho}\right\|_\infty^{-1} \cdot (t-1)\right\} \cdot \left[V^{\pi^*}(\rho) - V^{\pi_{\theta_1}}(\rho)\right]. \tag{448}$$

*Proof.* We have, for all state $s \in \mathcal{S}$ and $t \geq 1$,

$$\pi_{\theta_t}(\bar{a}_t(s)|s) \cdot \left(e^{\eta_t \cdot \Delta_t(s)} - 1\right) \geq \pi_{\theta_t}(\bar{a}_t(s)|s) \cdot \eta_t \cdot \Delta_t(s) \qquad (e^x \geq 1 + x) \tag{449}$$

$$= \frac{\pi_{\theta_t}(\bar{a}_t(s)|s) \cdot \Delta_t(s)}{\min_{s \in \mathcal{S}}\{\pi_{\theta_t}(\bar{a}_t(s)|s) \cdot \Delta_t(s)\}} \qquad \text{(by Eq. (447))} \tag{450}$$

$$\geq 1, \tag{451}$$

which implies that,

$$c(\theta_t) = \min_{s \in \mathcal{S}} \left[ 1 - \frac{1}{\pi_{\theta_t}(\bar{a}_t(s)|s) \cdot \left( e^{\eta \cdot \Delta_t(s)} - 1 \right) + 1} \right] \qquad \text{(by Eq. (436))} \qquad (452)$$

$$\geq 1 - 1/2 = 1/2. \qquad (453)$$

According to Lemma 10, we have, for all $t \geq 1$,

$$V^{\pi_{\theta_{t+1}}}(\rho) - V^{\pi_{\theta_t}}(\rho) \geq c(\theta_t) \cdot (1 - \gamma) \cdot \left\| \frac{d_\rho^{\pi^*}}{\rho} \right\|_\infty^{-1} \cdot \left[ V^{\pi^*}(\rho) - V^{\pi_{\theta_t}}(\rho) \right] \qquad (454)$$

$$\geq \frac{1 - \gamma}{2} \cdot \left\| \frac{d_\rho^{\pi^*}}{\rho} \right\|_\infty^{-1} \cdot \left[ V^{\pi^*}(\rho) - V^{\pi_{\theta_t}}(\rho) \right], \qquad (455)$$

which leads to the final result,

$$V^{\pi^*}(\rho) - V^{\pi_{\theta_t}}(\rho) = V^{\pi^*}(\rho) - V^{\pi_{\theta_{t-1}}}(\rho) - \left[ V^{\pi_{\theta_t}}(\rho) - V^{\pi_{\theta_{t-1}}}(\rho) \right] \qquad (456)$$

$$\leq \left( 1 - \frac{1 - \gamma}{2} \cdot \left\| \frac{d_\rho^{\pi^*}}{\rho} \right\|_\infty^{-1} \right) \cdot \left[ V^{\pi^*}(\rho) - V^{\pi_{\theta_{t-1}}}(\rho) \right] \qquad (457)$$

$$\leq \exp \left\{ -\frac{1 - \gamma}{2} \cdot \left\| \frac{d_\rho^{\pi^*}}{\rho} \right\|_\infty^{-1} \cdot (t - 1) \right\} \cdot \left[ V^{\pi^*}(\rho) - V^{\pi_{\theta_1}}(\rho) \right]. \qquad \square$$

### E.2.3 GNPG

Finally, GNPG also enjoys $O(e^{-c \cdot t})$ global convergence rate in general MDPs.

---

**Algorithm 3** Geometry-award normalized PG (GNPG), true gradient

**Input:** Learning rate $\eta > 0$.
**Output:** Policies $\pi_{\theta_t} = \text{softmax}(\theta_t)$.
Initialize parameter $\theta_1(s, a)$ for all $(s, a) \in \mathcal{S} \times \mathcal{A}$.
**while** $t \geq 1$ **do**
$\quad \theta_{t+1} \leftarrow \theta_t + \eta \cdot \frac{\partial V^{\pi_{\theta_t}}(\mu)}{\partial \theta_t} \Big/ \left\| \frac{\partial V^{\pi_{\theta_t}}(\mu)}{\partial \theta_t} \right\|_2$.
**end while**

---

**Proposition 9** (GNPG upper bound [9]). *Let Assumption 2 hold and let $\{\theta_t\}_{t \geq 1}$ be generated using Algorithm 3 with $\eta = \frac{(1-\gamma) \cdot \gamma}{6 \cdot (1-\gamma) \cdot \gamma + 4 \cdot (C_\infty - (1-\gamma))} \cdot \frac{1}{\sqrt{S}}$, where $C_\infty := \max_\pi \left\| \frac{d_\mu^\pi}{\mu} \right\|_\infty$. Denote $C'_\infty := \max_\pi \left\| \frac{d_\rho^\pi}{\mu} \right\|_\infty$. We have, for all $t \geq 1$,*

$$V^*(\rho) - V^{\pi_{\theta_t}}(\rho) \leq \frac{(V^*(\mu) - V^{\pi_{\theta_1}}(\mu)) \cdot C'_\infty}{1 - \gamma} \cdot e^{-C \cdot (t-1)}, \qquad (458)$$

*where $C = \frac{(1-\gamma)^2 \cdot \gamma \cdot c}{12 \cdot (1-\gamma) \cdot \gamma + 8 \cdot (C_\infty - (1-\gamma))} \cdot \frac{1}{S} \cdot \left\| \frac{d_\mu^{\pi^*}}{\mu} \right\|_\infty^{-1}$ and $c := \inf_{s \in \mathcal{S}, t \geq 1} \pi_{\theta_t}(a^*(s)|s) > 0$.*

The proof for Proposition 9 can be found in [9, Theorem 3].

### E.3 On-policy Stochastic Gradient Settings

For general MDPs with multiple states, we define the following on-policy parallel importance sampling (IS) estimator, using one sampled action under each state to estimate the policy gradient.

**Definition 3** (On-policy parallel IS). *At iteration $t$, under each state $s$, sample one action $a_t(s) \sim \pi_{\theta_t}(\cdot|s)$. The IS state-action value estimator $\hat{Q}^{\pi_{\theta_t}}$ is constructed as*

$$\hat{Q}^{\pi_{\theta_t}}(s, a) = \frac{\mathbb{I}\{a_t(s) = a\}}{\pi_{\theta_t}(a|s)} \cdot Q^{\pi_{\theta_t}}(s, a), \qquad (459)$$

*for all $(s, a) \in \mathcal{S} \times \mathcal{A}$.*

Definition 3 is a generalized version of Definition 1 to general MDPs, which does not specify how to estimate $Q^{\pi_{\theta_t}}(s, a)$. With Definition 3, the on-policy stochastic softmax PG, NPG, and GNPG methods can be generalized to general MDPs, replacing the true action value $Q^{\pi_{\theta_t}}(s, \cdot) \in \mathbb{R}^{|\mathcal{A}|}$ in the policy gradient with $\hat{Q}^{\pi_{\theta_t}}(s, \cdot)$, which uses one sampled action under each state $s$. In practice, $\hat{Q}^{\pi_{\theta_t}}(s, \cdot)$ can be calculated using realistic PG estimators from on-policy roll-outs [30, Algorithm 1]. Here we use Definition 3 to show the main ideas of this work.

### E.3.1 Softmax PG

We show that Theorem 2 can be generalized to general MDPs, using unbiased and bounded softmax PG properties.

According to the policy gradient theorem [1], the softmax PG used in Algorithm 1 is, for all $s \in \mathcal{S}$,

$$\frac{\partial V^{\pi_\theta}(\mu)}{\partial \theta(s, \cdot)} = \frac{1}{1 - \gamma} \cdot \sum_{s'} d_\mu^{\pi_\theta}(s') \cdot \left[ \sum_a \frac{\partial \pi_\theta(a|s')}{\partial \theta(s, \cdot)} \cdot Q^{\pi_\theta}(s', a) \right] \tag{460}$$

$$= \frac{1}{1 - \gamma} \cdot d_\mu^{\pi_\theta}(s) \cdot \left[ \sum_a \frac{\partial \pi_\theta(a|s)}{\partial \theta(s, \cdot)} \cdot Q^{\pi_\theta}(s, a) \right]. \qquad \left( \frac{\partial \pi_\theta(a|s')}{\partial \theta(s, \cdot)} = \mathbf{0}, \ \forall s' \neq s \right) \tag{461}$$

Calculating $\frac{\partial \pi_\theta(a|s)}{\partial \theta(s, \cdot)}$ for $\pi_\theta(\cdot|s) = \text{softmax}(\theta(s, \cdot))$, we have [7, 2], for all $(s, a) \in \mathcal{S} \times \mathcal{A}$,

$$\frac{\partial V^{\pi_\theta}(\mu)}{\partial \theta(s, a)} = \frac{1}{1 - \gamma} \cdot d_\mu^{\pi_\theta}(s) \cdot \pi_\theta(a|s) \cdot A^{\pi_\theta}(s, a). \tag{462}$$

Replacing $Q^{\pi_\theta}(s, a)$ with $\hat{Q}^{\pi_\theta}(s, a)$ in Definition 3, we have Algorithm 4.

---

**Algorithm 4** Softmax PG, on-policy stochastic gradient

**Input:** Learning rate $\eta > 0$.
**Output:** Policies $\pi_{\theta_t} = \text{softmax}(\theta_t)$.
Initialize parameter $\theta_1(s, a)$ for all $(s, a) \in \mathcal{S} \times \mathcal{A}$.
**while** $t \geq 1$ **do**
    Sample $a_t(s) \sim \pi_{\theta_t}(\cdot|s)$ for all $s \in \mathcal{S}$.
    $\hat{Q}^{\pi_{\theta_t}}(s, a) \leftarrow \frac{\mathbb{I}\{a_t(s) = a\}}{\pi_{\theta_t}(a|s)} \cdot Q^{\pi_{\theta_t}}(s, a).$     (by Definition 3)
    $\hat{g}_t(s, \cdot) \leftarrow \frac{1}{1-\gamma} \cdot d_\mu^{\pi_{\theta_t}}(s) \cdot \left[ \sum_a \frac{\partial \pi_{\theta_t}(a|s)}{\partial \theta_t(s, \cdot)} \cdot \hat{Q}^{\pi_{\theta_t}}(s, a) \right].$
    $\theta_{t+1} \leftarrow \theta_t + \eta \cdot \hat{g}_t.$
**end while**

---

Similarly, we have, for all $(s, a) \in \mathcal{S} \times \mathcal{A}$,

$$\theta_{t+1}(s, a) \leftarrow \theta_t(s, a) + \frac{\eta}{1 - \gamma} \cdot d_\mu^{\pi_{\theta_t}}(s) \cdot \pi_{\theta_t}(a|s) \cdot \left[ \hat{Q}^{\pi_{\theta_t}}(s, a) - \pi_{\theta_t}(\cdot|s)^\top \hat{Q}^{\pi_{\theta_t}}(s, \cdot) \right]. \tag{463}$$

We show that the PG estimator in Algorithm 4 is unbiased and bounded, generalizing Lemma 5.

**Lemma 11.** *Let $\hat{Q}^{\pi_\theta}(s, \cdot)$ be the IS parallel estimator using on-policy sampling $a(s) \sim \pi_\theta(\cdot|s)$, for all $s$. The stochastic softmax PG estimator is unbiased and bounded, i.e.,*

$$\underset{a(s) \sim \pi_\theta(\cdot|s)}{\mathbb{E}} \left[ \frac{1}{1 - \gamma} \cdot d_\mu^{\pi_\theta}(s) \cdot \pi_\theta(a|s) \cdot \left( \hat{Q}^{\pi_\theta}(s, a) - \pi_\theta(\cdot|s)^\top \hat{Q}^{\pi_\theta}(s, \cdot) \right) \right] = \frac{\partial V^{\pi_\theta}(\mu)}{\partial \theta(s, a)}, \tag{464}$$

$$\underset{a(s) \sim \pi_\theta(\cdot|s)}{\mathbb{E}} \left[ \sum_{(s,a)} \frac{d_\mu^{\pi_\theta}(s)^2 \cdot \pi_\theta(a|s)^2}{(1 - \gamma)^2} \cdot \left( \hat{Q}^{\pi_\theta}(s, a) - \pi_\theta(\cdot|s)^\top \hat{Q}^{\pi_\theta}(s, \cdot) \right)^2 \right] \leq \frac{2}{(1 - \gamma)^4}. \tag{465}$$

*Proof.* **First part.** Unbiased.

According to Definition 3, we have,

$$\pi_\theta(a|s) \cdot \left( \hat{Q}^{\pi_\theta}(s,a) - \pi_\theta(\cdot|s)^\top \hat{Q}^{\pi_\theta}(s,\cdot) \right) \tag{466}$$

$$= \pi_\theta(a|s) \cdot \left( \frac{\mathbb{I}\{a(s)=a\}}{\pi_\theta(a|s)} \cdot Q^{\pi_\theta}(s,a) - \sum_{a'\in\mathcal{A}} \mathbb{I}\{a(s)=a'\} \cdot Q^{\pi_\theta}(s,a') \right) \tag{467}$$

$$= \mathbb{I}\{a(s)=a\} \cdot Q^{\pi_\theta}(s,a) - \pi_\theta(a|s) \cdot Q^{\pi_\theta}(s,a(s)). \tag{468}$$

Taking expectation, we have,

$$\mathop{\mathbb{E}}_{a(s)\sim\pi_\theta(\cdot|s)} \left[ \pi_\theta(a|s) \cdot \left( \hat{Q}^{\pi_\theta}(s,a) - \pi_\theta(\cdot|s)^\top \hat{Q}^{\pi_\theta}(s,\cdot) \right) \right] \tag{469}$$

$$= \sum_{a(s)\in\mathcal{A}} \pi_\theta(a(s)|s) \cdot \left[ \mathbb{I}\{a(s)=a\} \cdot Q^{\pi_\theta}(s,a) - \pi_\theta(a|s) \cdot Q^{\pi_\theta}(s,a(s)) \right] \tag{470}$$

$$= \pi_\theta(a|s) \cdot Q^{\pi_\theta}(s,a) - \pi_\theta(a|s) \cdot V^{\pi_\theta}(s) \tag{471}$$

$$= \pi_\theta(a|s) \cdot A^{\pi_\theta}(s,a). \tag{472}$$

**Second part.** Bounded.

First, using similar calculations in the second part of Lemma 5, we have, for all $s \in \mathcal{S}$,

$$\sum_{a\in\mathcal{A}} \pi_\theta(a|s)^2 \cdot \left( \hat{Q}^{\pi_\theta}(s,a) - \pi_\theta(\cdot|s)^\top \hat{Q}^{\pi_\theta}(s,\cdot) \right)^2 \tag{473}$$

$$= \sum_{a\in\mathcal{A}} \pi_\theta(a|s)^2 \cdot \left[ \frac{\mathbb{I}\{a(s)=a\}}{\pi_\theta(a|s)^2} \cdot Q^{\pi_\theta}(s,a)^2 \right. \tag{474}$$

$$\left. - 2 \cdot \frac{\mathbb{I}\{a(s)=a\}}{\pi_\theta(a|s)} \cdot Q^{\pi_\theta}(s,a) \cdot \pi_\theta(\cdot|s)^\top \hat{Q}^{\pi_\theta}(s,\cdot) + \left( \pi_\theta(\cdot|s)^\top \hat{Q}^{\pi_\theta}(s,\cdot) \right)^2 \right] \tag{475}$$

$$= Q^{\pi_\theta}(s,a(s))^2 - 2 \cdot \pi_\theta(a(s)|s) \cdot Q^{\pi_\theta}(s,a(s))^2 + \sum_{a\in\mathcal{A}} \pi_\theta(a|s)^2 \cdot Q^{\pi_\theta}(s,a(s))^2 \tag{476}$$

$$= [1 - \pi_\theta(a(s)|s)]^2 \cdot Q^{\pi_\theta}(s,a(s))^2 + \sum_{a\neq a(s)} \pi_\theta(a|s)^2 \cdot Q^{\pi_\theta}(s,a(s))^2 \tag{477}$$

$$\leq 2 \cdot [1 - \pi_\theta(a(s)|s)]^2 \cdot Q^{\pi_\theta}(s,a(s))^2 \qquad (\|x\|_2 \leq \|x\|_1) \tag{478}$$

$$\leq \frac{2}{(1-\gamma)^2}. \qquad (\pi_\theta(a(s)|s) \in (0,1), \text{ and } Q^{\pi_\theta}(s,a(s)) \in (0,1/(1-\gamma)]) \tag{479}$$

Therefore we have,

$$\mathop{\mathbb{E}}_{a(s)\sim\pi_\theta(\cdot|s)} \left[ \sum_{(s,a)} \frac{d_\mu^{\pi_\theta}(s)^2 \cdot \pi_\theta(a|s)^2}{(1-\gamma)^2} \cdot \left( \hat{Q}^{\pi_\theta}(s,a) - \pi_\theta(\cdot|s)^\top \hat{Q}^{\pi_\theta}(s,\cdot) \right)^2 \right] \tag{480}$$

$$\leq \sum_{s\in\mathcal{S}} \frac{d_\mu^{\pi_\theta}(s)^2}{(1-\gamma)^2} \cdot \frac{2}{(1-\gamma)^2} \qquad (\text{by Eq. (473)}) \tag{481}$$

$$\leq \frac{2}{(1-\gamma)^4} \cdot \left[ \sum_{s\in\mathcal{S}} d_\mu^{\pi_\theta}(s) \right]^2 \qquad (\|x\|_2 \leq \|x\|_1) \tag{482}$$

$$= \frac{2}{(1-\gamma)^4}. \qquad\qquad \square \tag{483}$$

The following lemma generalizes Lemma 7.

**Lemma 12** (Non-uniform Smoothness (NS) between two iterations). *Using stochastic softmax PG update, i.e.,*

$$\theta' = \theta + \eta \cdot \hat{g} \tag{483}$$

$$:= \theta + \eta \cdot \frac{1}{1-\gamma} \cdot \mathop{\mathbb{E}}_{s'\sim d_\mu^{\pi_\theta}} \left[ \sum_a \frac{\partial \pi_\theta(a|s')}{\partial \theta} \cdot \hat{Q}^{\pi_\theta}(s',a) \right], \tag{484}$$

*and using the true softmax PG norm in learning rate, i.e.,*

$$\eta = \frac{(1-\gamma)^4}{4 \cdot C} \cdot \left\| \frac{\partial V^{\pi_\theta}(\mu)}{\partial \theta} \right\|_2, \tag{485}$$

*where*

$$C := \left[ 3 + \frac{2 \cdot (C_\infty - (1-\gamma))}{(1-\gamma) \cdot \gamma} \right] \cdot \sqrt{S}, \tag{486}$$

*and $C_\infty := \max_\pi \left\| \frac{d_\mu^\pi}{\mu} \right\|_\infty \le \frac{1}{\min_s \mu(s)} < \infty$ given Assumption 2 hold, we have,*

$$\left| V^{\pi_{\theta'}}(\mu) - V^{\pi_\theta}(\mu) - \left\langle \frac{\partial V^{\pi_\theta}(\mu)}{\partial \theta}, \theta' - \theta \right\rangle \right| \le C \cdot \left\| \frac{\partial V^{\pi_\theta}(\mu)}{\partial \theta} \right\|_2 \cdot \| \theta' - \theta \|_2^2. \tag{487}$$

*Proof.* According to the non-uniform smoothness of value function [9, Lemma 6], we have, for all $y \in \mathbb{R}^{SA}$ and $\theta$,

$$\left| y^\top \frac{\partial^2 V^{\pi_\theta}(\mu)}{\partial \theta^2} y \right| \le C \cdot \left\| \frac{\partial V^{\pi_\theta}(\mu)}{\partial \theta} \right\|_2 \cdot \| y \|_2^2. \tag{488}$$

The proof is then similar to Lemma 7. Denote $\theta_\zeta := \theta + \zeta \cdot (\theta' - \theta)$ with some $\zeta \in [0, 1]$. According to Taylor's theorem, $\forall \theta, \theta'$,

$$\left| V^{\pi_{\theta'}}(\mu) - V^{\pi_\theta}(\mu) - \left\langle \frac{\partial V^{\pi_\theta}(\mu)}{\partial \theta}, \theta' - \theta \right\rangle \right| = \frac{1}{2} \cdot \left| (\theta' - \theta)^\top \frac{\partial^2 V^{\pi_{\theta_\zeta}}(\mu)}{\partial \theta_\zeta^2} (\theta' - \theta) \right| \tag{489}$$

$$\le \frac{C}{2} \cdot \left\| \frac{\partial V^{\pi_{\theta_\zeta}}(\mu)}{\partial \theta_\zeta} \right\|_2 \cdot \| \theta' - \theta \|_2^2, \qquad \text{(by Eq. (488))} \tag{490}$$

where for conciseness we denote,

Denote $\zeta_1 := \zeta$. Also denote $\theta_{\zeta_2} := \theta + \zeta_2 \cdot (\theta_{\zeta_1} - \theta)$ with some $\zeta_2 \in [0, 1]$. We have,

$$\left\| \frac{\partial V^{\pi_{\theta_{\zeta_1}}}(\mu)}{\partial \theta_{\zeta_1}} - \frac{\partial V^{\pi_\theta}(\mu)}{\partial \theta} \right\|_2 = \left\| \int_0^1 \left\langle \frac{\partial^2 V^{\pi_{\theta_{\zeta_2}}}(\mu)}{\partial \theta_{\zeta_2}^2}, \theta_{\zeta_1} - \theta \right\rangle d\zeta_2 \right\|_2 \tag{491}$$

$$\le \int_0^1 \left\| \frac{\partial^2 V^{\pi_{\theta_{\zeta_2}}}(\mu)}{\partial \theta_{\zeta_2}^2} \right\|_2 \cdot \| \theta_{\zeta_1} - \theta \|_2 \, d\zeta_2 \qquad \text{(by Cauchy–Schwarz)} \tag{492}$$

$$\le \int_0^1 C \cdot \left\| \frac{\partial V^{\pi_{\theta_{\zeta_2}}}(\mu)}{\partial \theta_{\zeta_2}} \right\|_2 \cdot \zeta_1 \cdot \| \theta' - \theta \|_2 \, d\zeta_2 \qquad \text{(by Eq. (488))} \tag{493}$$

$$\le \int_0^1 C \cdot \left\| \frac{\partial V^{\pi_{\theta_{\zeta_2}}}(\mu)}{\partial \theta_{\zeta_2}} \right\|_2 \cdot \eta \cdot \| \hat{g} \|_2 \, d\zeta_2, \qquad (\zeta_1 \in [0, 1], \text{ using Eq. (483)}) \tag{494}$$

where the second inequality is because of the Hessian is symmetric, and its operator norm is equal to its spectral radius. Therefore we have,

$$\left\| \frac{\partial V^{\pi_{\theta_{\zeta_1}}}(\mu)}{\partial \theta_{\zeta_1}} \right\|_2 \le \left\| \frac{\partial V^{\pi_\theta}(\mu)}{\partial \theta} \right\|_2 + \left\| \frac{\partial V^{\pi_{\theta_{\zeta_1}}}(\mu)}{\partial \theta_{\zeta_1}} - \frac{\partial V^{\pi_\theta}(\mu)}{\partial \theta} \right\|_2 \qquad \text{(by triangle inequality)} \tag{495}$$

$$\le \left\| \frac{\partial V^{\pi_\theta}(\mu)}{\partial \theta} \right\|_2 + C \cdot \eta \cdot \| \hat{g} \|_2 \cdot \int_0^1 \left\| \frac{\partial V^{\pi_{\theta_{\zeta_2}}}(\mu)}{\partial \theta_{\zeta_2}} \right\|_2 d\zeta_2. \qquad \text{(by Eq. (491))} \tag{496}$$

Denote $\theta_{\zeta_3} := \theta + \zeta_3 \cdot (\theta_{\zeta_2} - \theta)$ with $\zeta_3 \in [0, 1]$. Using similar calculation in Eq. (491), we have,

$$\left\| \frac{\partial V^{\pi_{\theta_{\zeta_2}}}(\mu)}{\partial \theta_{\zeta_2}} \right\|_2 \le \left\| \frac{\partial V^{\pi_\theta}(\mu)}{\partial \theta} \right\|_2 + \left\| \frac{\partial V^{\pi_{\theta_{\zeta_2}}}(\mu)}{\partial \theta_{\zeta_2}} - \frac{\partial V^{\pi_\theta}(\mu)}{\partial \theta} \right\|_2 \tag{497}$$

$$\le \left\| \frac{\partial V^{\pi_\theta}(\mu)}{\partial \theta} \right\|_2 + C \cdot \eta \cdot \| \hat{g} \|_2 \cdot \int_0^1 \left\| \frac{\partial V^{\pi_{\theta_{\zeta_3}}}(\mu)}{\partial \theta_{\zeta_3}} \right\|_2 d\zeta_3. \tag{498}$$

Combining Eqs. (495) and (497), we have,

$$\left\|\frac{\partial V^{\pi_{\theta_{\zeta_1}}}(\mu)}{\partial \theta_{\zeta_1}}\right\|_2 \le (1 + C \cdot \eta \cdot \|\hat{g}\|_2) \cdot \left\|\frac{\partial V^{\pi_\theta}(\mu)}{\partial \theta}\right\|_2 \tag{499}$$

$$+ (C \cdot \eta \cdot \|\hat{g}\|_2)^2 \cdot \int_0^1 \int_0^1 \left\|\frac{\partial V^{\pi_{\theta_{\zeta_3}}}(\mu)}{\partial \theta_{\zeta_3}}\right\|_2 d\zeta_3 d\zeta_2, \tag{500}$$

which implies,

$$\left\|\frac{\partial V^{\pi_{\theta_{\zeta_1}}}(\mu)}{\partial \theta_{\zeta_1}}\right\|_2 \le \sum_{i=0}^{\infty} (C \cdot \eta \cdot \|\hat{g}\|_2)^i \cdot \left\|\frac{\partial V^{\pi_\theta}(\mu)}{\partial \theta}\right\|_2 \tag{501}$$

$$= \frac{1}{1 - C \cdot \eta \cdot \|\hat{g}\|_2} \cdot \left\|\frac{\partial V^{\pi_\theta}(\mu)}{\partial \theta}\right\|_2 \qquad (C \cdot \eta \cdot \|\hat{g}\|_2 \in (0,1), \text{ see below}) \tag{502}$$

$$= \frac{1}{1 - \frac{(1-\gamma)^4}{4} \cdot \|\hat{g}\|_2 \cdot \left\|\frac{\partial V^{\pi_\theta}(\mu)}{\partial \theta}\right\|_2} \cdot \left\|\frac{\partial V^{\pi_\theta}(\mu)}{\partial \theta}\right\|_2 \qquad \left(\eta = \frac{(1-\gamma)^4}{4 \cdot C} \cdot \left\|\frac{\partial V^{\pi_\theta}(\mu)}{\partial \theta}\right\|_2\right) \tag{503}$$

$$\le \frac{1}{1 - \frac{(1-\gamma)^2}{4} \cdot \|\hat{g}\|_2} \cdot \left\|\frac{\partial V^{\pi_\theta}(\mu)}{\partial \theta}\right\|_2 \qquad \left(\left\|\frac{\partial V^{\pi_\theta}(\mu)}{\partial \theta}\right\|_2 \le \frac{1}{(1-\gamma)^2}, \text{ see below}\right) \tag{504}$$

$$\le \frac{1}{1 - \frac{1}{2}} \cdot \left\|\frac{\partial V^{\pi_\theta}(\mu)}{\partial \theta}\right\|_2 \qquad \left(\|\hat{g}\|_2 \le \frac{2}{(1-\gamma)^2}, \text{ see below}\right) \tag{505}$$

$$= 2 \cdot \left\|\frac{\partial V^{\pi_\theta}(\mu)}{\partial \theta}\right\|_2, \tag{506}$$

where the last inequality is from,

$$\|\hat{g}\|_2^2 = \sum_{(s,a) \in \mathcal{S} \times \mathcal{A}} \hat{g}(s,a)^2 \tag{507}$$

$$= \sum_{(s,a)} \frac{d_\mu^{\pi_\theta}(s)^2 \cdot \pi_\theta(a|s)^2}{(1-\gamma)^2} \cdot \left(\hat{Q}^{\pi_\theta}(s,a) - \pi_\theta(\cdot|s)^\top \hat{Q}^{\pi_\theta}(s,\cdot)\right)^2 \qquad \text{(by Eq. (463))} \tag{508}$$

$$\le \frac{2}{(1-\gamma)^4}, \qquad \text{(by Lemma 11)} \tag{509}$$

which implies that,

$$\|\hat{g}\|_2 \le \frac{\sqrt{2}}{(1-\gamma)^2} \le \frac{2}{(1-\gamma)^2}, \tag{510}$$

and the second last inequality is from,

$$\left\|\frac{\partial V^{\pi_\theta}(\mu)}{\partial \theta}\right\|_2^2 = \sum_{s \in \mathcal{S}} \frac{d_\mu^{\pi_\theta}(s)^2}{(1-\gamma)^2} \cdot \sum_{a \in \mathcal{A}} \pi_\theta(a|s)^2 \cdot A^{\pi_\theta}(s,a)^2 \qquad \text{(by Eq. (462))} \tag{511}$$

$$\le \frac{1}{(1-\gamma)^4} \cdot \sum_{s \in \mathcal{S}} d_\mu^{\pi_\theta}(s)^2 \cdot \sum_{a \in \mathcal{A}} \pi_\theta(a|s)^2 \qquad \left(|A^{\pi_\theta}(s,a)| \le \frac{1}{1-\gamma}\right) \tag{512}$$

$$\le \frac{1}{(1-\gamma)^4} \cdot \sum_{s \in \mathcal{S}} d_\mu^{\pi_\theta}(s)^2 \cdot \left[\sum_{a \in \mathcal{A}} \pi_\theta(a|s)\right]^2 \qquad (\|x\|_2 \le \|x\|_1) \tag{513}$$

$$\le \frac{1}{(1-\gamma)^4} \cdot \left[\sum_{s \in \mathcal{S}} d_\mu^{\pi_\theta}(s)\right]^2 \qquad (\|x\|_2 \le \|x\|_1) \tag{514}$$

$$= \frac{1}{(1-\gamma)^4}, \tag{515}$$

$$\tag{516}$$

which implies that,

$$\left\|\frac{\partial V^{\pi_\theta}(\mu)}{\partial \theta}\right\|_2 \le \frac{1}{(1-\gamma)^2}. \tag{517}$$

Combining Eqs. (489) and (501) finishes the proof. $\qquad\square$

The following result generalizes Theorem 2.

**Theorem 13.** *Let* $\{\theta_t\}_{t\ge 1}$ *be generated by using Algorithm 4, i.e., for all* $t \ge 1$,

$$\theta_{t+1} = \theta_t + \eta \cdot \hat{g}_t \tag{518}$$

$$:= \theta_t + \eta \cdot \frac{1}{1-\gamma} \cdot \mathop{\mathbb{E}}_{s' \sim d_\mu^{\pi_{\theta_t}}}\left[\sum_a \frac{\partial \pi_{\theta_t}(a|s')}{\partial \theta_t} \cdot \hat{Q}^{\pi_{\theta_t}}(s',a)\right], \tag{519}$$

*with learning rate*

$$\eta = \frac{(1-\gamma)^4}{4 \cdot C} \cdot \left\|\frac{\partial V^{\pi_{\theta_t}}(\mu)}{\partial \theta_t}\right\|_2 \tag{520}$$

*for all* $t \ge 1$, *and*

$$C := \left[3 + \frac{2 \cdot (C_\infty - (1-\gamma))}{(1-\gamma) \cdot \gamma}\right] \cdot \sqrt{S} \tag{521}$$

*as defined in Eq. (486), where* $C_\infty := \max_\pi \left\|\frac{d_\mu^\pi}{\mu}\right\|_\infty \le \frac{1}{\min_s \mu(s)} < \infty$. *Denote* $C'_\infty :=$ $\max_\pi \left\|\frac{d_\rho^\pi}{\mu}\right\|_\infty$. *We have,* $V^*(\rho) - V^{\pi_{\theta_t}}(\rho) \to 0$ *as* $t \to \infty$ *in probability, and for all* $t \ge 1$,

$$\mathbb{E}\left[V^*(\rho) - V^{\pi_{\theta_t}}(\rho)\right] \le \frac{2 \cdot S}{\sqrt{c}} \cdot \frac{\sqrt{3 \cdot (1-\gamma) \cdot \gamma + 2 \cdot (C_\infty - (1-\gamma))}}{(1-\gamma)^5 \cdot \sqrt{\gamma}} \cdot \left\|\frac{d_\rho^{\pi^*}}{\mu}\right\|_\infty^{3/2} \cdot \frac{C'_\infty}{\sqrt{t}}, \tag{522}$$

*where* $c > 0$ *is independent with* $t$.

*Proof.* First note that for any $\theta$ and $\mu$,

$$d_\mu^{\pi_\theta}(s) = \mathop{\mathbb{E}}_{s_0 \sim \mu}\left[d_\mu^{\pi_\theta}(s)\right] \tag{523}$$

$$= \mathop{\mathbb{E}}_{s_0 \sim \mu}\left[(1-\gamma) \cdot \sum_{t=0}^\infty \gamma^t \Pr(s_t = s | s_0, \pi_\theta, \mathcal{P})\right] \tag{524}$$

$$\ge \mathop{\mathbb{E}}_{s_0 \sim \mu}\left[(1-\gamma) \cdot \Pr(s_0 = s | s_0)\right] \tag{525}$$

$$= (1-\gamma) \cdot \mu(s). \tag{526}$$

Next, according to Lemma 20, we have,

$$V^*(\rho) - V^{\pi_\theta}(\rho) = \frac{1}{1-\gamma} \sum_s d_\rho^{\pi_\theta}(s) \sum_a (\pi^*(a|s) - \pi_\theta(a|s)) \cdot Q^*(s,a) \tag{527}$$

$$= \frac{1}{1-\gamma} \sum_s \frac{d_\rho^{\pi_\theta}(s)}{d_\mu^{\pi_\theta}(s)} \cdot d_\mu^{\pi_\theta}(s) \sum_a (\pi^*(a|s) - \pi_\theta(a|s)) \cdot Q^*(s,a) \tag{528}$$

$$\le \frac{1}{1-\gamma} \cdot \left\|\frac{d_\rho^{\pi_\theta}}{d_\mu^{\pi_\theta}}\right\|_\infty \sum_s d_\mu^{\pi_\theta}(s) \sum_a (\pi^*(a|s) - \pi_\theta(a|s)) \cdot Q^*(s,a) \quad \left(\sum_a (\pi^*(a|s) - \pi_\theta(a|s)) \cdot Q^*(s,a) \ge 0\right) \tag{529}$$

$$\le \frac{1}{(1-\gamma)^2} \cdot \left\|\frac{d_\rho^{\pi_\theta}}{\mu}\right\|_\infty \sum_s d_\mu^{\pi_\theta}(s) \sum_a (\pi^*(a|s) - \pi_\theta(a|s)) \cdot Q^*(s,a) \quad \left(\text{by Eq. (523) and } \min_s \mu(s) > 0\right) \tag{530}$$

$$\le \frac{1}{(1-\gamma)^2} \cdot C'_\infty \cdot \sum_s d_\mu^{\pi_\theta}(s) \sum_a (\pi^*(a|s) - \pi_\theta(a|s)) \cdot Q^*(s,a) \tag{531}$$

$$= \frac{1}{1-\gamma} \cdot C'_\infty \cdot \left[V^*(\mu) - V^{\pi_\theta}(\mu)\right]. \quad \text{(by Lemma 20)} \tag{532}$$

Denote $\delta(\theta_t) \coloneqq V^*(\mu) - V^{\pi_{\theta_t}}(\mu)$. Let We have, for all $t \geq 1$,

$$\delta(\theta_{t+1}) - \delta(\theta_t) \tag{533}$$

$$= -V^{\pi_{\theta_{t+1}}}(\mu) + V^{\pi_{\theta_t}}(\mu) + \left\langle \frac{\partial V^{\pi_{\theta_t}}(\mu)}{\partial \theta_t}, \theta_{t+1} - \theta_t \right\rangle - \left\langle \frac{\partial V^{\pi_{\theta_t}}(\mu)}{\partial \theta_t}, \theta_{t+1} - \theta_t \right\rangle \tag{534}$$

$$\leq C \cdot \left\| \frac{\partial V^{\pi_{\theta_t}}(\mu)}{\partial \theta_t} \right\|_2 \cdot \|\theta_{t+1} - \theta_t\|_2^2 - \left\langle \frac{\partial V^{\pi_{\theta_t}}(\mu)}{\partial \theta_t}, \theta_{t+1} - \theta_t \right\rangle \qquad \text{(by Lemma 12)} \tag{535}$$

$$= C \cdot \eta^2 \cdot \left\| \frac{\partial V^{\pi_{\theta_t}}(\mu)}{\partial \theta_t} \right\|_2 \cdot \|\hat{g}_t\|_2^2 - \eta \cdot \left\langle \frac{\partial V^{\pi_{\theta_t}}(\mu)}{\partial \theta_t}, \hat{g}_t \right\rangle. \qquad \text{(using Eq. (518))} \tag{536}$$

Next, taking expectation over the random sampling on Eq. (533), we have,

$$\mathbb{E}\left[\delta(\theta_{t+1})\right] - \mathbb{E}\left[\delta(\theta_t)\right] \leq C \cdot \eta^2 \cdot \left\| \frac{\partial V^{\pi_{\theta_t}}(\mu)}{\partial \theta_t} \right\|_2 \cdot \mathbb{E}\left[\|\hat{g}_t\|_2^2\right] - \eta \cdot \left\langle \frac{\partial V^{\pi_{\theta_t}}(\mu)}{\partial \theta_t}, \mathbb{E}\left[\hat{g}_t\right] \right\rangle \tag{537}$$

$$= C \cdot \eta^2 \cdot \left\| \frac{\partial V^{\pi_{\theta_t}}(\mu)}{\partial \theta_t} \right\|_2 \cdot \mathbb{E}\left[\|\hat{g}_t\|_2^2\right] - \eta \cdot \left\| \frac{\partial V^{\pi_{\theta_t}}(\mu)}{\partial \theta_t} \right\|_2^2 \qquad \text{(unbiased PG, by Lemma 11)} \tag{538}$$

$$\leq \frac{2 \cdot C}{(1-\gamma)^4} \cdot \eta^2 \cdot \left\| \frac{\partial V^{\pi_{\theta_t}}(\mu)}{\partial \theta_t} \right\|_2 - \eta \cdot \left\| \frac{\partial V^{\pi_{\theta_t}}(\mu)}{\partial \theta_t} \right\|_2^2 \qquad \left( \mathbb{E}\left[\|\hat{g}_t\|_2^2\right] \leq \frac{2}{(1-\gamma)^4}, \text{ by Lemma 11} \right) \tag{539}$$

$$= -\frac{(1-\gamma)^4}{8 \cdot C} \cdot \left\| \frac{\partial V^{\pi_{\theta_t}}(\mu)}{\partial \theta_t} \right\|_2^3 \qquad \text{(by Eq. (520))} \tag{540}$$

$$\leq -\frac{(1-\gamma)^4}{8 \cdot C} \cdot \mathbb{E}\left[\min_s \pi_{\theta_t}(a^*(s)|s)^3\right] \cdot \mathbb{E}\left[\delta(\theta_t)^3\right] \cdot \left\| \frac{d_\rho^{\pi^*}}{d_\mu^{\pi_{\theta_t}}} \right\|_\infty^{-3} \cdot \frac{1}{S \cdot \sqrt{S}} \qquad \text{(by Lemma 9)} \tag{541}$$

$$\leq -\frac{(1-\gamma)^4}{8 \cdot C} \cdot (\mathbb{E}\left[\delta(\theta_t)\right])^3 \cdot \left\| \frac{d_\rho^{\pi^*}}{d_\mu^{\pi_{\theta_t}}} \right\|_\infty^{-3} \cdot \frac{c}{S \cdot \sqrt{S}}, \qquad \text{(by Jensen's inequality)} \tag{542}$$

where

$$c \coloneqq \inf_{t \geq 1} \mathbb{E}\left[\min_s \pi_{\theta_t}(a^*(s)|s)^3\right] \tag{543}$$

$$\geq \inf_{t \geq 1} \left( \mathbb{E}\left[\min_s \pi_{\theta_t}(a^*(s)|s)\right] \right)^3 \qquad \text{(by Jensen's inequality)} \tag{544}$$

$$> 0, \tag{545}$$

and the last inequality is from [2, Lemma 9], since the expected iteration equals the true gradient update, which converges to global optimal policy. According to Eq. (523), we have,

$$\mathbb{E}\left[\delta(\theta_{t+1})\right] - \mathbb{E}\left[\delta(\theta_t)\right] \leq -\frac{(1-\gamma)^7}{8 \cdot C} \cdot (\mathbb{E}\left[\delta(\theta_t)\right])^3 \cdot \left\| \frac{d_\rho^{\pi^*}}{\mu} \right\|_\infty^{-3} \cdot \frac{c}{S \cdot \sqrt{S}}. \tag{546}$$

Denote $\tilde{\delta}(\theta_t) \coloneqq \mathbb{E}\left[\delta(\theta_t)\right]$. Using similar calculations in Eq. (183), we have, for all $t \geq 1$,

$$\frac{1}{\tilde{\delta}(\theta_t)^2} \geq \frac{1}{\tilde{\delta}(\theta_1)^2} + 2 \cdot \sum_{s=1}^{t-1} \frac{1}{\tilde{\delta}(\theta_s)^3} \cdot \left(\tilde{\delta}(\theta_s) - \tilde{\delta}(\theta_{s+1})\right) \tag{547}$$

$$\geq \frac{1}{\tilde{\delta}(\theta_1)^2} + 2 \cdot \sum_{s=1}^{t-1} \frac{1}{\cancel{\tilde{\delta}(\theta_s)^3}} \cdot \frac{(1-\gamma)^7}{8 \cdot C} \cdot \left\|\frac{d_\rho^{\pi^*}}{\mu}\right\|_\infty^{-3} \cdot \frac{c}{S \cdot \sqrt{S}} \cdot \cancel{\tilde{\delta}(\theta_s)^3} \qquad \text{(by Eq. (546))} \tag{548}$$

$$= \frac{1}{\tilde{\delta}(\theta_1)^2} + \frac{(1-\gamma)^7}{4 \cdot C} \cdot \left\|\frac{d_\rho^{\pi^*}}{\mu}\right\|_\infty^{-3} \cdot \frac{c}{S \cdot \sqrt{S}} \cdot (t-1) \tag{549}$$

$$\geq \frac{(1-\gamma)^7}{4 \cdot C} \cdot \left\|\frac{d_\rho^{\pi^*}}{\mu}\right\|_\infty^{-3} \cdot \frac{c}{S \cdot \sqrt{S}} \cdot t \qquad \left(\tilde{\delta}(\theta_1)^2 \leq \frac{1}{(1-\gamma)^2} < \frac{4 \cdot C}{(1-\gamma)^7} \cdot \left\|\frac{d_\rho^{\pi^*}}{\mu}\right\|_\infty^3 \cdot \frac{S \cdot \sqrt{S}}{c}\right) \tag{550}$$

$$= \frac{(1-\gamma)^7}{4} \cdot \left\|\frac{d_\rho^{\pi^*}}{\mu}\right\|_\infty^{-3} \cdot \frac{c}{S^2} \cdot \frac{(1-\gamma) \cdot \gamma}{3 \cdot (1-\gamma) \cdot \gamma + 2 \cdot (C_\infty - (1-\gamma))} \cdot t \qquad \text{(by Eq. (521))} \tag{551}$$

$$= \frac{(1-\gamma)^8}{4} \cdot \left\|\frac{d_\rho^{\pi^*}}{\mu}\right\|_\infty^{-3} \cdot \frac{c}{S^2} \cdot \frac{\gamma}{3 \cdot (1-\gamma) \cdot \gamma + 2 \cdot (C_\infty - (1-\gamma))} \cdot t, \tag{552}$$

which implies that,

$$\mathbb{E}\left[V^*(\mu) - V^{\pi_{\theta_t}}(\mu)\right] \leq \frac{2 \cdot S}{\sqrt{c}} \cdot \frac{\sqrt{3 \cdot (1-\gamma) \cdot \gamma + 2 \cdot (C_\infty - (1-\gamma))}}{(1-\gamma)^4 \cdot \sqrt{\gamma}} \cdot \left\|\frac{d_\rho^{\pi^*}}{\mu}\right\|_\infty^{3/2} \cdot \frac{1}{\sqrt{t}}, \tag{553}$$

where $c$ is from Eq. (543). This leads to the final result,

$$\mathbb{E}\left[V^*(\rho) - V^{\pi_{\theta_t}}(\rho)\right] \leq \frac{1}{1-\gamma} \cdot C'_\infty \cdot \mathbb{E}\left[V^*(\mu) - V^{\pi_{\theta_t}}(\mu)\right] \tag{554}$$

$$\leq \frac{2 \cdot S}{\sqrt{c}} \cdot \frac{\sqrt{3 \cdot (1-\gamma) \cdot \gamma + 2 \cdot (C_\infty - (1-\gamma))}}{(1-\gamma)^5 \cdot \sqrt{\gamma}} \cdot \left\|\frac{d_\rho^{\pi^*}}{\mu}\right\|_\infty^{3/2} \cdot \frac{C'_\infty}{\sqrt{t}}, \tag{555}$$

which implies that $V^*(\rho) - V^{\pi_{\theta_t}}(\rho) \to 0$ as $t \to \infty$ in probability, i.e.,

$$\lim_{t \to \infty} \Pr\left(V^*(\rho) - V^{\pi_{\theta_t}}(\rho) > \epsilon\right) = 0, \tag{556}$$

for all $\epsilon > 0$. $\qquad\square$

### E.3.2 NPG

We show that results similar to Theorem 3 hold in general MDPs, given the following positive reward assumption, which generalizes Assumption 1.

**Assumption 3** (Positive reward). $r(s, a) \in (0, 1]$, for all $(s, a) \in \mathcal{S} \times \mathcal{A}$.

Given Assumption 3, we have, for all policy $\pi_\theta$,

$$Q^{\pi_\theta}(s, a) \in (0, 1/(1-\gamma)]. \tag{557}$$

Replacing $Q^{\pi_{\theta_t}}$ in Algorithm 2 with $\hat{Q}^{\pi_{\theta_t}}$ in Definition 3, we have Algorithm 5.

First, the following result shows that the on-policy stochastic natural PG is unbiased but unbounded, generalizing Lemma 6.

**Lemma 13.** Let $\hat{Q}^{\pi_\theta}(s, \cdot)$ be the IS parallel estimator using on-policy sampling $a(s) \sim \pi_\theta(\cdot|s)$, for all $s$. For NPG, we have,

$$\mathop{\mathbb{E}}_{a(s) \sim \pi_\theta(\cdot|s)}\left[\hat{Q}^{\pi_\theta}\right] = Q^{\pi_\theta}, \tag{558}$$

$$\mathop{\mathbb{E}}_{a(s) \sim \pi_\theta(\cdot|s)}\left[\sum_{(s,a)} \hat{Q}^{\pi_\theta}(s, a)^2\right] = \sum_{(s,a)} \frac{Q^{\pi_\theta}(s, a)^2}{\pi_\theta(a|s)}. \tag{559}$$

---
**Algorithm 5** Natural PG (NPG), on-policy stochastic gradient

---
**Input:** Learning rate $\eta > 0$.
**Output:** Policies $\pi_{\theta_t} = \mathrm{softmax}(\theta_t)$.
Initialize parameter $\theta_1(s, a)$ for all $(s, a) \in \mathcal{S} \times \mathcal{A}$.
**while** $t \geq 1$ **do**
    Sample $a_t(s) \sim \pi_{\theta_t}(\cdot|s)$ for all $s \in \mathcal{S}$.
    $\hat{Q}^{\pi_{\theta_t}}(s, a) \leftarrow \frac{\mathbb{I}\{a_t(s)=a\}}{\pi_{\theta_t}(a|s)} \cdot Q^{\pi_{\theta_t}}(s, a)$.     (by Definition 3)
    $\theta_{t+1} \leftarrow \theta_t + \eta \cdot \hat{Q}^{\pi_{\theta_t}}$.
**end while**

---

*Proof.* **First part.** Unbiased.

According to Definition 3, we have, for all $(s, a) \in \mathcal{S} \times \mathcal{A}$,

$$\underset{a(s) \sim \pi_\theta(\cdot|s)}{\mathbb{E}} \left[ \hat{Q}^{\pi_\theta}(s, a) \right] = \sum_{a(s) \in \mathcal{A}} \pi_\theta(a(s)|s) \cdot \frac{\mathbb{I}\{a(s)=a\}}{\pi_\theta(a|s)} \cdot Q^{\pi_\theta}(s, a) \tag{560}$$

$$= Q^{\pi_\theta}(s, a). \tag{561}$$

**Second part.** Unbounded.

We have, for all $s \in \mathcal{S}$,

$$\sum_{a \in \mathcal{A}} \hat{Q}^{\pi_\theta}(s, a)^2 = \sum_{a \in \mathcal{A}} \frac{\mathbb{I}\{a(s)=a\}}{\pi_\theta(a|s)^2} \cdot Q^{\pi_\theta}(s, a)^2. \tag{562}$$

Then we have,

$$\underset{a(s) \sim \pi_\theta(\cdot|s)}{\mathbb{E}} \left[ \sum_{(s,a)} \hat{Q}^{\pi_\theta}(s, a)^2 \right] = \sum_{s \in \mathcal{S}} \sum_{a(s) \in \mathcal{A}} \pi_\theta(a(s)|s) \cdot \sum_{a \in \mathcal{A}} \frac{\mathbb{I}\{a(s)=a\}}{\pi_\theta(a|s)^2} \cdot Q^{\pi_\theta}(s, a)^2 \tag{563}$$

$$= \sum_{s \in \mathcal{S}} \sum_{a(s) \in \mathcal{A}} \pi_\theta(a(s)|s) \cdot \frac{1}{\pi_\theta(a(s)|s)^2} \cdot Q^{\pi_\theta}(s, a(s))^2 \tag{564}$$

$$= \sum_{(s,a)} \frac{Q^{\pi_\theta}(s, a)^2}{\pi_\theta(a|s)}. \qquad \square$$

The following results generalize Theorem 3 to general MDPs.

**Theorem 14.** *Let $a^*(s)$ be the action that $\pi^*$ selects in state s. Using Algorithm 5, we have:* ***(i)*** *for any state $s \in \mathcal{S}$, with positive probability, $\sum_{a \neq a^*(s)} \pi_{\theta_t}(a|s) \to 1$ as $t \to \infty$;* ***(ii)*** *for all $(s, a) \in \mathcal{S} \times \mathcal{A}$, with positive probability, $\pi_{\theta_t}(a|s) \to 1$, as $t \to \infty$.*

*Proof.* **First part.** For any state $s \in \mathcal{S}$, with positive probability, $\sum_{a \neq a^*(s)} \pi_{\theta_t}(a|s) \to 1$ as $t \to \infty$.

The proof is a generalization of the first part of Theorem 3.

Let Pr denote the probability measure that over the probability space $(\Omega, \mathcal{F})$ that holds all of our random variables. Let $\mathcal{B} = \{a \in \mathcal{A} : a \neq a^*(s)\}$. By abusing notation, for any $\pi(\cdot|s) : \mathcal{A} \to [0, 1]$ map we let $\pi_{\theta_t}(\mathcal{B}|s)$ to stand for $\sum_{a \in \mathcal{B}} \pi_{\theta_t}(a|s)$. Define for $t \geq 1$ the event $\mathcal{B}_t = \{a_t(s) \neq a^*(s)\}(= \{a_t(s) \in \mathcal{B}\})$ and let $\mathcal{E}_t = \mathcal{B}_1 \cap \cdots \cap \mathcal{B}_t$. Thus, $\mathcal{E}_t$ is the event that $a^*$ was not chosen in the first $t$ time steps. Note that $\{\mathcal{E}_t\}_{t \geq 1}$ is a nested sequence and thus, by the monotone convergence theorem,

$$\lim_{t \to \infty} \Pr(\mathcal{E}_t) = \Pr(\mathcal{E}), \tag{565}$$

where $\mathcal{E} = \cap_{t \geq 1} \mathcal{B}_t$. We start with a lower bound on the probability of $\mathcal{E}_t$. The lower bound is stated in a generic form: In particular, let $(b_t)_{t \geq 1}$ be a deterministic sequence which satisfies that for any $t \geq 1$,

$$\mathbb{I}_{\mathcal{E}_{t-1}} \cdot \pi_{\theta_t}(\mathcal{B}|s) \geq \mathbb{I}_{\mathcal{E}_{t-1}} \cdot b_t \qquad \text{holds Pr-almost surely,} \tag{566}$$

where we let $\mathcal{E}_0 = \Omega$ and for an event $\mathcal{E}$, $\mathbb{I}_{\mathcal{E}}$ stands for the characteristic function of $\mathcal{E}$ (i.e., $\mathbb{I}_{\mathcal{E}}(\omega) = 1$ if $\omega \in \mathcal{E}$ and $\mathbb{I}_{\mathcal{E}}(\omega) = 0$, otherwise). We make the following claim:

Claim 1: Under the above assumption, for any $t \geq 1$ it holds that

$$\Pr(\mathcal{E}_t) \geq \prod_{s=1}^{t} b_s. \tag{567}$$

For the proof of this claim let $\mathcal{H}_t$ denote the sequence formed of the first $t$ actions:

$$\mathcal{H}_t := (a_1(s), a_2(s), \cdots, a_t(s)). \tag{568}$$

By definition,

$$\theta_t(s, \cdot) = \mathcal{A}(\theta_1(s, \cdot), a_1(s), Q^{\pi_{\theta_1}}(s, a_1(s)), \cdots, \theta_{t-1}(s, \cdot), a_{t-1}(s), Q^{\pi_{\theta_{t-1}}}(s, a_{t-1}(s))). \tag{569}$$

By our assumption that the $t$th action $a_t(s)$ is chosen from $\pi_{\theta_t}(\cdot|s)$, it follows that $\Pr$ satisfies that for all $a$ and $t \geq 1$,

$$\Pr(a_t(s) = a \mid \mathcal{H}_{t-1}) = \pi_{\theta_t}(a|s) \qquad \text{Pr-almost surely.} \tag{570}$$

We prove the claim by induction on $t$. For $t = 1$, from Eqs. (566) and (570), using that $\mathcal{E}_0 = \Omega$ and $H_0 = ()$, we have that Pr-almost surely,

$$\Pr(\mathcal{E}_1) = \pi_{\theta_1}(\mathcal{B}|s). \tag{571}$$

Suppose the claim holds up to $t - 1$. We have,

$$\Pr(\mathcal{E}_t) = \mathbb{E}[\Pr(\mathcal{E}_t \mid \mathcal{H}_{t-1})] \qquad \text{(by the tower rule)} \tag{572}$$

$$= \mathbb{E}[\mathbb{I}_{\mathcal{E}_{t-1}} \cdot \Pr(\mathcal{B}_t \mid \mathcal{H}_{t-1})] \qquad (\mathcal{E}_{t-1} \text{ is } \mathcal{H}_{t-1}\text{-measurable}) \tag{573}$$

$$= \mathbb{E}[\mathbb{I}_{\mathcal{E}_{t-1}} \cdot \pi_{\theta_t}(\mathcal{B}|s)] \qquad \text{(by Eq. (570))} \tag{574}$$

$$\geq \mathbb{E}[\mathbb{I}_{\mathcal{E}_{t-1}} \cdot b_t] \qquad \text{(by Eq. (566))} \tag{575}$$

$$= b_t \cdot \Pr(\mathcal{E}_{t-1}) \qquad (b_t \text{ is deterministic}) \tag{576}$$

$$= \prod_{s=1}^{t} b_s. \qquad \text{(induction hypothesis)} \tag{577}$$

Now, we claim the following:

Claim 2: A suitable choice for $b_t$ is

$$b_t = \exp\left\{ \frac{-\exp\{\theta_1(s, a^*(s))\}}{(A-1) \cdot \exp\left\{ \frac{\sum_{a \neq a^*(s)} \theta_1(s,a) + \eta \cdot Q_{\min} \cdot (t-1)}{A-1} \right\}} \right\}. \tag{578}$$

Proof of Claim 2: Clearly, it suffices to show that for any sequence $(a_1(s), \ldots, a_{t-1}(s))$ such that $a_k(s) \neq a^*(s)$, $\theta_t(s, \cdot) := \mathcal{A}(\theta_1(s, \cdot), a_1(s), Q^{\pi_{\theta_1}}(s, a_1(s)), \ldots, a_{t-1}(s), Q^{\pi_{\theta_{t-1}}}(s, a_{t-1}(s)))$ is such that $\pi_{\theta_t}(\mathcal{B}|s) \geq b_t$ with $b_t$ as defined in Eq. (578).

We have, for each sub-optimal action $a \neq a^*(s)$,

$$\theta_t(s, a) = \theta_1(s, a) + \eta \cdot \sum_{k=1}^{t-1} \hat{Q}^{\pi_{\theta_k}}(s, a) \qquad \text{(by Algorithm 5)} \tag{579}$$

$$= \theta_1(s, a) + \eta \cdot \sum_{k=1}^{t-1} \frac{\mathbb{I}\{a_k(s) = a\}}{\pi_{\theta_k}(a|s)} \cdot Q^{\pi_{\theta_k}}(s, a) \qquad \text{(by Definition 3)} \tag{580}$$

$$\geq \theta_1(s, a) + \eta \cdot \sum_{k=1}^{t-1} \mathbb{I}\{a_k(s) = a\} \cdot Q^{\pi_{\theta_k}}(s, a) \qquad (\pi_{\theta_k}(a|s) \in (0, 1), \text{ and Eq. (557)}) \tag{581}$$

$$\geq \theta_1(s, a) + \eta \cdot Q_{\min} \cdot \sum_{k=1}^{t-1} \mathbb{I}\{a_k(s) = a\}, \qquad \left( Q_{\min} := \min_{\pi} \min_{(s,a)} Q^{\pi}(s, a) \right) \tag{582}$$

where $Q_{\min} \in (0, 1/(1-\gamma)]$ according to Eq. (557). Then we have,

$$\sum_{a \neq a^*(s)} \exp\{\theta_t(s,a)\} \geq (A-1) \cdot \exp\left\{\frac{\sum_{a \neq a^*} \theta_t(s,a)}{A-1}\right\} \qquad \text{(by Jensen's inequality)} \quad (583)$$

$$\geq (A-1) \cdot \exp\left\{\frac{\sum_{a \neq a^*(s)} \theta_1(s,a) + \eta \cdot Q_{\min} \cdot \sum_{a \neq a^*(s)} \sum_{k=1}^{t-1} \mathbb{I}\{a_k(s) = a\}}{A-1}\right\} \qquad \text{(by Eq. (579))}$$
$$(584)$$

$$= (A-1) \cdot \exp\left\{\frac{\sum_{a \neq a^*(s)} \theta_1(s,a) + \eta \cdot Q_{\min} \cdot (t-1)}{A-1}\right\}. \qquad (a_k(s) \neq a^*(s), \text{ for all } k \in \{1,2,\ldots,t-1\})$$
$$(585)$$

On the other hand, we have,

$$\theta_t(s,a^*(s)) = \theta_1(s,a^*(s)) + \eta \cdot \sum_{k=1}^{t-1} \frac{\mathbb{I}\{a_k(s) = a^*(s)\}}{\pi_{\theta_k}(a^*(s)|s)} \cdot Q^{\pi_{\theta_k}}(s,a^*(s)) \qquad \text{(by Algorithm 5 and Definition 3)}$$
$$(586)$$

$$= \theta_1(s,a^*(s)). \qquad (a_k(s) \neq a^*(s) \text{ for all } k \in \{1,2,\ldots,t-1\}) \qquad (587)$$

Next, we have,

$$\sum_{a \neq a^*(s)} \pi_{\theta_t}(a|s) = 1 - \pi_{\theta_t}(a^*(s)|s) \qquad (588)$$

$$= 1 - \frac{\exp\{\theta_t(s,a^*(s))\}}{\sum_{a \neq a^*(s)} \exp\{\theta_t(s,a)\} + \exp\{\theta_t(s,a^*(s))\}} \qquad (589)$$

$$\geq 1 - \frac{\exp\{\theta_1(s,a^*(s))\}}{(A-1) \cdot \exp\left\{\frac{\sum_{a \neq a^*(s)} \theta_1(s,a) + \eta \cdot Q_{\min} \cdot (t-1)}{A-1}\right\} + \exp\{\theta_1(s,a^*(s))\}} \qquad \text{(by Eqs. (583) and (586))}$$
$$(590)$$

$$\geq \exp\left\{\frac{-1}{\frac{(A-1) \cdot \exp\left\{\frac{\sum_{a \neq a^*(s)} \theta_1(s,a) + \eta \cdot Q_{\min} \cdot (t-1)}{A-1}\right\} + \exp\{\theta_1(s,a^*(s))\}}{\exp\{\theta_1(s,a^*(s))\}} - 1}\right\} \qquad \text{(by Lemma 14)}$$
$$(591)$$

$$= \exp\left\{\frac{-\exp\{\theta_1(s,a^*(s))\}}{(A-1) \cdot \exp\left\{\frac{\sum_{a \neq a^*(s)} \theta_1(s,a) + \eta \cdot Q_{\min} \cdot (t-1)}{A-1}\right\}}\right\} = b_t, \qquad (592)$$

Combining Eq. (565) with the conclusions of Claim 1 and 2 together, we get

$$\Pr(\mathcal{E}) \geq \prod_{t=1}^{\infty} \exp\left\{\frac{-\exp\{\theta_1(s,a^*(s))\}}{(A-1) \cdot \exp\left\{\frac{\sum_{a \neq a^*(s)} \theta_1(s,a) + \eta \cdot Q_{\min} \cdot (t-1)}{A-1}\right\}}\right\} \qquad \text{(by Eq. (588))} \quad (593)$$

$$\geq \exp\left\{-\frac{\exp\{\theta_1(s,a^*(s))\}}{\exp\left\{\frac{\sum_{a \neq a^*(s)} \theta_1(s,a)}{A-1}\right\}} \cdot \frac{\exp\left\{\frac{\eta \cdot Q_{\min}}{A-1}\right\}}{A-1} \cdot \int_{t=0}^{\infty} \frac{1}{\exp\left\{\frac{\eta \cdot Q_{\min} \cdot t}{A-1}\right\}} dt\right\} \qquad (594)$$

$$= \exp\left\{-\frac{\exp\{\theta_1(s,a^*(s))\}}{\exp\left\{\frac{\sum_{a \neq a^*(s)} \theta_1(s,a)}{A-1}\right\}} \cdot \frac{\exp\left\{\frac{\eta \cdot Q_{\min}}{A-1}\right\}}{A-1} \cdot \frac{A-1}{\eta \cdot Q_{\min}}\right\} \qquad (595)$$

$$= \exp\left\{-\frac{\exp\{\theta_1(s,a^*(s))\}}{\exp\left\{\frac{\sum_{a \neq a^*(s)} \theta_1(s,a)}{A-1}\right\}} \cdot \frac{\exp\left\{\frac{\eta \cdot Q_{\min}}{A-1}\right\}}{\eta \cdot Q_{\min}}\right\}. \qquad (596)$$

Note that $Q_{\min} \in \Theta(1)$, $\exp\{\theta_1(s, a^*(s))\} \in \Theta(1)$, $\eta \in \Theta(1)$, $\exp\left\{\frac{\eta \cdot Q_{\min}}{A-1}\right\} \in \Theta(1)$ and,

$$\exp\left\{\frac{\sum_{a \neq a^*(s)} \theta_1(s, a)}{A - 1}\right\} \in \Theta(1). \tag{597}$$

Therefore, we have under state $s \in \mathcal{S}$, " the probability of sampling sub-optimal actions forever using on-policy sampling $a_t(s) \sim \pi_{\theta_t}(\cdot|s)$" is lower bounded by a constant of $\frac{1}{\exp\{\Theta(1)\}} \in \Theta(1)$, which implies that with positive probability $\Theta(1)$, we have $\sum_{a \neq a^*(s)} \pi_{\theta_t}(a|s) \to 1$ as $t \to \infty$.

**Second part.** For all $(s, a) \in \mathcal{S} \times \mathcal{A}$, with positive probability, $\pi_{\theta_t}(a|s) \to 1$, as $t \to \infty$.

The proof is similar to the second part of Theorem 3. Let $\mathcal{B} = \{a\}$. For any $\pi(\cdot|s) : \mathcal{A} \to [0, 1]$ map we let $\pi_{\theta_t}(\mathcal{B}|s)$ to stand for $\pi_{\theta_t}(a|s)$. Define for $t \geq 1$ the event $\mathcal{B}_t = \{a_t(s) = a\}(= \{a_t(s) \in \mathcal{B}\})$ and let $\mathcal{E}_t = \mathcal{B}_1 \cap \cdots \cap \mathcal{B}_t$. Thus, $\mathcal{E}_t$ is the event that $a$ was chosen in the first $t$ time steps. Note that $\{\mathcal{E}_t\}_{t \geq 1}$ is a nested sequence and thus, by the monotone convergence theorem, $\lim_{t \to \infty} \Pr(\mathcal{E}_t) = \Pr(\mathcal{E})$, where $\mathcal{E} = \cap_{t \geq 1} \mathcal{B}_t$. We show that by letting

$$b_t = \exp\left\{\frac{-\sum_{a' \neq a} \exp\{\theta_1(s, a')\}}{\exp\{\theta_1(s, a) + \eta \cdot Q_{\min} \cdot (t - 1)\}}\right\}, \tag{598}$$

we have Eqs. (566) and (567) hold using the arguments in the first part.

It suffices to show that for any sequence $(a_1(s), \ldots, a_{t-1}(s))$ such that $a_k(s) = a$ for all $k \in \{1, 2, \ldots, t - 1\}$, $\theta_t(s, \cdot) := \mathcal{A}(\theta_1(s, \cdot), a_1(s), Q^{\pi_{\theta_1}}(s, a_1(s)), \ldots, a_{t-1}(s), Q^{\pi_{\theta_{t-1}}}(s, a_{t-1}(s)))$ is such that $\pi_{\theta_t}(\mathcal{B}|s) \geq b_t$ with $b_t$ as defined in Eq. (598). Now suppose $a_1(s) = a, a_2(s) = a, \cdots, a_{t-1}(s) = a$. We have,

$$\theta_t(s, a) = \theta_1(s, a) + \eta \cdot \sum_{k=1}^{t-1} \hat{Q}^{\pi_{\theta_k}}(s, a) \qquad \text{(by Algorithm 5)} \tag{599}$$

$$= \theta_1(s, a) + \eta \cdot \sum_{k=1}^{t-1} \frac{\mathbb{I}\{a_k(s) = a\}}{\pi_{\theta_k}(a|s)} \cdot Q^{\pi_{\theta_k}}(s, a) \qquad \text{(by Definition 3)} \tag{600}$$

$$= \theta_1(s, a) + \eta \cdot \sum_{k=1}^{t-1} \frac{Q^{\pi_{\theta_k}}(s, a)}{\pi_{\theta_k}(a|s)} \qquad (a_k(s) = a \text{ for all } k \in \{1, 2, \ldots, t - 1\}) \tag{601}$$

$$\geq \theta_1(s, a) + \eta \cdot \sum_{k=1}^{t-1} Q^{\pi_{\theta_k}}(s, a) \qquad (\pi_{\theta_k}(a|s) \in (0, 1)) \tag{602}$$

$$\geq \theta_1(s, a) + \eta \cdot Q_{\min} \cdot (t - 1). \qquad \left(Q_{\min} := \min_{\pi} \min_{(s, a)} Q^{\pi}(s, a)\right) \tag{603}$$

On the other hand, we have, for any other action $a' \neq a$,

$$\theta_t(s, a') = \theta_1(s, a') + \eta \cdot \sum_{k=1}^{t-1} \frac{\mathbb{I}\{a_k(s) = a'\}}{\pi_{\theta_k}(a'|s)} \cdot Q^{\pi_{\theta_k}}(s, a') \qquad \text{(by Algorithm 5 and Definition 3)} \tag{604}$$

$$= \theta_1(s, a'). \qquad (a_k(s) \neq a' \text{ for all } k \in \{1, 2, \ldots, t - 1\}) \tag{605}$$

Therefore, we have,

$$\pi_{\theta_t}(a|s) = 1 - \sum_{a' \neq a} \pi_{\theta_t}(a'|s) \tag{606}$$

$$= 1 - \frac{\sum_{a' \neq a} \exp\{\theta_t(s, a')\}}{\exp\{\theta_t(s, a)\} + \sum_{a' \neq a} \exp\{\theta_t(s, a')\}} \tag{607}$$

$$\geq 1 - \frac{\sum_{a' \neq a} \exp\{\theta_1(s, a')\}}{\exp\{\theta_1(s, a) + \eta \cdot Q_{\min} \cdot (t-1)\} + \sum_{a' \neq a} \exp\{\theta_1(s, a')\}}. \qquad \text{(by Eqs. (599) and (604))} \tag{608}$$

$$\geq \exp\left\{ \frac{-1}{\frac{\exp\{\theta_1(s,a) + \eta \cdot Q_{\min} \cdot (t-1)\} + \sum_{a' \neq a} \exp\{\theta_1(s,a')\}}{\sum_{a' \neq a} \exp\{\theta_1(s,a')\}} - 1} \right\} \qquad \text{(by Lemma 14)} \tag{609}$$

$$= \exp\left\{ \frac{-\sum_{a' \neq a} \exp\{\theta_1(s, a')\}}{\exp\{\theta_1(s, a) + \eta \cdot Q_{\min} \cdot (t-1)\}} \right\} \tag{610}$$

$$= b_t. \tag{611}$$

Therefore we have,

$$\prod_{t=1}^{\infty} \pi_{\theta_t}(a|s) \geq \prod_{t=1}^{\infty} \exp\left\{ \frac{-\sum_{a' \neq a} \exp\{\theta_1(s, a')\}}{\exp\{\theta_1(s, a) + \eta \cdot Q_{\min} \cdot (t-1)\}} \right\} \qquad \text{(by Eq. (606))} \tag{612}$$

$$= \exp\left\{ -\sum_{a' \neq a} \exp\{\theta_1(s, a')\} \cdot \frac{\exp\{\eta \cdot Q_{\min}\}}{\exp\{\theta_1(s, a)\}} \cdot \sum_{t=1}^{\infty} \frac{1}{\exp\{\eta \cdot Q_{\min} \cdot t\}} \right\} \tag{613}$$

$$\geq \exp\left\{ -\sum_{a' \neq a} \exp\{\theta_1(s, a')\} \cdot \frac{\exp\{\eta \cdot Q_{\min}\}}{\exp\{\theta_1(s, a)\}} \cdot \int_{t=0}^{\infty} \frac{1}{\exp\{\eta \cdot Q_{\min} \cdot t\}} dt \right\} \tag{614}$$

$$= \exp\left\{ -\frac{\exp\{\eta \cdot Q_{\min}\}}{\eta \cdot Q_{\min}} \cdot \frac{\sum_{a' \neq a} \exp\{\theta_1(s, a')\}}{\exp\{\theta_1(s, a)\}} \right\} \tag{615}$$

$$\in \Omega(1), \tag{616}$$

where the last line is due to $Q_{\min} \in \Theta(1)$, $\exp\{\theta_1(s, a)\} \in \Theta(1)$ for all $(s, a) \in \mathcal{S} \times \mathcal{A}$, and $\eta \in \Theta(1)$. With Eq. (612), we have under state $s \in \mathcal{S}$, "the probability of sampling action $a$ forever using on-policy sampling $a_t(s) \sim \pi_{\theta_t}(\cdot|s)$" is lower bounded by a constant of $\Omega(1)$. Therefore, for all $(s, a) \in \mathcal{S} \times \mathcal{A}$, with positive probability $\Omega(1)$, $\pi_{\theta_t}(a|s) \to 1$, as $t \to \infty$. $\qquad\square$

### E.3.3 GNPG

Replacing $Q^{\pi_{\theta_t}}$ in Algorithm 3 with $\hat{Q}^{\pi_{\theta_t}}$ in Definition 3, we have Algorithm 6.

---
**Algorithm 6** Geometry-award normalized PG (GNPG), on-policy stochastic gradient
---
**Input:** Learning rate $\eta > 0$.
**Output:** Policies $\pi_{\theta_t} = \text{softmax}(\theta_t)$.
Initialize parameter $\theta_1(s, a)$ for all $(s, a) \in \mathcal{S} \times \mathcal{A}$.
**while** $t \geq 1$ **do**
    Sample $a_t(s) \sim \pi_{\theta_t}(\cdot|s)$ for all $s \in \mathcal{S}$.
    $\hat{Q}^{\pi_{\theta_t}}(s, a) \leftarrow \frac{\mathbb{I}\{a_t(s)=a\}}{\pi_{\theta_t}(a|s)} \cdot Q^{\pi_{\theta_t}}(s, a).$     (by Definition 3)
    $\hat{g}_t(s, \cdot) \leftarrow \frac{1}{1-\gamma} \cdot d_\mu^{\pi_{\theta_t}}(s) \cdot \left[ \sum_a \frac{\partial \pi_{\theta_t}(a|s)}{\partial \theta_t(s, \cdot)} \cdot \hat{Q}^{\pi_{\theta_t}}(s, a) \right].$
    $\theta_{t+1} \leftarrow \theta_t + \eta \cdot \hat{g}_t / \|\hat{g}_t\|_2.$
**end while**

---

The following result generalizes Theorem 4. The complication of the proofs is from the difference between NPG of Algorithm 5 and GNPG of Algorithm 6. In NPG, only $Q^{\pi_{\theta_t}}(s, \cdot)$ appears in the update of $\theta_t(s, \cdot)$. While in GNPG, other states also contribute to the update of $\theta_t(s, \cdot)$ through the normalization factor $\|\hat{g}_t\|_2$.

**Theorem 15.** *Using Algorithm 6, for all $(s, a) \in \mathcal{S} \times \mathcal{A}$, with positive probability, we have, $\pi_{\theta_t}(a|s) \to 1$ as $t \to \infty$.*

*Proof.* Consider any deterministic policy $\bar{\pi} : s \mapsto \bar{\pi}(s)$, i.e.,

$$\bar{\pi}(a|s) = \begin{cases} 1, & \text{if } a = \bar{\pi}(s), \\ 0. & \text{otherwise} \end{cases} \tag{617}$$

We then make the following three claims.

**(i)** Using any fixed deterministic policy $\bar{\pi}$ to sample, i.e., $a_t(s) = \bar{\pi}(s)$, for all $s \in \mathcal{S}$ and for all $t \geq 1$, and using GNPG update to generate a deterministic sequence $\{\tilde{\theta}_t\}_{t \geq 1}$, we have $\pi_{\tilde{\theta}_t}(a|s) \to \bar{\pi}(a|s)$ as $t \to \infty$ for all $(s, a) \in \mathcal{S} \times \mathcal{A}$.

**(ii)** The speed of $\pi_{\tilde{\theta}_t}$ approaches $\bar{\pi}$ is exponential, i.e., $1 - \pi_{\tilde{\theta}_t}(\bar{\pi}(s)|s) \in O(e^{-c \cdot t})$ for some $c > 0$, for all $t \geq 1$ and all $s \in \mathcal{S}$.

**(iii)** Using on-policy GNPG of Algorithm 6, i.e., $a_t(s) \sim \pi_{\theta_t}(\cdot|s)$, for all $(s, a) \in \mathcal{S} \times \mathcal{A}$, with positive probability, $\pi_{\theta_t}(a|s) \to 1$ as $t \to \infty$ (i.e., the main claim).

**First part. (i).** We show that with the fixed sampling $a_t(s) = \bar{\pi}(s)$, the following holds,

$$\tilde{\theta}_{t+1}(s, \bar{\pi}(s)) > \tilde{\theta}_t(s, \bar{\pi}(s)), \text{ and} \tag{618}$$

$$\tilde{\theta}_{t+1}(s, a') < \tilde{\theta}_t(s, a'), \text{ for all } a' \neq \bar{\pi}(s) \tag{619}$$

which according to $\pi_{\tilde{\theta}}(\cdot|s) = \mathrm{softmax}(\tilde{\theta}(s, \cdot))$ implies that, for all $t \geq 1$,

$$\pi_{\tilde{\theta}_{t+1}}(\bar{\pi}(s)|s) > \pi_{\tilde{\theta}_t}(\bar{\pi}(s)|s). \tag{620}$$

Since $\pi_{\tilde{\theta}_t}(\bar{\pi}(s)|s) \leq 1$, according to the monotone convergence, $\pi_{\tilde{\theta}_t}(\bar{\pi}(s)|s) \to c > 0$ as $t \to \infty$. And $c$ has to be 1, otherwise due to Eq. (618) the probability of action $\bar{\pi}(s)$ can be further improved, which is a contradiction with convergence. Next, we show that Eq. (618) holds.

By the assumption of using fixed sampling, we have, at iteration $t \geq 1$, $a_t(s) = \bar{\pi}(s)$. Then we have,

$$\hat{g}_t(s, \bar{\pi}(s)) = \frac{1}{1 - \gamma} \cdot d_\mu^{\pi_{\tilde{\theta}_t}}(s) \cdot \pi_{\tilde{\theta}_t}(\bar{\pi}(s)|s) \cdot \left[ \frac{\mathbb{I}\{a_t(s) = \bar{\pi}(s)\}}{\pi_{\tilde{\theta}_t}(a_t(s)|s)} \cdot Q^{\pi_{\tilde{\theta}_t}}(s, a_t(s)) - \sum_{a' \in \mathcal{A}} \mathbb{I}\{a_t(s) = a'\} \cdot Q^{\pi_{\tilde{\theta}_t}}(s, a') \right] \tag{621}$$

$$= \frac{1}{1 - \gamma} \cdot d_\mu^{\pi_{\tilde{\theta}_t}}(s) \cdot \left(1 - \pi_{\tilde{\theta}_t}(\bar{\pi}(s)|s)\right) \cdot Q^{\pi_{\tilde{\theta}_t}}(s, \bar{\pi}(s)) \qquad (a_t(s) = \bar{\pi}(s)) \tag{622}$$

$$> 0, \qquad \left( d_\mu^{\pi_{\tilde{\theta}_t}}(s) > 0, \ \pi_{\tilde{\theta}_t}(\bar{\pi}(s)|s) \in (0, 1), \ Q^{\pi_{\tilde{\theta}_t}}(s, \bar{\pi}(s)) \in (0, 1/(1 - \gamma)] \right) \tag{623}$$

where the last inequality is from Eq. (523), Assumption 2, and Eq. (557). On the other hand, for any other action $a' \neq \bar{\pi}(s)$, we have,

$$\hat{g}_t(s, a') = -\frac{1}{1 - \gamma} \cdot d_\mu^{\pi_{\tilde{\theta}_t}}(s) \cdot \pi_{\tilde{\theta}_t}(a'|s) \cdot Q^{\pi_{\tilde{\theta}_t}}(s, \bar{\pi}(s)) < 0. \tag{624}$$

Therefore, according to the GNPG update, i.e.,

$$\tilde{\theta}_{t+1}(s, \cdot) \leftarrow \tilde{\theta}_t(s, \cdot) + \eta \cdot \hat{g}_t(s, \cdot)/\|\hat{g}_t\|_2, \tag{625}$$

we have Eq. (618) holds.

**Second part. (ii).** We calculate the progress $\tilde{\theta}_t(s, \bar{\pi}(s))$ can get. We have,

$$\|\hat{g}_t(s, \cdot)\|_2^2 = \hat{g}_t(s, \bar{\pi}(s))^2 + \sum_{a' \neq \bar{\pi}(s)} \hat{g}_t(s, a')^2 \tag{626}$$

$$\leq \frac{1}{(1 - \gamma)^2} \cdot d_\mu^{\pi_{\tilde{\theta}_t}}(s)^2 \cdot Q^{\pi_{\tilde{\theta}_t}}(s, \bar{\pi}(s))^2 \cdot 2 \cdot \left(1 - \pi_{\tilde{\theta}_t}(\bar{\pi}(s)|s)\right)^2. \qquad (\|x\|_2 \leq \|x\|_1) \tag{627}$$

At iteration $t \geq 1$, we define $\bar{s}_t$ as follows,

$$\bar{s}_t = \arg\max_{s' \in \mathcal{S}} \left(1 - \pi_{\tilde{\theta}_t}(\bar{\pi}(s')|s')\right) \cdot Q^{\pi_{\tilde{\theta}_t}}(s', \bar{\pi}(s')). \tag{628}$$

According to Eq. (626), we have,

$$\|\hat{g}_t\|_2^2 = \|\hat{g}_t(\bar{s}_t, \cdot)\|_2^2 + \sum_{s' \neq \bar{s}_t} \|\hat{g}_t(s', \cdot)\|_2^2 \tag{629}$$

$$\leq \frac{1}{(1-\gamma)^2} \cdot d_\mu^{\pi_{\tilde{\theta}_t}}(\bar{s}_t)^2 \cdot Q^{\pi_{\tilde{\theta}_t}}(\bar{s}_t, \bar{\pi}(\bar{s}_t))^2 \cdot 2 \cdot \left(1 - \pi_{\tilde{\theta}_t}(\bar{\pi}(\bar{s}_t)|\bar{s}_t)\right)^2 \tag{630}$$

$$+ \sum_{s' \neq \bar{s}_t} \frac{1}{(1-\gamma)^2} \cdot d_\mu^{\pi_{\tilde{\theta}_t}}(s')^2 \cdot Q^{\pi_{\tilde{\theta}_t}}(s', \bar{\pi}(s'))^2 \cdot 2 \cdot \left(1 - \pi_{\tilde{\theta}_t}(\bar{\pi}(s')|s')\right)^2 \quad (\|x\|_2 \leq \|x\|_1) \tag{631}$$

$$\leq \frac{1}{(1-\gamma)^2} \cdot Q^{\pi_{\tilde{\theta}_t}}(\bar{s}_t, \bar{\pi}(\bar{s}_t))^2 \cdot 2 \cdot \left(1 - \pi_{\tilde{\theta}_t}(\bar{\pi}(\bar{s}_t)|\bar{s}_t)\right)^2 \cdot \sum_{s' \in \mathcal{S}} d_\mu^{\pi_{\tilde{\theta}_t}}(s')^2 \quad (\text{by Eq. (628)}) \tag{632}$$

$$\leq \frac{1}{(1-\gamma)^2} \cdot Q^{\pi_{\tilde{\theta}_t}}(\bar{s}_t, \bar{\pi}(\bar{s}_t))^2 \cdot 2 \cdot \left(1 - \pi_{\tilde{\theta}_t}(\bar{\pi}(\bar{s}_t)|\bar{s}_t)\right)^2, \quad (\|x\|_2 \leq \|x\|_1) \tag{633}$$

which implies that,

$$\|\hat{g}_t\|_2 \leq \frac{\sqrt{2}}{1-\gamma} \cdot \left(1 - \pi_{\tilde{\theta}_t}(\bar{\pi}(\bar{s}_t)|\bar{s}_t)\right) \cdot Q^{\pi_{\tilde{\theta}_t}}(\bar{s}_t, \bar{\pi}(\bar{s}_t)). \tag{634}$$

According to Eqs. (523), (621) and (629), we have,

$$\tilde{\theta}_{t+1}(s, \bar{\pi}(\bar{s}_t)) \leftarrow \tilde{\theta}_t(s, \bar{\pi}(\bar{s}_t)) + \eta \cdot \frac{\hat{g}_t(\bar{s}_t, \bar{\pi}(\bar{s}_t))}{\|\hat{g}_t\|_2} \tag{635}$$

$$\geq \tilde{\theta}_t(s, \bar{\pi}(\bar{s}_t)) + \eta \cdot \frac{1-\gamma}{\sqrt{2}}, \tag{636}$$

which implies that,

$$1 - \pi_{\tilde{\theta}_{t+1}}(\bar{\pi}(\bar{s}_t)|\bar{s}_t) = \frac{\sum_{a' \neq \bar{\pi}(\bar{s}_t)} \exp\{\tilde{\theta}_{t+1}(\bar{s}_t, a')\}}{\exp\{\tilde{\theta}_{t+1}(\bar{s}_t, \bar{\pi}(\bar{s}_t))\} + \sum_{a' \neq \bar{\pi}(\bar{s}_t)} \exp\{\tilde{\theta}_{t+1}(\bar{s}_t, a')\}} \tag{637}$$

$$\leq \frac{\sum_{a' \neq \bar{\pi}(\bar{s}_t)} \exp\{\tilde{\theta}_t(\bar{s}_t, a')\}}{\exp\{\tilde{\theta}_{t+1}(\bar{s}_t, \bar{\pi}(\bar{s}_t))\} + \sum_{a' \neq \bar{\pi}(\bar{s}_t)} \exp\{\tilde{\theta}_t(\bar{s}_t, a')\}} \quad (\text{by Eq. (624)}) \tag{638}$$

$$\leq \frac{\sum_{a' \neq \bar{\pi}(\bar{s}_t)} \exp\{\tilde{\theta}_t(\bar{s}_t, a')\}}{\exp\left\{\frac{\eta \cdot (1-\gamma)}{\sqrt{2}}\right\} \cdot \exp\{\tilde{\theta}_{t+1}(\bar{s}_t, \bar{\pi}(\bar{s}_t))\} + \sum_{a' \neq \bar{\pi}(\bar{s}_t)} \exp\{\tilde{\theta}_t(\bar{s}_t, a')\}} \tag{639}$$

$$= \frac{\sum_{a' \neq \bar{\pi}(\bar{s}_t)} \pi_{\tilde{\theta}_t}(a'|\bar{s}_t)}{\left(\exp\left\{\frac{\eta \cdot (1-\gamma)}{\sqrt{2}}\right\} - 1\right) \cdot \pi_{\tilde{\theta}_t}(\bar{\pi}(\bar{s}_t)|\bar{s}_t) + 1} \tag{640}$$

$$= \frac{1}{\left(\exp\left\{\frac{\eta \cdot (1-\gamma)}{\sqrt{2}}\right\} - 1\right) \cdot \pi_{\tilde{\theta}_t}(\bar{\pi}(\bar{s}_t)|\bar{s}_t) + 1} \cdot \left(1 - \pi_{\tilde{\theta}_t}(\bar{\pi}(\bar{s}_t)|\bar{s}_t)\right) \tag{641}$$

$$\leq \frac{1}{\left(\exp\left\{\frac{\eta \cdot (1-\gamma)}{\sqrt{2}}\right\} - 1\right) \cdot \pi_{\tilde{\theta}_1}(\bar{\pi}(\bar{s}_t)|\bar{s}_t) + 1} \cdot \left(1 - \pi_{\tilde{\theta}_t}(\bar{\pi}(\bar{s}_t)|\bar{s}_t)\right), \quad (\text{by Eq. (620)}) \tag{642}$$

which means that after one update, 1 minus the probability of the sampled action $\bar{\pi}(\bar{s}_t)$ under $\bar{s}_t$ (by Eq. (628), there must be at least one such state) is reduced by a constant. As a consequence, if a state $s$ is the $\arg\max$ in Eq. (628) for $O(t)$ times, then we have,

$$1 - \pi_{\tilde{\theta}_t}(\bar{\pi}(s)|s) \in O(e^{-c \cdot t}), \tag{643}$$

where $c > 0$. Now we argue by contradiction that Eq. (643) holds for all state $s \in \mathcal{S}$. Suppose there is at least one state $s'$ which has been selected as the $\arg\max$ in Eq. (628) for only $o(t)$ times during the first $t$ iterations. Then for $s'$, the term $1 - \pi_{\tilde{\theta}_t}(\bar{\pi}(s')|s')$ in the r.h.s. of Eq. (628) is at order of $\omega(e^{-c \cdot t})$, dominating the corresponding terms of other actions, which are at order of $O(e^{-c \cdot t})$ according to Eq. (643). This makes the other actions cannot be selected as the $\arg\max$ in Eq. (628) for $O(t)$ times.

Therefore, there does not exist such an state $s'$ which has been selected as the $\arg\max$ in Eq. (628) for only $o(t)$ times. And for all state $s \in \mathcal{S}$, we have Eq. (643).

**Third part. (iii).** Using similar arguments in first part of Theorem 4, the probability of $\bar{\pi}(s)$ is sampled forever if we run GNPG with on-policy sampling $a_t(s) \sim \pi_{\theta_t}(\cdot|s)$ is lower bounded by $\prod_{t=1}^{\infty} \pi_{\tilde{\theta}_t}(\bar{\pi}(s)|s)$, where $\{\tilde{\theta}_t\}_{t \geq 1}$ is the deterministic sequence generated by fixed sampling with the deterministic policy $\bar{\pi}$. Using similar arguments in the second part of Theorem 4 and according to Eq. (643), we have $\prod_{t=1}^{\infty} \pi_{\tilde{\theta}_t}(\bar{\pi}(s)|s) > 0$, which means that with positive probability, $\bar{\pi}(s)$ will be sampled for all $t \geq 1$ using on-policy sampling. This implies that for all $(s, a) \in \mathcal{S} \times \mathcal{A}$, with positive probability, $\pi_{\theta_t}(a|s) \to 1$ as $t \to \infty$. $\qquad\square$

## E.4 Committal Rates

The following Definition 4 generalizes Definition 2. The difference is we now fix the sampling to be using one fixed deterministic policy to sample forever, and then define the committal rate for all the deterministic state action pairs, generalizing the proof idea of Theorem 15.

**Definition 4** (Committal Rate). *Fix a reward function $r \in (0, 1]^{S \times A}$ and an initial parameter vector $\theta_1 \in \mathbb{R}^{S \times A}$. Consider a policy optimization algorithm $\mathcal{A}$. Consider any deterministic policy $\bar{\pi} : s \mapsto \bar{\pi}(s)$. Under state $s$, let action $\bar{\pi}(s)$ be the sampled action **forever** after initialization and let $\theta_t$ be the resulting parameter vector obtained by using $\mathcal{A}$ on the first $t$ observations. The committal rate of algorithm $\mathcal{A}$ on the deterministic policy $\bar{\pi}$ (given $r$ and $\theta_1$) is then defined as*

$$\kappa(\mathcal{A}, \bar{\pi}) = \min_{s \in \mathcal{S}} \sup \left\{ \alpha \geq 0 : \limsup_{t \to \infty} t^{\alpha} \cdot [1 - \pi_{\theta_t}(\bar{\pi}(s)|s)] < \infty \right\}. \tag{644}$$

The following Theorem 16 generalizes Theorem 5.

**Theorem 16.** *Consider a policy optimization method $\mathcal{A}$, together with $r \in (0, 1]^{S \times A}$ and an initial parameter vector $\theta_1 \in \mathbb{R}^{S \times A}$. Then,*

$$\max_{\bar{\pi}: \, sub\text{-}optimal \, deterministic, \, \pi_{\theta_1}(a|s) > 0} \kappa(\mathcal{A}, \bar{\pi}) \leq 1 \tag{645}$$

*is a necessary condition for ensuring the almost sure convergence of the policies obtained using $\mathcal{A}$ and online sampling to the global optimum starting from $\theta_1$, where $\bar{\pi}$ under the $\max$ is for all sub-optimal deterministic policies, i.e., $\bar{\pi}(s) \neq a^*(s)$ for at least one $s \in \mathcal{S}$.*

*Proof.* The proof is a extension of Theorems 5 and 15.

Let $\{\tilde{\theta}_t\}_{t \geq 1}$ be generated by using a sub-optimal deterministic policy $\bar{\pi} : s \mapsto \bar{\pi}(s)$ to sample, and using the algorithm $\mathcal{A}$ to update with the sampled actions. And let $\{\theta_t\}_{t \geq 1}$ be generated using on-policy sampling and updating with the algorithm $\mathcal{A}$.

Using similar arguments in the first part of Theorem 5, the probability of $\bar{\pi}(s)$ is sampled forever under state $s$ using on-policy sampling is lower bounded by $\prod_{t=1}^{\infty} \pi_{\tilde{\theta}_t}(\bar{\pi}(s)|s)$.

Suppose the committal rate of one sub-optimal deterministic policy is strictly larger than 1, i.e., $\kappa(\mathcal{A}, \bar{\pi}) > 1$, where $\bar{\pi}(s) \neq a^*(s)$ for at least one state $s \in \mathcal{S}$. According to similar arguments in the first and second parts of Theorem 15, we have $\prod_{t=1}^{\infty} \pi_{\tilde{\theta}_t}(\bar{\pi}(s)|s) > 0$.

Combining the two arguments, we have the the probability of $\bar{\pi}(s)$ is sampled forever under state $s$ using on-policy sampling is positive, which implies that for at least one state $s \in \mathcal{S}$, $\pi_{\theta_t}(\cdot|s)$ will converge to some sub-optimal deterministic policies. $\qquad\square$

## E.5 Geometry-Convergence Trade-off

First, we generalize the definition of *optimality-smart* from the main paper. A policy optimization method is said to be optimality-smart if for any $t \geq 1$, $\pi_{\tilde{\theta}_t}(a^*(s)|s) \geq \pi_{\theta_t}(a^*(s)|s)$ holds where $\tilde{\theta}_t$ is the parameter vector obtained when $a^*(s)$ is chosen in every time step, starting at $\theta_1$, while $\theta_t$ is *any* parameter vector that can be obtained with $t$ updates (regardless of the action sequence chosen), but also starting from $\theta_1$.

With this definition, results similar to Proposition 5 hold by using similar arguments, i.e., if $a_t(s) = a^*(s)$, we have $\pi_{\theta_{t+1}}(a^*(s)|s) \geq \pi_{\theta_t}(a^*(s)|s)$, and otherwise if $a_t(s) \neq a^*(s)$, we have $\pi_{\theta_{t+1}}(a^*(s)|s) \leq \pi_{\theta_t}(a^*(s)|s)$.

Next, the following result generalizes Theorem 9.

**Theorem 17.** *Let $\mathcal{A}$ be optimality-smart and pick a MDP instance. If $\mathcal{A}$ together with on-policy sampling leads to $\{\theta_t\}_{t\geq 1}$ such that $\{V^{\pi_{\theta_t}}(\rho)\}_{t\geq 1}$ (where $\min_{s\in\mathcal{S}} \rho(s) > 0$) converges to a globally optimal policy at a rate $O(1/t^\alpha)$ with positive probability, for $\alpha > 0$, then we have $\kappa(\mathcal{A}, \pi^*) \geq \alpha$, where $\pi^*$ is the global optimal deterministic policy.*

*Proof.* Denote $\Delta^*(s,a) = Q^*(s, a^*(s)) - Q^*(s,a)$, $\Delta^*(s) = \min_{a\neq a^*(s)} \Delta^*(s,a)$, and $\Delta^* = \min_{s\in\mathcal{S}} \Delta^*(s) > 0$ as the optimal value gap of the MDP. According to Lemma 20, we have,

$$V^*(\rho) - V^{\pi_{\theta_t}}(\rho) = \frac{1}{1-\gamma} \cdot \sum_s d_\rho^{\pi_{\theta_t}}(s) \cdot \sum_a (\pi^*(a|s) - \pi_{\theta_t}(a|s)) \cdot Q^*(s,a) \tag{646}$$

$$= \frac{1}{1-\gamma} \cdot \sum_s d_\rho^{\pi_{\theta_t}}(s) \cdot \left[ \sum_a \pi_{\theta_t}(a|s) \cdot Q^*(s,a^*(s)) - \sum_a \pi_{\theta_t}(a|s) \cdot Q^*(s,a) \right] \tag{647}$$

$$= \frac{1}{1-\gamma} \cdot \sum_s d_\rho^{\pi_{\theta_t}}(s) \cdot \left[ \sum_{a\neq a^*(s)} \pi_{\theta_t}(a|s) \cdot Q^*(s,a^*(s)) - \sum_{a\neq a^*(s)} \pi_{\theta_t}(a|s) \cdot Q^*(s,a) \right] \tag{648}$$

$$= \frac{1}{1-\gamma} \cdot \sum_s d_\rho^{\pi_{\theta_t}}(s) \cdot \left[ \sum_{a\neq a^*(s)} \pi_{\theta_t}(a|s) \cdot \Delta^*(s,a) \right] \tag{649}$$

$$\geq \frac{1}{1-\gamma} \cdot \sum_s d_\rho^{\pi_{\theta_t}}(s) \cdot \left[ \sum_{a\neq a^*(s)} \pi_{\theta_t}(a|s) \right] \cdot \Delta^*(s) \tag{650}$$

$$\geq \frac{1}{1-\gamma} \cdot \sum_s d_\rho^{\pi_{\theta_t}}(s) \cdot \left[ \sum_{a\neq a^*(s)} \pi_{\theta_t}(a|s) \right] \cdot \Delta^*. \qquad (\Delta^* \leq \Delta^*(s)) \tag{651}$$

Therefore we have,

$$V^*(\rho) - V^{\pi_{\theta_t}}(\rho) \geq \sum_s \rho(s) \cdot \left[ \sum_{a\neq a^*(s)} \pi_{\theta_t}(a|s) \right] \cdot \Delta^* \qquad \text{(by Eqs. (523) and (646))} \tag{652}$$

$$= \sum_s \rho(s) \cdot (1 - \pi_{\theta_t}(a^*(s)|s)) \cdot \Delta^* \tag{653}$$

$$\geq \min_{s\in\mathcal{S}} \rho(s) \cdot (1 - \pi_{\theta_t}(a^*(s)|s)) \cdot \Delta^*. \tag{654}$$

For $\alpha > 0$ let $\mathcal{E}_\alpha$ be the event when for all $t \geq 1$,

$$V^*(\rho) - V^{\pi_{\theta_t}}(\rho) \leq \frac{C}{t^\alpha}. \tag{655}$$

By our assumption, there exists $\alpha > 0$ such that $\Pr(\mathcal{E}_\alpha) > 0$. On this event, for any $t \geq 1$, we have,

$$t^\alpha \cdot (1 - \pi_{\theta_t}(a^*(s)|s)) \leq \frac{1}{\min_{s\in\mathcal{S}} \rho(s)} \cdot \frac{t^\alpha}{\Delta^*} \cdot (V^*(\rho) - V^{\pi_{\theta_t}}(\rho)) \qquad \text{(by Eq. (652))} \tag{656}$$

$$\leq \frac{1}{\min_{s\in\mathcal{S}} \rho(s)} \cdot \frac{C}{\Delta^*}. \qquad \text{(by Eq. (655))} \tag{657}$$

Let $\{\tilde{\theta}_t\}_{t\geq 1}$ with $\tilde{\theta}_1 = \theta_1$ be the sequence obtained by using $\mathcal{A}$ with fixed sampling on $r$, such that $a_t(s) = a^*(s)$ for all $t \geq 1$. Since, by the assumption, $\mathcal{A}$ is optimality-smart, we have $\pi_{\tilde{\theta}_t}(a^*(s)|s) \geq \pi_{\theta_t}(a^*(s)|s)$. Then, on $\mathcal{E}_\alpha$, for any $t \geq 1$

$$t^\alpha \cdot \left(1 - \pi_{\tilde{\theta}_t}(a^*(s)|s)\right) \leq t^\alpha \cdot \left(1 - \pi_{\theta_t}(a^*(s)|s)\right) \tag{658}$$

$$\leq \frac{1}{\min_{s\in\mathcal{S}} \rho(s)} \cdot \frac{C}{\Delta}, \qquad \text{(by Eq. (656))} . \tag{659}$$

Since $\mathbb{P}(\mathcal{E}_\alpha) > 0$ and $t^\alpha \cdot \left(1 - \pi_{\tilde{\theta}_t}(a^*(s)|s)\right)$ is non-random, it follows that for any $t \geq 1$, $t^\alpha \cdot \left(1 - \pi_{\tilde{\theta}_t}(a^*(s)|s)\right) \leq C/\Delta$, which, by Definition 4, means that $\kappa(\mathcal{A}, \pi^*) \geq \alpha$, where $\pi^*$ is the global optimal deterministic policy. $\qquad\square$

# F   Miscellaneous Extra Supporting Results

**Lemma 14.** *We have, for all $x \in (0, 1)$,*

$$1 - x \geq e^{-1/(1/x-1)}. \tag{660}$$

*Proof.* See the proof in [12, Proposition 1]. We include a proof for completeness.

We have, for all $x \in (0, 1)$,

$$1 - x = \exp\left\{\log\left(1 - x\right)\right\} \tag{661}$$

$$\geq \exp\left\{1 - e^{-\log(1-x)}\right\} \qquad \left(y \geq 1 - e^{-y}\right) \tag{662}$$

$$= \exp\left\{\frac{-1}{1/x - 1}\right\}. \qquad\square \tag{ }$$

**Lemma 15.** *Let $\alpha > 0$. We have,*

**(i)** *if $\alpha \in (1, \infty)$, then for all $C > 0$,*

$$\sum_{t=1}^\infty \frac{C}{t^\alpha} < \infty, \tag{663}$$

*which means the series $\sum_{t=1}^\infty \frac{C}{t^\alpha}$ converges to a finite value.*

**(ii)** *if $\alpha \in (0, 1]$, then for all $C > 0$,*

$$\sum_{t=1}^\infty \frac{C}{t^\alpha} = \infty, \tag{664}$$

*which means the series $\sum_{t=1}^\infty \frac{C}{t^\alpha}$ diverges to positive infinity.*

**(iii)** *for all $C > 0$, $C' > 0$,*

$$\sum_{t=1}^\infty \frac{C}{\exp\{C' \cdot t\}} < \infty, \tag{665}$$

*which means the series $\sum_{t=1}^\infty \frac{C}{\exp\{C'\cdot t\}}$ converges to a finite value.*

*Proof.* It is easy to verify the results by calculating integrals. We include a proof for completeness.

**First part.** We have, for all $\alpha \in (1, \infty)$ and $C > 0$,

$$\sum_{t=1}^\infty \frac{C}{t^\alpha} \leq C \cdot \left(1 + \int_{t=1}^\infty \frac{1}{t^\alpha} dt\right) \tag{666}$$

$$= \frac{C \cdot \alpha}{\alpha - 1}. \tag{667}$$

**Second part.** We have, for all $\alpha \in (0, 1)$, $C > 0$, and $T \geq 1$,

$$\sum_{t=1}^{T} \frac{C}{t^\alpha} \geq \int_{t=1}^{T+1} \frac{C}{t^\alpha} dt \tag{668}$$

$$= \frac{C \cdot \left((T+1)^{1-\alpha} - 1\right)}{1 - \alpha}. \tag{669}$$

Similarly, for $\alpha = 1$,

$$\sum_{t=1}^{T} \frac{C}{t} \geq \int_{t=1}^{T+1} \frac{C}{t} dt \tag{670}$$

$$= C \cdot \log(T + 1). \tag{671}$$

Therefore, the partial sum approaches to positive infinity as $T \to \infty$.

**Third part.** We have, for all $C > 0$ and $C' > 0$,

$$\sum_{t=1}^{\infty} \frac{C}{\exp\{C' \cdot t\}} \leq \int_{t=0}^{\infty} \frac{C}{\exp\{C' \cdot t\}} \tag{672}$$

$$= \frac{C}{C'}. \qquad \square$$

**Lemma 16.** *Let $u_t \in (0, 1)$ for all $t \geq 1$. The infinite product $\prod_{t=1}^{\infty} (1 - u_t)$ converges to a positive value if and only if the series $\sum_{t=1}^{\infty} u_t$ converges to a finite value.*

*Proof.* See Knopp [31, p. 220]. We include a proof for completeness.

Define the following partial products and partial sums,

$$p_T := \prod_{t=1}^{T} (1 - u_t), \tag{673}$$

$$s_T := \sum_{t=1}^{T} u_t. \tag{674}$$

Since $p_T$ is monotonically decreasing and non-negative, the infinite product converges to positive values, i.e.,

$$\prod_{t=1}^{\infty} (1 - u_t) = \lim_{T \to \infty} \prod_{t=1}^{T} (1 - u_t) = \lim_{T \to \infty} p_T > 0, \tag{675}$$

if and only if $p_T$ is lower bounded away from zero (boundedness convergence criterion for monotone sequence) [31, p. 80].

Similarly, since $s_T$ is monotonically increasing, the series converges to finite values, i.e.,

$$\sum_{t=1}^{\infty} u_t = \lim_{T \to \infty} \sum_{t=1}^{T} u_t = \lim_{T \to \infty} s_T < \infty, \tag{676}$$

if and only if $s_T$ is upper bounded.

**First part.** $\prod_{t=1}^{\infty} (1 - u_t)$ converges to a positive value only if $\sum_{t=1}^{\infty} u_t$ converges to a finite value.

Suppose $\prod_{t=1}^{\infty} (1 - u_t)$ converges to a positive value. We have, for all $T \geq 1$,

$$q_T \geq q > 0. \tag{677}$$

Then we have,

$$q \leq q_T \tag{678}$$

$$= \exp\left\{ \log\left( \prod_{t=1}^{T} (1 - u_t) \right) \right\} \tag{679}$$

$$= \exp\left\{ \sum_{t=1}^{T} \log(1 - u_t) \right\} \tag{680}$$

$$\leq \exp\left\{ -\sum_{t=1}^{T} u_t \right\} \qquad (\log(1-x) < -x) \tag{681}$$

$$= \exp\{-s_T\}, \tag{682}$$

which implies that,

$$s_T \leq -\log q < \infty. \tag{683}$$

Therefore, we have $\sum_{t=1}^{\infty} u_t$ converges to a finite value.

**Second part.** $\prod_{t=1}^{\infty} (1 - u_t)$ converges to a positive value if $\sum_{t=1}^{\infty} u_t$ converges to a finite value.

Suppose $\sum_{t=1}^{\infty} u_t$ converges to a finite value. Then we have, $u_t \to 0$ as $t \to \infty$. There exists a finite number $t_0 \geq 1$, such that for all $t \geq t_0$, we have $u_t \leq 1/2$. Also, we have, for all $T \geq 1$,

$$s_T \leq s < \infty. \tag{684}$$

Then we have,

$$\prod_{t=t_0}^{T} (1 - u_t) = \exp\left\{ \sum_{t=t_0}^{T} \log(1 - u_t) \right\} \tag{685}$$

$$\geq \exp\left\{ -\sum_{t=t_0}^{T} 2 \cdot u_t \right\} \qquad (-2 \cdot x \leq \log(1-x) \text{ for all } x \in [0, 1/2]) \tag{686}$$

$$= \exp\{-2 \cdot s_T\}, \tag{687}$$

which implies that, for all large enough $T \geq 1$,

$$q_T = \left( \prod_{t=1}^{t_0-1} (1 - u_t) \right) \cdot \left( \prod_{t=t_0}^{T} (1 - u_t) \right) \tag{688}$$

$$\geq \left( \prod_{t=1}^{t_0-1} (1 - u_t) \right) \cdot \exp\{-2 \cdot s_T\} \tag{689}$$

$$\geq \left( \prod_{t=1}^{t_0-1} (1 - u_t) \right) \cdot \exp\{-2 \cdot s\} \tag{690}$$

$$> 0. \tag{691}$$

Therefore, we have $\prod_{t=1}^{\infty} (1 - u_t)$ converges to a positive value. $\qquad \square$

**Lemma 17.** *Let $u_t \in (0, 1)$ for all $t \geq 1$. We have $\prod_{t=1}^{\infty} (1 - u_t) = \lim_{T \to \infty} \prod_{t=1}^{T} (1 - u_t) = 0$ if and only if the series $\sum_{t=1}^{\infty} u_t$ diverges to positive infinity.*

*Proof.* **First part.** $\prod_{t=1}^{\infty} (1 - u_t)$ diverges to 0 only if $\sum_{t=1}^{\infty} u_t$ diverges to positive infinity.

Suppose $\prod_{t=1}^{\infty} (1 - u_t)$ diverges to 0. According to Lemma 16, $\sum_{t=1}^{\infty} u_t$ diverges. And since the partial sum $s_T := \sum_{t=1}^{T} u_t$ is monotonically increasing, we have $\sum_{t=1}^{\infty} u_t$ diverges to positive infinity.

**Second part.** $\prod_{t=1}^{\infty} (1 - u_t)$ diverges to 0 if $\sum_{t=1}^{\infty} u_t$ diverges to a positive infinity.

Suppose $\sum_{t=1}^{\infty} u_t$ diverges to positive infinity. According to Lemma 16, $\prod_{t=1}^{\infty} (1 - u_t)$ diverges. And since the partial product $q_T := \prod_{t=1}^{T} (1 - u_t)$ is non-negative and monotonically decreasing, we have $\prod_{t=1}^{\infty} (1 - u_t)$ diverges to 0. $\qquad\square$

The following Lemma 18 indicates that if $\pi_{\theta_t}(a)$ approaches 1 **slowly**, i.e., no faster than $O(1/t)$, then the probability of sampling $a$ forever using on-policy sampling $a_t \sim \pi_{\theta_t}(\cdot)$ is **zero**, i.e., the other actions $a' \neq a$ always have a chance to be sampled.

**Lemma 18.** *Let $\pi_{\theta_t}(a) \in (0, 1)$ be the probability of sampling action $a$ using online sampling $a_t \sim \pi_{\theta_t}(\cdot)$, for all $t \geq 1$. If $1 - \pi_{\theta_t}(a) \in \Theta(1/t^\alpha)$ with $\alpha \in [0, 1]$, then $\prod_{t=1}^{\infty} \pi_{\theta_t}(a) = 0$.*

*Proof.* Suppose $1 - \pi_{\theta_t}(a) \in \Theta(1/t^\alpha)$ and $\alpha \in (0, 1]$. Let $u_t := 1 - \pi_{\theta_t}(a) \in (0, 1)$ for all $t \geq 1$. According to Lemma 15, we have,

$$\sum_{t=1}^{\infty} u_t = \sum_{t=1}^{\infty} (1 - \pi_{\theta_t}(a)) = \infty, \tag{692}$$

i.e., the series diverges to positive infinity. According to Lemma 17, we have,

$$\prod_{t=1}^{\infty} \pi_{\theta_t}(a) = \prod_{t=1}^{\infty} (1 - u_t) = 0, \tag{693}$$

which means it is impossible to sample $a$ forever using on-policy sampling $a_t \sim \pi_{\theta_t}(\cdot)$. $\qquad\square$

**Lemma 19** (Performance difference lemma [32]). *For any policies $\pi$ and $\pi'$,*

$$V^{\pi'}(\rho) - V^{\pi}(\rho) = \frac{1}{1 - \gamma} \cdot \sum_s d_\rho^{\pi'}(s) \cdot \sum_a (\pi'(a|s) - \pi(a|s)) \cdot Q^{\pi}(s, a) \tag{694}$$

$$= \frac{1}{1 - \gamma} \cdot \sum_s d_\rho^{\pi'}(s) \cdot \sum_a \pi'(a|s) \cdot A^{\pi}(s, a). \tag{695}$$

*Proof.* According to the definition of value function,

$$V^{\pi'}(s) - V^{\pi}(s) = \sum_a \pi'(a|s) \cdot Q^{\pi'}(s, a) - \sum_a \pi(a|s) \cdot Q^{\pi}(s, a) \tag{696}$$

$$= \sum_a \pi'(a|s) \cdot \left(Q^{\pi'}(s, a) - Q^{\pi}(s, a)\right) + \sum_a (\pi'(a|s) - \pi(a|s)) \cdot Q^{\pi}(s, a) \tag{697}$$

$$= \sum_a (\pi'(a|s) - \pi(a|s)) \cdot Q^{\pi}(s, a) + \gamma \cdot \sum_a \pi'(a|s) \cdot \sum_{s'} \mathcal{P}(s'|s, a) \cdot \left[V^{\pi'}(s') - V^{\pi}(s')\right] \tag{698}$$

$$= \frac{1}{1 - \gamma} \cdot \sum_{s'} d_s^{\pi'}(s') \cdot \sum_{a'} (\pi'(a'|s') - \pi(a'|s')) \cdot Q^{\pi}(s', a') \tag{699}$$

$$= \frac{1}{1 - \gamma} \cdot \sum_{s'} d_s^{\pi'}(s') \cdot \sum_{a'} \pi'(a'|s') \cdot (Q^{\pi}(s', a') - V^{\pi}(s')) \tag{700}$$

$$= \frac{1}{1 - \gamma} \cdot \sum_{s'} d_s^{\pi'}(s') \cdot \sum_{a'} \pi'(a'|s') \cdot A^{\pi}(s', a'). \qquad\square$$

**Lemma 20** (Value sub-optimality lemma). *For any policy $\pi$,*

$$V^*(\rho) - V^{\pi}(\rho) = \frac{1}{1 - \gamma} \cdot \sum_s d_\rho^{\pi}(s) \cdot \sum_a (\pi^*(a|s) - \pi(a|s)) \cdot Q^*(s, a). \tag{701}$$

*Proof.* See the proof in [2, Lemma 21]. We include a proof for completeness.

We denote $V^*(s) \coloneqq V^{\pi^*}(s)$ and $Q^*(s,a) \coloneqq Q^{\pi^*}(s,a)$ for conciseness. We have, for any policy $\pi$,

$$V^*(s) - V^\pi(s) = \sum_a \pi^*(a|s) \cdot Q^*(s,a) - \sum_a \pi(a|s) \cdot Q^\pi(s,a) \tag{702}$$

$$= \sum_a \left(\pi^*(a|s) - \pi(a|s)\right) \cdot Q^*(s,a) + \sum_a \pi(a|s) \cdot \left(Q^*(s,a) - Q^\pi(s,a)\right) \tag{703}$$

$$= \sum_a \left(\pi^*(a|s) - \pi(a|s)\right) \cdot Q^*(s,a) + \gamma \cdot \sum_a \pi(a|s) \cdot \sum_{s'} \mathcal{P}(s'|s,a) \cdot \left[V^{\pi^*}(s') - V^\pi(s')\right] \tag{704}$$

$$= \frac{1}{1-\gamma} \cdot \sum_{s'} d_s^\pi(s') \cdot \sum_{a'} \left(\pi^*(a'|s') - \pi(a'|s')\right) \cdot Q^*(s',a'). \qquad \square$$

## G   The Intuition of Committal Rate Definition

The following Figure 2 illustrates the intuition of the committal rate definition. Using fixed sampling, we decouple the coupling between sampling and updating, and then focus on only characterizing the aggressiveness of different update rules.

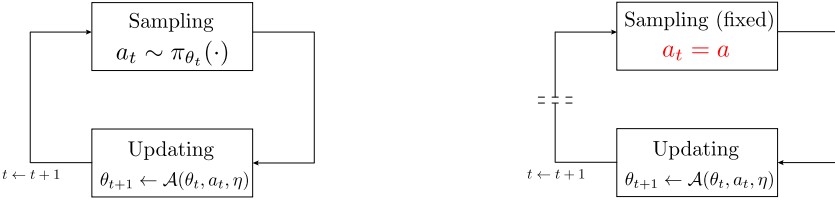

(a) Coupled on-policy sampling and updating.       (b) Decoupled with fixed sampling.

Figure 2:  An illustration for the intuition of the committal rate definition.