# OpenReview forum: "Understanding the Effect of Stochasticity in Policy Optimization"
_NeurIPS.cc/2021/Conference — NeurIPS 2021 Poster_

### Official Review · Reviewer_FFTX · 2021-07-15

**Rating:** 6
**Confidence:** 4

**Summary:**

This paper studied how stochasticity of on-policy evaluation affects the convergence speed of various policy optimization methods. The paper demonstrated an inherent trade-off between fast convergence versus achieving optimality almost surely.

**Limitations And Societal Impact:**

See main review.

**Main Review:**

This paper studied how stochasticity of on-policy evaluation affects the convergence speed of various policy optimization methods and demonstrated an inherent trade-off between fast convergence versus achieving optimality almost surely.

In particular, the authors proposed "committal rate" to characterize how fast the probability $\pi(a|s)$ converges to 1 under some algorithm. When a sub-optimal state action pair has a committal rate greater than 1, the sub-optimal may be sampled forever and hence leads to sub-optimality. On the other hand, the convergence speed to the optimal policy is also characterized by the committal rate of the best state-action pairs. Therefore, any algorithm that fails to give highest committal rate to best actions must face the trade-off between fast convergence versus achieving optimality almost surely.

However, the reviewer has some doubts when verifying some Theorems and the proofs:
1. Theorem 3 says "there is a positive probability that $\pi_{\theta_t}(a) \to 1$, $\forall a\in [K]$". Does it mean to say "$\forall a\in [K]$, there is a positive probability that $\pi_{\theta_t}(a) \to 1$"?

2. At the first glance, the proof of Theorem 3 seems to be saying "When $a$ is always sampled, then with a positive probability $a$ is always sampled". Reorganizing the proofs would fix the issue.

3. The lower bound of the probability of "$a$ is always sampled" (173) is roughly at the order $O(\exp(-|A|/\eta))$. Is it reasonable to say that the event is negligible when $|A|$ is large or $\eta$ is small? If so, the discussion in section 5.1 can be misleading, since the "positive probability" can be fairly small.

Typos:
Line 60, "the a";

**Time Spent Reviewing:**

2

---

> ### Author Response · Authors · 2021-08-10
> **Response to Reviewer FFTX**
>
> We thank the reviewer for very carefully checking the results. The main concerns are addressed as follows.
>
> 1. The reviewer is right. This is a slip, and we will change the language to "$\forall a \in [K]$, there is a positive probability that $\pi_{\theta_t}(a) \to 1$" as you suggested, since this is what we proved in this theorem.
>
> 2. Sorry for the confusion here. We meant to say "When the update is too aggressive (in terms of a repeatedly sampled action $a$ has its probability approaching $1$ too fast), then that action $a$ has a chance to be sampled forever". This is what we proved, and we will reorganize the proof as suggested to avoid such a potential confusion.
>
> 3. The problem of large action number $| \mathcal{A} |$ can be fixed as follows. Consider $| \mathcal{A} | = 2$. Then $ \exp{ (| \mathcal{A} | )} \in \Theta(1)$. The multiple action cases can be reduced to two-action cases by considering $\sum_{a^\prime \not= a^*}{ \pi_{\theta_t}(a^\prime)}$, i.e., the sum of probabilities of all the sub-optimal actions. A proof sketch is as follows.
>
>     First, Eq. (154) becomes the following. For each sub-optimal action $a^\prime \not= a^*$, we have
>
>     $\theta_t(a^\prime) \ge \theta_1(a^\prime) + \eta \cdot r(a^\prime) \cdot t(a^\prime) \ge \theta_1(a^\prime) + \eta \cdot r_{\text{min}} \cdot t(a^\prime)$,
>
>     where $t(a^\prime)$ denotes how many times action $a^\prime$ has been sampled for the first $t$ iterations, and $r_{\text{min}} = \min_{a \not= a^*}{ r(a) }$ denotes the smallest reward.
>
>     On the other hand, Eq. (159) becomes $\theta_t(a^*) = \theta_1(a^*)$.
>
>     According to the Jensen's inequality, we have,
>
>     $\sum_{a^\prime \not= a^*}{ \exp{(\theta_t(a^\prime))} } \ge (K-1) \cdot \exp{ \left( \frac{ \sum_{a^\prime \not= a^*}{ \theta_t(a^\prime) } }{K-1} \right) } \ge (K-1) \cdot \exp{ \left( \frac{ \eta \cdot r_{\text{min}} \cdot t +  \sum_{a^\prime \not= a^*}{ \theta_1(a^\prime) } }{K-1} \right) }$,
>
>     where the last inequality is from lower bounding $\theta_t(a^\prime)$ (the above first inequality) and $t = \sum_{a^\prime \not= a^*}{ t(a^\prime)} $ (sampling sub-optimal actions for the first $t$ iterations; this can be relaxed to sampling sub-optimal actions for $t - c$ times among the first $t$ iterations, where $c \in \Theta(1)$).
>
>     Then Eq. (164) becomes as follows,
>
>     $\sum_{a^\prime \not= a^*}{ \pi_{\theta_t}(a^\prime)} \ge 1 - \frac{ \exp{ (\theta_1(a^*)) }}{ (K-1) \cdot \exp{ \left( \frac{ 1 }{K-1} \cdot ( \eta \cdot r_{\text{min}} \cdot t +  \sum_{a^\prime \not= a^*}{ \theta_1(a^\prime) } ) \right) } + \exp{ (\theta_1(a^*)) } } $.
>
>     Eq. (169) becomes
>
>     $\sum_{a^\prime \not= a^*}{ \pi_{\theta_t}(a^\prime)} \ge \exp{ \left( \frac{ - \exp{ (\theta_1(a^*)) } }{ (K-1) \cdot \exp{ \left( \frac{ 1 }{K-1} \cdot ( \eta \cdot r_{\text{min}} \cdot t +  \sum_{a^\prime \not= a^*}{ \theta_1(a^\prime) } ) \right) } } \right)}$
>
>     The infinite product in Eq. (170) becomes
>
>     $\prod_{t=1}^{\infty}{ \left( \sum_{a^\prime \not= a^*}{ \pi_{\theta_t}(a^\prime)} \right) } \ge \exp{\left( - \frac{ \exp{ (\theta_1(a^*)) } }{K-1} \cdot \frac{1}{ \exp{ ( \frac{1}{K-1} \cdot \sum_{a^\prime \not= a^*}{ \theta_1(a^\prime) } )}} \cdot \int_{t=0}^{\infty}{ \frac{1}{ \exp{( \frac{1}{K-1} \cdot \eta \cdot r_{\text{min}} \cdot t)} } dt } \right)}$
>
>     $= \exp{\left( - \frac{ \exp{ (\theta_1(a^*)) } }{K-1} \cdot \frac{1}{ \exp{ ( \frac{1}{K-1} \cdot \sum_{a^\prime \not= a^*}{ \theta_1(a^\prime) } )}} \cdot \frac{K-1}{ \eta \cdot r_{\text{min}} }  \right)}$
>
>     $= \exp{\left( - \frac{ \exp{ (\theta_1(a^*)) } }{ \eta \cdot r_{\text{min}} } \cdot \frac{1}{ \exp{ ( \frac{1}{K-1} \cdot \sum_{a^\prime \not= a^*}{ \theta_1(a^\prime) } )}}  \right)}$.
>
>     Now since $\exp{ (\theta_1(a^*) )} \in \Theta(1)$, $r_{\text{min}} \in \Theta(1)$, $\eta \in \Theta(1)$, and $\exp{ \left( \frac{1}{K-1} \cdot \sum_{a^\prime \not= a^*}{ \theta_1(a^\prime) } \right) }\in \Theta(1)$, we have the probability of "sampling sub-optimal actions forever" lower bounded by $\frac{1}{ \exp{ (\Theta(1) )} }$, which does not become small as $| \mathcal{A}|$ becomes large.
>
>     For the learning rate, as long as $\eta \in \Theta(1)$ (constant), the above argument holds (the probability is always a constant).
>
>     In practice (e.g., $|\mathcal{A}| = 10$ or $20$), this amount of failure probability is a constant and cannot be ignored (simulations in [12] have been observed to verify this). This leads to linear regret of the stochastic PG based method as pointed also by [12].

---

> > ### Comment · Reviewer_FFTX · 2021-08-23
> > **Response**
> >
> > Thanks for the detailed explanation. The rebuttal addressed my main concerns and I have raised my score accordingly and voted for accepting the paper.

---

### Official Review · Reviewer_JR5d · 2021-07-17

**Rating:** 8
**Confidence:** 4

**Summary:**

Summary

The authors show how the convergence rates of stochastic policy gradient algorithms can differ from the full-information/expected case. Even before diving into the stochastic case they prove a linear convergence rate for NPG in the expected setting. Then for the stochastic setting they show convergence of vanilla PG with probability 1. For NPG (and GNPG) they show that there is a strictly positive probability of converging to a suboptimal solution. To do so they introduce the notion of committal rate, a useful tool, not without some limitations that they discuss.
More generally they explicit the tradeoff between geometry and noise for RL which is a very important topic.
Finally they propose a method using ensembles to show how we can still find the optimal policy for committal algoritms in high probabablity.




**Limitations And Societal Impact:**

Theoretical work, no societal impact to foresee.

**Main Review:**

Clarity
---
I found the paper clear and well written.

Significance
---
I think this is a very important topic as we are seeing more and more convergence results in RL theory, mostly limited to
the expected case. This paper stresses how the stochastic case can be fundamentally different, and how convergence of the expectation does not translate to convergence in probability.

Main review
---

Here are some thoughts and questions I would like the authors to discuss.

1) I think I am missing something about the proof of Thm 7. You show in the appendix that kappa = 1 by showing it is both bigger and smaller than 1.
Thm5 only informs you that if kappa > 1, then probability of sampling any suboptimal action an infinity of times is > 0.
(i) Where is the proof that this is 0 for the case kappa = 1?
(ii) And once this is proven, how do you get to the result that the optimal action will be taken an
infinite number of times?

2) Do you think the 1/t lower bound, was, in some way very unsurprising? For instance, we know that for bandit the regret is Omega(log(t)).
Let us assume that V* - V_t = o(1/t) (small o notation), then we would have by
summation that regret = o(log t) which contradicts the lower bound on regret,
hence V* - V_t = Omega(1/t).
This is a pretty handwavy proof (for bandits), but I suspect it might easy to
formalize and extend to MDPs (as you ust need to extract a bandit problem from
the MDP to show the impossibility).

Under that light, could you then discuss the link between your lower bound result and the one on
regret in bandits?

3) In RL, contrary to supervised learning, the data distribution changes after each action taken, and this introduces new dependencies.
The variance of the parameter theta_t, will be composed of the variance of the gradients, but also of the different covariances between the gradients.
Intuitively, the results of Chung and Thomas et al. even if not formalized that way, could be understood that choosing some baselines can guarantee convergence by ensuring this variance/spread is small (negative correlations) while some other baselines (including potentially the one that minimizes the variance of the gradient) cannot.
I don't think committal rates in their current form can capture this accurately, this difference, do you have any idea how it could be extended to capture these effects?

4) Could you clarify the following point: In the setting where we have a 2-arm bandit with deterministic rewards -1 and 1, then natural policy gradient will be guaranteed to increase the probability of the right action at each time step, and the convergence will be exponential. Therefore are your results about the O(1/t) rate for all MDPs?

I will change my score accordingly depending on how these questions are answered.


**Time Spent Reviewing:**

4

---

> ### Author Response · Authors · 2021-08-10
> **Response to Reviewer JR5d**
>
> We thank the reviewer for appreciating our work and the detailed review. The main concerns are addressed below.
>
> * "Theorem 7": Sorry for the confusion here. The second half of Theorem 7, i.e., " and $\pi_{\theta_t}$ converges to a globally optimal deterministic policy with probability $1$", is not a result of the first half of $\kappa(\text{PG}, a) = 1$. It is a result stated in Theorem 2.
>
> (i) Therefore in Theorem 7 we do not need to show that $\prod_{t=1}^{\infty}{\pi_{\theta_t}(a)} = 0$ if $\kappa = 1$. But it is true that "if $\kappa = 1$, then $\prod_{t=1}^{\infty}{\pi_{\theta_t}(a)} = 0$". For example, if $\pi_{\theta_t}(a) = 1- 1/t$ for all $t \ge 2$, then we have
>
> $\prod_{t=2}^{\infty}{\pi_{\theta_t}(a)} = \lim_{T \to \infty} \prod_{t=2}^{T}{\pi_{\theta_t}(a)} = \lim_{T \to \infty} \left( \frac{1}{2} \cdot \frac{2}{3} \cdots \frac{T-1}{T} \right)  = \lim_{T \to \infty}{ \frac{1}{T} } = 0.$
>
> Replacing $1/t$ with $C/t$ does not change the fact that $\prod_{t=1}^{\infty}{\pi_{\theta_t}(a)} = 0$, but more effort is needed to prove it. This is another example of $\kappa \le 1$ is not necessarily sufficient for global convergence (although we are not aware of any algorithms behaving like this, i.e., converging toward sub-optimal deterministic policies with slower than $O(1/t)$ rate). However, it is necessary as we show in Theorem 5.
>
> (ii) As explained above, the convergence result is from Theorem 2 (not from the fact that $\kappa = 1$). We claimed that $\kappa \le 1$ is a necessary condition rather than a sufficient condition. To make it sufficient, one needs to argue that no algorithm would behave like the above, i.e., converging toward sub-optimal deterministic policies with a slower than $O(1/t)$ rate. Intuitively, we believe this is true, since "slower than $O(1/t)$" implies that the optimal action always has a chance to be sampled and learned.
>
> * "bandit": Thank you for asking this interesting question. The reviewer is right that the bandit lower bound results imply that the convergence speed cannot be faster than $O(1/t)$. However, our results are surprising by noting the following difference with bandit results.
>
> **First**, the bandit lower bounds are information-theoretic, which hold for both adversarial (changing reward signals) and stochastic settings (the difficulty is to estimate the reward accurately enough). Our results hold for even simpler optimization settings: the reward signal is fixed (not changing), and it is deterministic (the difficulty is not from estimation), but the policy gradient is estimated, since on-policy sampling is used. The difficulty and trade-off is then from stochastic optimization (how to balance the aggressiveness of update/convergence rate and almost sure global convergence).  In this sense, the argument that the reviewer proposed cannot be directly used here, since the difficulty is no longer from estimation, and the information-theoretic lower bound construction does not apply.
>
> **Second**, our results provide explanations for the sensitivity to random seeds and the effect of ensemble methods, which have been observed in practice and cannot be explained by using bandit results. This means our results are more useful in some settings where bandit results do not apply.
>
> **Third**, our trade-off results have a condition of $\kappa(a^*) = \kappa(a)$ for at least one sub-optimal action $a$. This is basically saying the algorithm cannot achieve both almost sure global convergence and fast convergence if it cannot tell the difference between the optimal action $a^*$ and any sub-optimal actions. As shown in Update 7 (as well as the example below with rewards $-1$ and $1$), if an oracle baseline is used, then this condition is broken, and Update 7 achieves both almost sure global convergence and linear convergence rate.
>
> Then the results here could be linked to bandit literature if we further extend this result to stochastic reward and transition settings, since using empirical estimation is a way to separate $a^*$ and $a$ (by obtaining small enough confidence intervals). It is not clear what kind of results can be obtained, and whether there exist other better methods (e.g., the baseline study in [12]). We will add discussion with bandit results and leave the above open questions.
>
> * "covariance in Chung and Thomas et al.": We checked [12] Chung and Thomas et al., and found that the conclusion of reducing variance effect using different baselines is not clear ([12] did not make it clear what kind of covariance is good for convergence). To our understanding, the main difficulty is that unlike variance or committal rate (which are scalars, one can compare for example different variances to say which method has smaller/better variance), covariance is more than a scalar (it is a matrix), which makes it difficult to compare different matrices and make quantitative conclusions.
>
> A strategy would be as suggested by the reviewer, i.e., defining some quantities (scalar valued functions with covariances as input) which can characterize "ensuring this variance/spread is small (negative correlations)" effect, such that one can compare the quantities of different methods, and draw connections between those quantities and convergence behaviors. This would make this "effect of covariance" mathematically clear. It is not clear what would be the right quantity here.
>
> Then the next step is then to attempt to see if there is any relationship between the committal rate (which characterize how aggressive an update is) with the above potential quantity (which characterizes how small the covariance spreads). It is definitely worth studying, and answering this question would need a deeper understanding of committal rate and a thorough investigation of covariance.
>
> * "two action example": This is a good example of explaining oracle baseline (Update 7) and the condition in Theorem 10 ($\kappa(a^*) = \kappa(a)$ for at least one sub-optimal action $a$). Our results apply to MDPs with positive rewards ($r(s,a) > 0$ as mentioned in Line $84$). The clarification is as follows.
>
> The two problems with $r_1 = (-1, 1)$ and $r_2 = (1, 3)$ have no difference in true gradient settings, since $r_1 = r_2 - 2$, and the constant shift contributes $\mathbf{0}$ to policy gradient. This means true NPG converges with a linear rate on both of the two problems.
>
> However, they are quite different in on-policy stochastic settings. This is because of unlike true gradient settings (in each iteration the gradient contains the information of all the actions, which make the gradient do "an implicit comparison" between different actions), in stochastic settings, in each iteration one action is sampled, and there is no comparison with other actions (only one action's reward is available). In this case, without baseline, the sampled action's reward is actually **compared with** $0$, and therefore its sign matters. Basically, this means sampling $r_1(1) = -1$ will decrease the first action's probability, and sampling $r_2(1) = 1$ will increase the first action's probability, in these two problems respectively.
>
> If we use baseline $b = 2$ and NPG update on $r_2 = (1, 3)$, then it is equivalent to using NPG without baseline on $r_1$, and this is actually the oracle baseline for $r_2$ (the baseline is in between the optimal action's true reward and the reward of the best sub-optimal action). This will break the condition in Theorem 10 ($\kappa(a^*) = \kappa(a)$ for at least one sub-optimal action $a$), since $\kappa(1) = 0$ ($\pi_{\theta_t}(1) \not\to 1$ since the first action's probability never increases), and $\kappa(2) = \infty$. Therefore, this makes it possible for the oracle baseline to break the condition as well as the trade-off.
>
> This actually motivates how to make the problem easier in stochastic settings ($r_1$ is easier than $r_2$ in this on-policy setting) by using other techniques as mentioned in Section 4.3 (could possibly be bandit techniques as mentioned above, or the baseline study in [12]).

---

> > ### Comment · Reviewer_JR5d · 2021-09-01
> > **Answer**
> >
> > Thank you for the answer and the details. Indeed for the 2 actions example I missed your positive reward assumption, this makes sense!
> > I will keep my score as it is, please in your manuscript make sure to precise which results hold for one-state MDPs (totally fine for counter examples) vs more general MDPs (more interesting for convergence results).

---

### Official Review · Reviewer_ha89 · 2021-07-17

**Rating:** 6
**Confidence:** 4

**Summary:**

This paper aims to answer a fundamental question: what is the key factor that ensures the convergence of stochastic policy gradient (PG) methods? This question is built on the fact that bounded variance appears to be only a sufficient condition for convergence of the PG methods. To begin with, the paper starts by summarizing the existing convergence results of three PG methods, namely Softmax PG, Natural PG (NPG), and Geometry-aware Normalized PG (GNPG), in both the exact and stochastic gradient settings. Then, the paper proposes the notion of “committal rate” as an indicator for characterizing the convergence behavior of the PG methods. Accordingly, for one-state MDPs, the authors show that $\kappa\leq 1$ for all sub-optimal actions serves as a necessary condition for ensuring convergence as well as provide the committal rates of the three PG methods.


**Ethical Concerns:**

This paper does not raise any ethical concerns.

**Limitations And Societal Impact:**

In the checklist, this paper mentions that it has described the limitations, but I did not see this part mentioned in any section.

**Main Review:**

The problem studied in this paper is important and interesting, but I find the overall contribution of this paper quite limited due to the following issues with the theoretical results and presentation:

- The theoretical results are not strong: Most of the main theoretical contributions are established only for one-state MDPs (Theorems 5-8 and 10). While it is okay to present the results in the main text using one-state MDPs for didactic purposes, these results are quite limited as the PG methods are typically used in the context of general MDPs. It remains unknown whether the results of the committal rate and their proof techniques can be extended to the case of general MDPs. On the other hand, the convergence results of Theorems 2-4 are mostly adapted from [12] (with some direct extensions).

- The definition of committal rate in (6) seems problematic: Since $\kappa$ involves $\pi_{\theta_t}(a)$ which is itself a random quantity and could vary with the sample path, then $\kappa$ shall also depend on the underlying sample path. However, throughout the paper, $\kappa$ is viewed as a deterministic value that depends only on the arm and the update scheme. Moreover, in Line 228, it is stated that “Let $t$ denote the number of times action $a$ is sampled after initialization.” This appears inconsistent with the notation of $\pi_{\theta_t}$, where $t$ is the total number of samples. Could the authors comment on this?

- While the committal rate might provide some high-level intuition, it is unclear whether it offers any new insight into the characterization of the convergence behavior. Based on the definition in (6), the committal rate seems a bit tautological when being used to state “convergence to a sub-optimal policy”.

- Regarding the presentation, I find Sections 4.2 and 4.3 not very informative. For example, Theorem 9 follows directly from the definition of the committal rate. Similarly, Theorem 10 also appears quite straightforward and provides very little new insight. As a result, this part can be greatly simplified for clarity.

- In Section 2, it is mentioned that “The main results below extend to general finite MDPs, but for clarity of exposition we move details for general cases to the appendix.” This is a bit misleading as that a large portion of the theoretical results (cf. Theorems 5-8) are established only for one-state MDPs.

- In Section 4.1, two examples are provided to show that $\kappa$ does not serve as a necessary condition, but the examples are a bit artificial. Built on this, another interesting question is: Under which (non-trivial) class of algorithms does $\kappa$ offer a necessary and sufficient condition for convergence?

Other minor comments:
- Line 178: Theorem 2 is adapted from [12]. It would be good to clearly mention this in the main text.
- Line 813: It would be nice to directly provide the proof of Lemma 8 here for completeness.

----- Post-rebuttal updates -----

I’d like to thank the authors for the response. I have raised my score because the authors addressed my main concerns by:

- Clarifying the definition of the committal rate for one-state MDPs

- Clarifying the main results (i.e., Theorems 5, 9, and 10)

- Extending the main result (i.e., Theorems 5) from one-state MDPs to general MDPs


**Time Spent Reviewing:**

10

---

> ### Author Response · Authors · 2021-08-10
> **Response to Reviewer ha89**
>
> We thank the reviewer for carefully reading and asking valuable questions, which helps improve our work. After reading the review, we consider the main concerns are mainly from confusion/misunderstanding rather than real technical weakness. We address all the main concerns as follows, and hopefully the reviewer would reconsider the rating once the issue is clarified.
>
> A large portion of the paper is about constructing counterexamples which shows that the stochastic policy optimization algorithms do not perform well as in the true gradient setting. A counterexample for one-state MDP problem is also a counterexample for general MDPs (since they are special cases). Therefore, there is no loss of generality by establishing negative results using one-state MDPs.
>
> For completeness, we still provide ideas about how the similar techniques would apply to general MDPs as follows.
>
> * "results are not strong, general MDPs": Since $r(s, a) \in \Theta(1)$ implies $Q^{\pi_\theta}(s,a) \in \Theta(1)$ as mentioned in Line $696$. For the NPG update, we have after sampling one action $a$ under one state $s$ for $t$ times, Eq. (161) holds, i.e., for all $a^\prime \not= a$, $\pi_{\theta_t}(a^\prime | s) \le \frac{C}{\exp{(c \cdot (t-1))}}$. This makes $1 - \pi_{\theta_t}(a | s)$ at the order of $1 - 1/e^t$, and $\kappa(\text{NPG}, (s,a)) = \infty$.
>
> Using similar arguments in Theorem 3, sampling action $a$ once under state $s$ will make its probability $\pi_{\theta_t}(a | s)$ increase aggressively, such that $\prod_{t=1}^{\infty}{\pi_{\theta_t}(a | s)}$ is lower bounded by $\prod_{t=1}^{\infty}{(1 - 1/e^t)} > 0$, which means "for each action $a$, with positive probability, under at least one state $s$, $a$ will be sampled forever".
>
> We will be happy to incorporate a detailed proofs into the final version, though the proof will be mostly routine, and the one-state MDP results already show the main findings of this work.
>
> * "results are not strong, [12]": It is true that our work is inspired by [12], as we also noted at the beginning of Section 3. However, the contribution is significant rather than incremental, since our work has the following differences to that in Chung et al.:
>
> **First**, we introduce a quantity (committal rate) to set up the boundary of almost sure global convergence, while [12] provided intuitions without contemplating this quantitative characterization.
>
> **Second**, we use committal rate as an analysis tool to reveal the geometry-convergence trade-off, and to explain the sensitivity to random seeds and the effect of ensemble methods, while [12] does not explain any of those phenomena.
>
> **Third**, the results in [12] are mostly for $2-$ and $3-$armed bandits, and for two algorithms (softmax PG and NPG). Our results are more general: for general $K$ actions (will add extensions to MDPs as mentioned above), and for multiple algorithms (softmax PG, NPG, GNPG, baselines, SAMBA, etc., as summarized in Table 1). Due to our general approach, definitions and tools, it is likely that one can also obtain extensions of our results to cover more algorithms and settings.
>
> Most importantly, the above differences are based on the new concept of committal rate. We will add the above discussion to Section 3.
>
> * "problematic definition of committal rate": We think this is the source of main confusion/misunderstanding. Sorry for being not clear enough here. By "Let $t$ denote the number of times action $a$ is sampled after initialization." in the definition, we meant to say "let $a$ be the sampled action for the first $t$ iterations". Therefore, in this definition, we fix the "sampling" procedure to be "always sampling one action $a$", and this makes the $\kappa$ not depend on the sample path.
>
> This also makes the committal rate a property that "characterizes how aggressive an update is", and it does not say anything about how to sample (the sample path is always sampling one same action).
>
> In a nutshell, an algorithm here is really just an update rule with leaving the action choice open.
>
> * "tautological": The necessary condition in Theorem 5 is then saying: "if an update is too aggressive (in terms of becoming committal too fast, a repeatedly sampled action $a$ would have its probability $\pi_{\theta_t}(a)$ approaching $1$ too fast), then this aggressive update will have a bad effect on the trajectory $\pi_{\theta_t}$, such that there is a positive probability that $a$ will be sampled forever if we run the algorithm/update using on-policy sampling $\pi_{\theta_t}$".
>
> Therefore, the committal rate is not about "convergence to a sub-optimal policy", and it is about "how aggressive an update is". The necessary condition then means "an aggressive update with $\kappa > 1$ is guaranteed to converge to sub-optimal deterministic policy with positive probability", which is decidedly not "tautological".
>
> We learned from this how the current presentation could lead to a major confusion. Thank you for raising this point, and we will definitely add the above clarification into the revision. Especially, we will change the sentence "Let $t$ denote the number of times action $a$ is sampled after initialization" and state its meaning in a much more clear way to avoid such a confusion.
>
> * "not informative": This is a consequence of the above confusion. Theorem 9 relies on the relationship between $1 - \pi_{\theta_t}(a^*)$ and the sub-optimality gap $\left( \pi^* - \pi_{\theta_t} \right)^\top r$ (and the proofs in the appendix show that this relationship hold under general MDPs), and also the fact that "pulling $a^*$ always and do update will lead to larger $\pi_{\theta_t}(a^*)$ than the algorithm itself" (e.g., Eq. (222) shows that after pulling one action $a$ and one PG update, we have $\pi_{\theta_{t+1}}(a) \ge \pi_{\theta_{t}}(a)$; the algorithm does not necessarily always pull $a^*$). Therefore Theorem 9 is not "directly from the definition".
>
> The reviewer considered Theorem 10 not informative because of the confusion on previous results, including Theorems 5 and 9. As we clarified, the committal rate and 5 and 9 are not simple and direct, nor is Theorem 10 as a result. It is true that the proof of Theorem 10 is short, but this is based on all the previous results we established in this work.
>
> Based on the condition of Theorem 10 ($\kappa(a^*) = \kappa(a)$ for at least one sub-optimal action $a$), we are studying how to leverage a better algorithmic design to break such a condition to achieve both global convergence and fast convergence rate. The preliminary results are promising. This indicates that the results in Section 4.2 and 4.3 would provide useful information and new insight for how to design principled algorithms in on-policy stochastic settings.
>
> * "necessary and sufficient conditions": Thank you for asking this really interesting, and in our opinion, important question. Our ideas on this are as follows. As suggested, under "the class of algorithms converging to deterministic policies asymptotically" (why this is non-trivial will be argued below), we conjecture that if $\kappa(a^*) = \kappa(a)$ for at least one sub-optimal action $a$, then $\kappa \in (0, 1]$ offers a necessary and sufficient condition for global convergence with **polynomial rates**.
>
> **First**, since we are in on-policy settings, if we care about asymptotic convergence, then asymptotically the policy would converge to some deterministic policies (otherwise with non-deterministic policies, there is always randomness of sampling, which is contradiction to convergence).
>
> **Second**, Theorem 9 can potentially be strengthened with the claim that $\kappa(a^*) \ge \alpha$ is a sufficient and necessary condition for global convergence with rate $O(1/t^\alpha)$ ($\alpha > 0$). Currently, it is a necessary condition because of Eq. (239), i.e., the sub-optimality gap dominating $1 - \pi_{\theta_t}(a^*)$. The observation here is that a reversed version of Eq. (239) still holds. If the reward $r$ is bounded ($r \in (0,1)^K$), then we have $r(a^*) - r(a) \le 1$, which will lead to
>
> $\left( \pi^* - \pi_{\theta_t} \right)^\top r = \sum_{a \not= a^*}{ \pi_{\theta_t}(a) \cdot \left( r(a^*) - r(a) \right) } \le 1 - \pi_{\theta_t}(a^*).$
>
> This suggests that if $\kappa(a^*) \ge \alpha$ (i.e., $1 - \pi_{\theta_t}(a^*) \in O(1/t^\alpha)$), then $\left( \pi^* - \pi_{\theta_t} \right)^\top r$ would be $O(1/t^\alpha)$, i.e., global convergence with $O(1/t^\alpha)$. However, a small gap here is "$\kappa(a^*) \ge \alpha$ means $1 - \pi_{\theta_t}(a^*) \in O(1/t^\alpha)$ if we force sampling $a^*$ always", and whether this is equivalent to "$1 - \pi_{\theta_t}(a^*) \in O(1/t^\alpha)$ if we run the algorithm using on-policy sampling" is not clear. This gap requires further deeper understanding of the relationship between committal rate and the bahaviours of algorithms using on-policy sampling.
>
> **Finally**, this would be a sufficient and necessary condition for polynomial rate of $O(1/t^\alpha)$ if the above results go through. A slower rate would be poly-log rate (or even slower) like $O(1/\log{t})$, which is not interesting. This is because of if we specify an $\epsilon$ threshold on sub-optimality gap, we would need $\exp\{ (1/ \epsilon) \}$ iteration to achieve that, which does not scale up for moderately small $\epsilon$. Therefore, considering polynomial rate is reasonable enough.
>
> Since we think that the question discussed here is important, we will arrange space in Section 4 to add the above discussion as the reviewer suggested. Note that this discussion indicates that the committal rate theory can provide useful information and insights for future research, thus strengthening the paper. We hope the reviewer will consider this clarification helpful.
>
> * Line $178$: We mentioned [12] in the appendix, and we will mention it in the main text of Theorem 2.
> * Line $813$: A standard analysis based on references will be added to the appendix as suggested.

---

### Official Review · Reviewer_RnW5 · 2021-07-18

**Rating:** 7
**Confidence:** 4

**Summary:**

This paper studies the convergence of various policy gradient methods. The authors introduce a novel conception, the committal rate, to characterize the convergence speed and convergence quality of policy gradient methods, which is defined as the speed of the policy on a particular action converges to 1. For a simple MDP with 1 state, the authors show that the committal rate can explain the phenomenon that faster convergent algorithms using true policy gradients can be dominated by slower ones in the stochastic setting.


**Limitations And Societal Impact:**

One drawback is the lack of discussion of how the proofs of Section 3 can be generalized to more common MDPs, which have more than 1 state. It is unclear whether the conclusions made in Section 3 about the committal rates of PG and NPG hold in more general cases.

I do not see any potential negative societal impact directly related to this work


**Main Review:**

In general, the paper is well written and easy to understand. The insight on the committal rate is also novel and very helpful for researchers to understand the connections and differences among different policy gradient methods. One drawback is the paper did not provide the discussion on the generalization to more common MDPs with more than one state. Another point that could be potentially more interesting as stated in the conclusion part is to identify the sufficient conditions and necessary conditions for such type of methods to converge using the committal rate theory.

Why do we need the importance sampling estimator since the action is chosen from the same policy we want to evaluate?

In Definition 2, is it correct that you need to specify that kappa (or equivalently alpha) is a constant that does not depend on t? Otherwise, it is easy to construct counter examples by setting $\kappa(A,a)$ to be a function of t, where the examples presented in Line 233-236 are not true. Moreover, in Line 235, it should be “if $1-\pi_{\theta_t}(a) \in O(1)$, then $\kappa(A,a)=0$”, since $\pi_{\theta_t}(a)$ itself can be O(1), just like in the previous two examples you mentioned.

In the proof of Theorem 5, at the last line, how do you guarantee that $\pi_{\theta_t}(a)>0$ for all t? Does this mean the general algorithm A here should not have any truncation procedures?

Theorem 6 claims that $\kappa(A,a)=\infty$ for all actions. By the definition of kappa, does this mean $\pi_{\theta_t}(a)$ converges to 1 for all actions? However, this obviously seems wrong.

Typos:
Line 60: “the a variant ...”
Line 129: “can *be* lower bounded”


**Time Spent Reviewing:**

6

---

> ### Author Response · Authors · 2021-08-10
> **Response to Reviewer RnW5**
>
> We thank the reviewer for appreciating our contributions and the valuable comments. We address all the main concerns as follows.
>
> The reviewer considered that establishing results only for one-state MDPs is a weakness of the paper. Here we would like to note that a large portion of the paper is about constructing counterexamples which shows that the stochastic policy optimization algorithms do not perform well as in the true gradient setting. A counterexample for one-state MDP problem is also a counterexample for general MDPs (If one algorithm is performing well on general MDPs, then it must also perform well on one-state MDPs, which are special cases). Therefore, there is no loss of generality by establishing negative results using one-state MDPs.
>
> For completeness, we still provide ideas about how the similar techniques would apply to general MDPs as follows.
>
> * "general MDPs": Since $r(s, a) \in \Theta(1)$ implies $Q^{\pi_\theta}(s,a) \in \Theta(1)$ as mentioned in Line $696$. For the NPG update, we have after sampling one action $a$ under one state $s$ for $t$ times, Eq. (161) holds, i.e., for all $a^\prime \not= a$, $\pi_{\theta_t}(a^\prime | s) \le \frac{C}{\exp{(c \cdot (t-1))}}$. This makes $1 - \pi_{\theta_t}(a | s)$ at the order of $1 - 1/e^t$, and $\kappa(\text{NPG}, (s,a)) = \infty$. In short, replacing $r(a)$ in the proof with $Q^{\pi_\theta}(s,a)$ (or sampled cumulative reward) and using the mentioned fact that $Q^{\pi_\theta}(s,a) \in \Theta(1)$ will give us results similar to Eq. (161).
>
> Using similar arguments in Theorem 3, sampling action $a$ once under state $s$ will make its probability $\pi_{\theta_t}(a | s)$ increase aggressively, such that the infinite product $\prod_{t=1}^{\infty}{\pi_{\theta_t}(a | s)}$ is lower bounded by $\prod_{t=1}^{\infty}{(1 - 1/e^t)} > 0$, which means "for each action $a$, with positive probability, under at least one state $s$, the action $a$ will be sampled forever".
>
> We will be happy to incorporate a detailed proofs into the final version, though the proof will be mostly routine, and the one-state MDP results already show the main findings of this work.
>
> * "why importance sampling": This is because of the policy gradient method requires estimating $Q^{\pi_{\theta}}(s, a)$ for each $(s,a)$ rather than a scalar $V^{\pi_{\theta}}(s)$ to update each $\theta(s,a)$ (policy gradient theorem). In one-state MDPs, $Q(s,a)$ becomes $r(s,a)$ and $V^{\pi_{\theta}}(s)$ becomes $\pi_\theta^\top r$. Without importance sampling, only a scalar $\pi_\theta^\top r$ can be estimated, which is not enough to do PG updates for $\theta \in \mathbb{R}^K$.
>
> * "$\kappa$ does not depend on $t$": We define $\kappa$ in this way on purpose, the $\lim_{t \to \infty}$ in the definition removes the dependence on $t$ and this means we care the asymptotic speed of how aggressive an update is. This asymptotic behavior matters for almost sure global convergence and convergence rate. Consider one example suggested by the reviewer (to our understanding):
>
> $\pi_{\theta_t}(a) = 1 - 1/t$, if $t < T$;
>
>  and $\pi_{\theta_t}(a) = 1 - 1/e^t$, for all $t \ge T$.
>
> Then in this case, only the asymptotic part "for all $t \ge T$" matters, since we will have $\prod_{t=T}^{\infty}{\pi_{\theta_t}(a)} = \prod_{t=T}^{\infty}{(1 - 1/e^t)} > 0$. This implies "with positive probability, $a$ will be sampled forever after a finite time $T > 0$", which implies both asymptotic convergence toward $\pi_{\theta_t}(a) \to 1$ as $t \to \infty$ as well as the speed of convergence toward that deterministic policy (with $\pi_{\theta}(a) = 1$) is $1/e^t$.
>
> * $1 - \pi_{\theta_t} \in O(1)$: Sorry for the confusion here. The reviewer is right. We wanted to say "if $\pi_{\theta_t}(a)$ will not approach $1$ asymptotically, then $\kappa(\mathcal{A}, a) = 0$". We will fix it.
>
> * $\pi_{\theta_t} > 0$: For algorithms that use the parameterized policies in Eq. (2) (including softmax PG, NPG and GNPG), $\pi_{\theta_t} > 0$ is guaranteed by the softmax transform since $e^x > 0$ for all $x \in \mathbb{R}$. For algorithms that do not use similar transforms, an explicit truncation is needed, e.g., see Line $792$ after Update 11 (SAMBA algorithm).
>
> * "$\kappa = \infty$": No. This means "for each action $a \in [K]$, with positive probability, $\pi_{\theta_t}(a) \to 1$ as $t \to \infty$". Our current language in Theorem 6 is a slip, and we will change it to "$\forall a \in [K]$, there is a positive probability that $\pi_{\theta_t}(a) \to 1$" as suggested by another reviewer FFTX, since the later one is what we proved in Theorem 6.
>
> * Thank you for catching typos, which will be fixed.
>
> *"necessary and sufficient condition": Another reviewer ha89 asked a similar question. That would require restricting algorithms in a meaningful way (to exclude the behaviours like staying and wandering). We use our committal rate results to give some ideas as follows. Fully answering this questions requires further study.
>
> If we consider all the algorithms in "the class of algorithms converging to deterministic policies asymptotically" (why this is non-trivial will be argued below), we conjecture that if $\kappa(a^*) = \kappa(a)$ for at least one sub-optimal action $a$, then $\kappa \in (0, 1]$ offers a necessary and sufficient condition for global convergence with **polynomial rates**.
>
> **First**, since we are in on-policy settings, if we care about asymptotic convergence, then asymptotically the policy would converge to some deterministic policies (otherwise with non-deterministic policies, there is always randomness of sampling, which is contradiction to convergence).
>
> **Second**, Theorem 9 can potentially be strengthened with the claim that $\kappa(a^*) \ge \alpha$ is a sufficient and necessary condition for global convergence with rate $O(1/t^\alpha)$ ($\alpha > 0$). Currently, it is a necessary condition because of Eq. (239), i.e., the sub-optimality gap dominating $1 - \pi_{\theta_t}(a^*)$. The observation here is that a reversed version of Eq. (239) still holds. If the reward $r$ is bounded ($r \in (0,1)^K$), then we have $r(a^*) - r(a) \le 1$, which will lead to
>
> $\left( \pi^* - \pi_{\theta_t} \right)^\top r = \sum_{a \not= a^*}{ \pi_{\theta_t}(a) \cdot \left( r(a^*) - r(a) \right) } \le 1 - \pi_{\theta_t}(a^*).$
>
> This suggests that if $\kappa(a^*) \ge \alpha$ (i.e., $1 - \pi_{\theta_t}(a^*) \in O(1/t^\alpha)$), then $\left( \pi^* - \pi_{\theta_t} \right)^\top r$ would be $O(1/t^\alpha)$, i.e., global convergence with $O(1/t^\alpha)$. However, a small gap here is "$\kappa(a^*) \ge \alpha$ means $1 - \pi_{\theta_t}(a^*) \in O(1/t^\alpha)$ if we force sampling $a^*$ always", and whether this is equivalent to "$1 - \pi_{\theta_t}(a^*) \in O(1/t^\alpha)$ if we run the algorithm using on-policy sampling" is not clear. This gap requires further deeper understanding of the relationship between committal rate and the bahaviours of algorithms using on-policy sampling.
>
> **Finally**, this would be a sufficient and necessary condition for polynomial rate of $O(1/t^\alpha)$ if the above results go through. A slower rate would be poly-log rate (or even slower) like $O(1/\log{t})$, which is not interesting. This is because of if we specify an $\epsilon$ threshold on sub-optimality gap, we would need $\exp\{ (1/ \epsilon) \}$ iteration to achieve that, which does not scale up for moderately small $\epsilon$. Therefore, considering polynomial rate is reasonable enough.
>
> Since we think that the question discussed here is important, we will arrange space in Section 4 to add the above discussion. Note that this discussion indicates that the committal rate theory can provide useful information and insights for future research, as the reviewer also mentioned.

---

> > ### Comment · Reviewer_RnW5 · 2021-08-25
> > **Re: Response to Reviewer RnW5**
> >
> > Thank you for the response. It addressed my previous questions and I still think this paper deserves publication. I will keep my score as “accept”.

---

### Decision · Program_Chairs · 2021-09-27

**Decision:**

Accept (Poster)

**Comment:**

The paper studies the effect of stochastic on policy optimization in Reinforcement Learning, i.e., the impact of not having the true gradient. The paper argues that there are significant differences between the case of true gradient and a stochastic version. While this is not surprising, the reviewers agree that the paper makes a good contribution to the literature by pointing out the differences. There was some concern that many of the results are for a one-state MDP, but the reviewers felt that the paper makes some interesting points which can  potentially stimulate further work.